# The Mantle $Fe^{3+}/\Sigma Fe$ Ratio Has Doubled Since the Early Archean

Xiao-Xi Zhu [1,2], Wen-Yong Duan [1,3] ✉, Taras Gerya[1], Xin Zhou [1] & Jia-Cheng Tian[1]

How mantle redox state developed, particularly the mantle source associated with mid-ocean ridge-like settings, remains a subject of ongoing debate. Here, we employ thermodynamic-thermomechanical numerical simulations to explore the redox properties of melts formed under mid-ocean ridge-like settings in both Archean and modern conditions. By comparing these results with a global database of mid-ocean ridge-like rocks extending back to 3.8 Ga, we reconstruct the mantle's redox evolution since the early Archean. Using the whole-rock $Fe^{3+}/\Sigma Fe$ ratio as a robust redox proxy, derived from integrated numerical modeling and thermodynamic inversion, we find that the mantle's average $Fe^{3+}/\Sigma Fe$ ratio has approximately doubled since the early Archean. Our calculations further indicate that ultra-low-oxygen-fugacity mantle domains in modern oceanic lithosphere reflect an initially reduced origin rather than deeper or hotter melting. Our results suggest that Earth's oxygenation and tectono-magmatic evolution may have been coupled.

The redox state of Earth is a critical factor in determining its habitability, shaping the chemical composition of the surface and atmosphere while influencing biological activity and interior geological processes[1–5]. As the largest component of Earth's mass, the mantle plays a pivotal role in regulating the planet's redox balance[2,4]. This regulation is achieved through processes such as mantle convection, degassing, and the cycling of redox materials between Earth's interior and surface[4–7]. These mechanisms not only govern the mantle chemical and physical properties but also have far-reaching effects on Earth's surface environment, atmospheric composition, and conditions necessary for sustaining life[4–10].

The mechanisms controlling the secular redox evolution of the MORB-source mantle have been debated for decades[6,7,11–15]. Notably, an increase in mantle oxygen fugacity ($fO_2$) exceeding 0.5 log units relative to the QFM (quartz-fayalite-magnetite) buffer could significantly influence atmospheric oxygen levels, potentially acting as a catalyst for the Great Oxidation Event (GOE)[4]. Some studies suggest that the mantle's redox state has remained unchanged since the Archean or Hadean[11,12]. For example, the modern mantle's $fO_2$ is considered to be comparable to Archean mantle based on similar V/Sc (vanadium/

scandium) ratios in basalts[12]. However, growing petrological evidence from basalts, eclogites, komatiites, and ancient mantle[6,7,13–15] indicates that the Archean mantle $fO_2$ was ~1–1.5 log units lower than QFM buffer, even 4–5 log units lower than modern levels.

Recent research, however, hypothesizes that the increasing $fO_2$ inferred from petrological records might be caused by the mantle gradual cooling[11,15]. During the Archean, a higher mantle potential temperature ($Tp$) likely resulted in deeper melt extraction depths[11]. The garnet-bearing mantle extends beyond ~8–10 GPa, but metal saturation at around these pressures begins to buffer the mantle $fO_2$, which generally decreases with increasing pressure. At pressures ($P$) < 3–3.5 GPa, the $fO_2$-depth profile is more complex, as demonstrated by phase equilibria and empirical models[15–17]. Previous studies have attempted to account for the pressure dependence of mantle redox conditions, but with differing approaches and limitations. A correction for the decrease in $fO_2$ with increasing pressure was explicitly applied, revealing significant differences between Archean and modern mantle-derived samples[6]. In contrast, a linear correction was applied to V/Ti-derived $fO_2$ values based on melt evolution during ascent[11]. However, since V/Ti reflects the redox state of the source

[1]Department of Earth and Planetary Sciences, Swiss Federal Institute of Technology, Zurich, Switzerland. [2]State Key Laboratory of Tropical Oceanography, South China Sea Institute of Oceanology, Chinese Academy of Science, Guangzhou, China. [3]Institute of Geology, Mineralogy and Geophysics, Faculty of Geosciences, Ruhr-University Bochum, Bochum, Germany. ✉e-mail: wenyong.duan@rub.de

condition rather than the melt after evolution, such a correction is inappropriate. Furthermore, both studies likely overestimated the $fO_2$ of Archean samples, which are typically derived from greater depths due to higher $Tp$, by extrapolating the $fO_2$−$P$ relationship established for garnet peridotite to pressures down to 1 GPa[6] or to surface conditions[11]. This extrapolation has since been shown to be invalid by phase equilibrium calculations[17]. It is also worth noting that the phase equilibrium is based on solid-solid relationships[17], and the calculated results may also differ from those of melt-bearing systems[15].

For a long time, the redox state of the mantle and mantle-derived melts has been characterized primarily using either oxygen fugacity[6,7,11,13–15,18] or geochemical ratios such as V/Sc [12]. As mentioned above, although previous studies have attempted to correct for the effects of temperature ($T$) and $P$ on $fO_2$[6,11], different calibration approaches have led to significant discrepancies in the results. Fundamentally, these reliances have sparked controversy over whether the mantle's redox state has evolved over time. Oxygen fugacity is inherently complex, being influenced by multidimensional factors such as $P$, $T$, whole-rock composition, and oxygen content (e.g., $Fe^{3+}$/$\Sigma Fe$)[2,16].

Fig. 1A shows a $P$-$T$ phase diagram of a melt-bearing mantle calculated by this study, using the whole-rock composition of the depleted mantle source of MORB (Supplementary Data 1). Basaltic melts are produced on the high-temperature side and exhibit varying $Fe^{3+}$/$\Sigma Fe$ ratios (different-color areas). The rock system exhibits varying oxygen fugacity under different $P$-$T$ conditions, but with no fixed slope. Notably, $fO_2$ decouples from the $Fe^{3+}$/$\Sigma Fe$ in the generated melt. Especially above 1–1.5 GPa conditions, $fO_2$ is pressure-dependent and decreases with increasing pressure (Fig. 1A). At low pressures, the $fO_2$ slope changes when orthopyroxene disappears. But the $Fe^{3+}$/$\Sigma Fe$ ratio in the melt remains primarily temperature-dependent, decreasing with increasing temperature. We selected two mantle potential temperatures to illustrate the complexity of the $P$-$T$-$fO_2$ relationships. The pink

star represents the modern $Tp$ (1350 °C)[19], while the yellow star represents the Archean $Tp$ (1500 °C)[19,20]. The $P$-$T$ conditions of the melt source regions (red star 1 and yellow star 3) are based on previous observations[11,13,18]. The pink star, when adiabatically extrapolated to the surface along a 0.5 °C/km geotherm, shows an increase in $fO_2$ of more than 0.7 log units. In contrast, the yellow star, representing the Archean mantle, exhibits a total $fO_2$ increase of only -0.4 log units along the same geotherm to the surface (Fig. 1A).

Notably, when the calibrated $fO_2$ and $Fe^{3+}$/$\Sigma Fe$ ratio of the Archean mantle melt are adiabatically extrapolated to the average melt extraction pressure of the modern mantle (-1.2 GPa)[18], the two conditions display comparable $fO_2$ values, yet the bulk melt $Fe^{3+}$/$\Sigma Fe$ ratio differs by -0.02, highlighting the critical role of temperature. Upon further extrapolation of the Archean melt to the surface, its $fO_2$ remains unchanged relative to its value at 1.2 GPa. This indicates that the relationships among $P$-$T$-$fO_2$ and $Fe^{3+}$/$\Sigma Fe$ are highly nonlinear and cannot be accurately normalized or extrapolated using simple linear formulations. As a result, variations in oxygen fugacity do not necessarily reflect the whole-rock oxygen content (or redox budgets)[21,22] changes, and the reverse is also true. Drawing accurate conclusions requires careful control of these thermodynamic variables.

Similarly, direct elemental proxies, like V/Sc ratio[12], are also affected by $P$, $T$, and whole-rock composition[13,18]. Variations in $P$-$T$ conditions can cause the V/Sc ratio to shift by several units at the same $fO_2$ (Fig. 1B). When comparing records across different geological periods, it is essential to consider these complexities (Fig. 1B). Therefore, neither oxygen fugacity nor elemental ratios alone serve as ideal indicators for historical mantle redox trends.

Another reference framework for representing the redox state of the mantle is the redox budget[21,22], which is primarily governed by the gain or loss of electrons by multivalent elements (such as iron, carbon, sulfur, etc.) within the rock system. This approach relies more on mass and charge conservation than on other thermodynamic parameters,

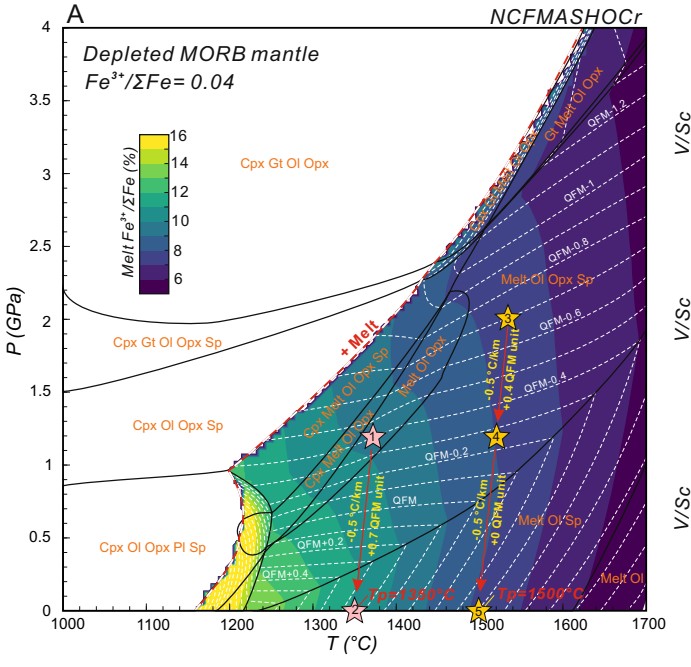
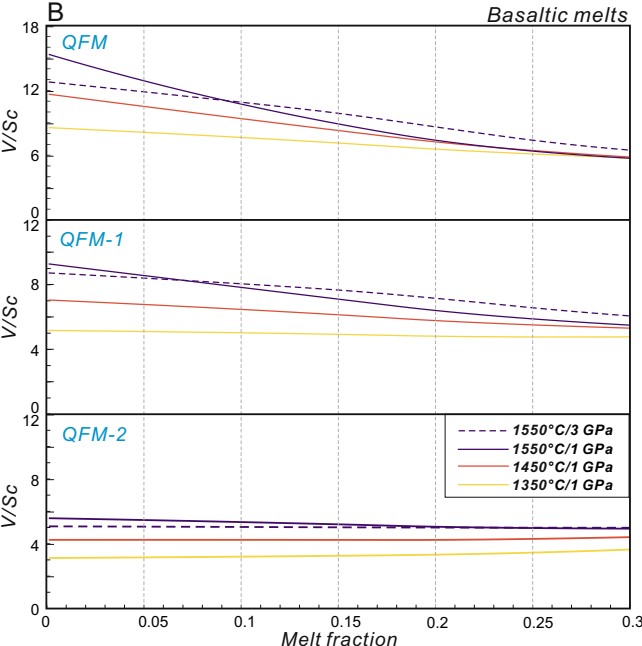

**Fig. 1 | The influence of different thermodynamic conditions on oxygen fugacity and V/Sc. A** $P$-$T$ phase diagram of a depleted MORB mantle (whole-rock $Fe^{3+}$/$\Sigma Fe = 0.04$[25,26]); solid black lines denote mineral-in and mineral-out reaction boundaries; white dashed lines indicate variations in the oxygen fugacity of the rock system relative to the QFM buffer under different $P$-$T$ conditions. The region to the right of the thick solid red line marks the melt-producing region (i.e., the supersolidus region). The color gradient reflects changes in $Fe^{3+}$/$\Sigma Fe$ in the melt as a function of temperature; yellow and pink stars indicate the contrasting $fO_2$ results derived from correcting typical MORB melting $P$-$T$ conditions[11,13,18] for modern and Archean settings upward along an adiabat toward the surface, despite both sharing the same mantle source. **B** Evolution of V/Sc at different temperatures, pressures, and melting conditions under varying $fO_2$ conditions[13,18]; V/Sc can vary by several units under the same oxygen fugacity but different $P$-$T$ conditions, indicating that $P$-$T$ effects must be taken into account.

allowing for a quantitative expression of a rock's redox state. Among these multivalent elements, iron is the most important, as the concentrations of other redox-sensitive elements are typically very low. Under uppermost mantle reference conditions[22], sulfur in the MORB mantle source exists predominantly as $S^{2-}$, while carbon is mainly present as $C^{4+}$ that can remain stable even under the reducing conditions (e.g., QFM-1.5 to -2 buffer at 2–3 GPa)[23,24]. Given these low concentrations and stable mantle reference states of carbon and sulfur, variations in the primary $Fe^{3+}/\Sigma Fe$ in mantle become particularly important for tracking changes in the redox budget as a redox proxy. Therefore, an accurate and quantitative approach involves tracking changes in the whole-rock $Fe^{3+}/\Sigma Fe$ of melts and corresponding mantle sources under well-constrained $P$-$T$-$fO_2$ conditions.

Here, we performed a series of advanced thermodynamic-thermomechanical numerical experiments to calculate the redox state of melts generated under varying $Tp$ and whole-rock $Fe^{3+}/\Sigma Fe$ ratios.

To investigate the redox evolution of mantle-derived melts, we adopted an integrated approach combining thermomechanical and thermodynamic modeling. First, thermomechanical simulations were performed to constrain the $P$-$T$ conditions of melt generation under varying mantle potential temperatures and extension rates. These simulations are governed by the conservation of momentum, mass, and energy, and fully account for rheology and melt extraction processes. Subsequently, thermodynamic modeling was conducted. Under open-system conditions, the thermodynamic model tracks the generation and extraction of melts from a depleted mantle source, and iteratively calculates the evolution of $Fe^{3+}/\Sigma Fe$ as a function of melt fraction and redox conditions (see Methods for details). This integrated approach allows for a quantitative assessment of the redox state of mantle melts and their residual sources under both modern and Archean thermal regimes.

Additionally, we compiled a petrological database of MORB-like samples from up to 3.8 Ga and utilized thermodynamic and geochemical methods to estimate their whole-rock $Fe^{3+}/\Sigma Fe$ ratios. Our ultimate goal was to quantify the $Fe^{3+}/\Sigma Fe$ in mantle as a single redox-proxy variable, with particular emphasis on MORB-like mantle sources (identified based on their depleted REE patterns and Nb/La ratios; see Methods for details), which are the most debated[6,7,11,12,] and yield the smallest uncertainties in model results—offering a robust means to reconstruct the redox history of the mantle since the early Archean. Our integrated approach combines empirical observations with high-resolution numerical simulations. The simulations allow for in-depth analysis of how thermodynamic and thermomechanical conditions regulate redox mechanisms and control the redox budget of mantle and the derived rocks, while empirical observations provide the most compelling evidence supporting these processes. The two lines of evidence reinforce each other. Building on this, we present a robust quantification of $Fe^{3+}/\Sigma Fe$ in mantle sources as a dynamic variable evolving through geological time. Our results indicate that the $Fe^{3+}/\Sigma Fe$ ratio of MORB sources, used here as a proxy for the mantle redox budget, has approximately doubled since the early Archean.

## Results and discussion
### Modern and Archean MORB redox modeling
Basalt is suggested as a better proxy for the mantle's redox state compared to komatiite, because its $P$-$T$ estimates in natural records carry relatively small uncertainties[11], and because its source characteristics and the melt extraction processes leading to MORB generation are relatively well constrained. Numerical simulations indicate that even when the mantle potential temperature reaches 1600 °C—considered by some studies to be representative of the early Archean mantle[19], though others argue this is too high[20]—MORB-like rocks still dominate the composition of the oceanic crust. We thus conducted a series of numerical experiments on mid-ocean ridge spreading for

different $Tp$ values representative of modern and Archean conditions (Supplementary Data 2). This forward model is based on simulations conducted within a coupled thermo-mechanical-thermodynamic framework, using prescribed geodynamic conditions and rock compositions. The modeling does not rely on V-based oxybarometers to determine the $Fe^{3+}/\Sigma Fe$ ratio in basaltic melts. In reference to numerical experiments (half-spreading rate = 3 cm/yr; Fig. 2), the extraction/formation $P$-$T$ conditions of newly formed basaltic melts were analyzed at ~6.5 Ma model time of mid-ocean ridge activity.

The results demonstrate that as the $Tp$ increases, the $P$-$T$ conditions for basaltic melt extraction also increase (Fig. 2). For modern $Tp$ conditions, the average extraction $P$-$T$ conditions are 1.1 GPa and 1344 °C (Fig. 2F, G; Fig. S1A), consistent with petrological records[18]. In contrast, in Archean simulations, a $Tp$ of 1500 °C, which represents the previously estimated warm Archean mantle $Tp$[20] and the lower bound of the hot Archean $Tp$ range (1500–1600 °C)[19], indicates extraction conditions of 1475 °C and 2.1 GPa. Further increases in $Tp$ to 1550°C and 1600 °C result in extraction $P$-$T$ conditions of 1491 °C and 2.3 GPa, and 1528°C and 2.6 GPa, respectively (Fig. 2G–I; Fig. S1B). This shows a positive feedback effect between $Tp$ and the $P$-$T$ conditions of melt extraction, consistent with previous suggestions[11,15]. The relatively lower melting extraction temperatures observed beneath mid-ocean ridges compared to $Tp$ are primarily attributed to cooling effects induced by melting processes. This phenomenon generates a temperature-depth gradient that significantly exceeds the conventional value (0.5 °C/km) in the partially melted regions. In Archean models, the presence of distinct cold drips and the subsequent mixing between the colder lithospheric mantle and asthenosphere enhances this cooling effect (Fig. 2; see also Fig. S6).

A key strength of our numerical model is its incorporation of redox thermodynamic iterative calculations within an open system (Methods). This approach effectively integrates the influence of geophysical parameters, the thermodynamic equilibrium among complex solution models, and the findings from experimental petrology. This allows for the calculation of the whole-rock $Fe^{3+}/\Sigma Fe$ of basaltic melts based on their extraction $P$-$T$ conditions and the given mantle composition (Fig. 2). In the reference experiment, two initial whole-rock $Fe^{3+}/\Sigma Fe$ values were imposed for the mantle to assess their impact on the redox state of the derived basaltic melts. The first value = 0.04 falls within the reported modern mantle range (0.036–0.053)[10,25,26], corresponding to source forming MORB with $fO_2$ near QFM at the related $P$-$T$ conditions. The second value = 0.02 reflects a relatively reduced mantle state[10].

The results indicate that at the modern $Tp$, the mantle composition with an initial $Fe^{3+}/\Sigma Fe$ of 0.04 can produce basaltic melts with $Fe^{3+}/\Sigma Fe$ ranging from 0.10 to 0.11 (Fig. 2F, G, K, L), aligning well with observed values of 0.10–0.14[27–30]. However, the reduced mantle composition with an initial $Fe^{3+}/\Sigma Fe$ of 0.02 produces reduced basaltic melts with $Fe^{3+}/\Sigma Fe$ ratios of 0.05–0.06 (Fig. 2F, G, K, L). At the Archean $Tp$, the basaltic melt exhibits a more reduced redox state. The mantle composition with an initial $Fe^{3+}/\Sigma Fe$ of 0.04 yields basaltic melts with $Fe^{3+}/\Sigma Fe$ ranging from 0.07 to 0.08 (Fig. 2H–J, M–O), while the reduced mantle (initial $Fe^{3+}/\Sigma Fe$=0.02) produces basaltic melts with a low $Fe^{3+}/\Sigma Fe$ of 0.04 (Fig. 2H–J, M–O). This low $Fe^{3+}/\Sigma Fe$ of basaltic melts is roughly similar to that reconstructed for Archean eclogite xenoliths, which have been interpreted as metamorphosed oceanic crust derived from mid-ocean ridges[7]. Compared to modern results, these findings indicate that an increased $Tp$ during the Archean promoted the reduction of basaltic melts. This is because the temperature of melt extraction is higher under the hotter $Tp$ condition (Figs. 1 and 2; Supplementary Data 2).

### The double $Fe^{3+}/\Sigma Fe$ of modern mantle vs. Archean
Numerical forward modeling indicates that a higher $Tp$ can indeed result in a lower $Fe^{3+}/\Sigma Fe$ of basaltic melts within a certain range, which

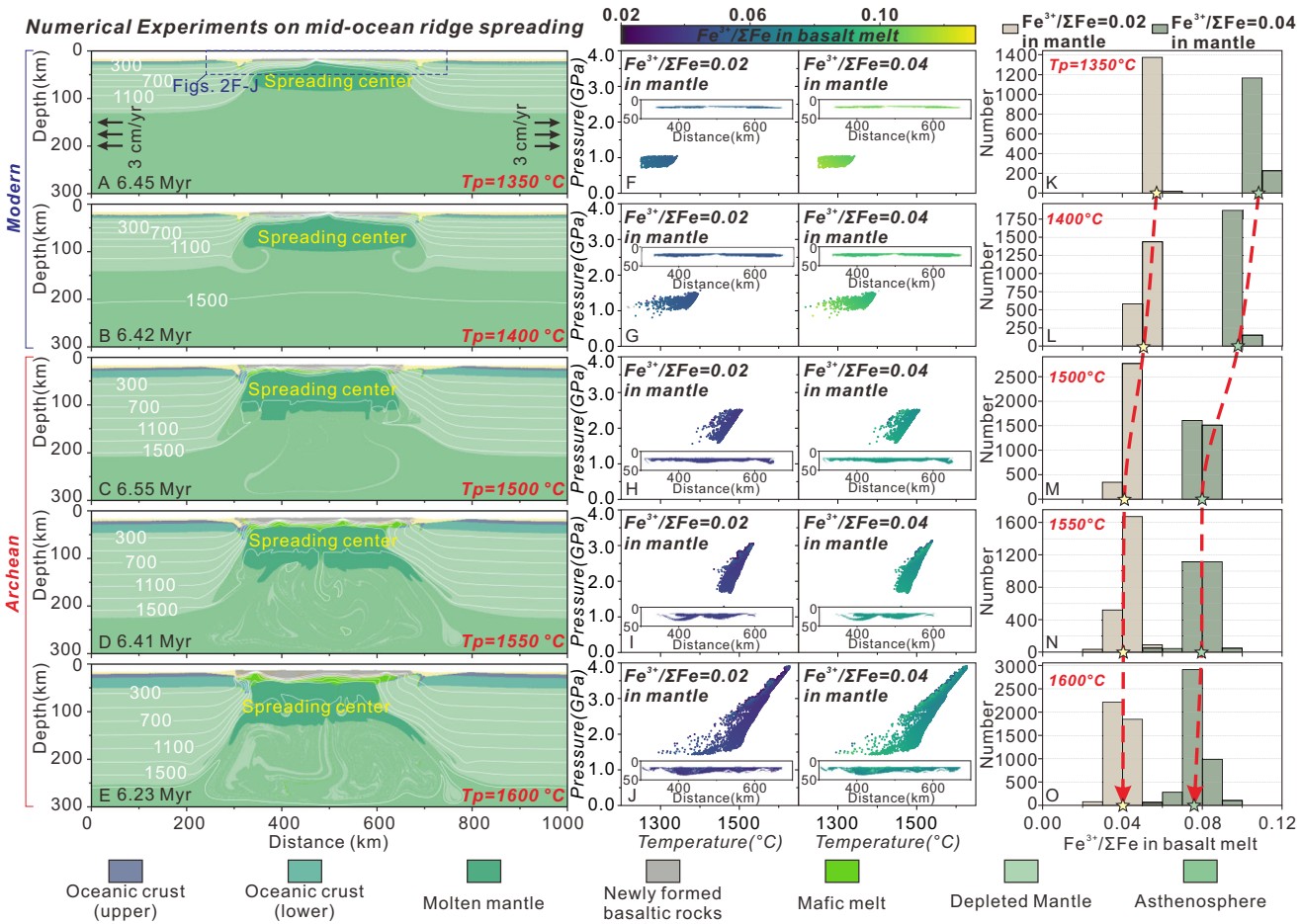

**Fig. 2 | Reference numerical experiments of mid-ocean ridge spreading under possible Archean and modern *Tp* conditions. A**–**E** Rock types and temperature fields in the experiments after ~6.5 Ma. **F**–**J** Spatial information of newly formed basalt and its in-situ temperature, pressure, and whole-rock $Fe^{3+}/\Sigma Fe$ during melt extraction marked, with initial mantle whole-rock $Fe^{3+}/\Sigma Fe$ = 0.04 and 0.02[10,25,26], respectively; These pixels represent all the basaltic melt data accumulated over time after reaching the melt extraction threshold. **K**–**O** Bar frequency diagrams of the whole-rock $Fe^{3+}/\Sigma Fe$ evolution of basaltic melts under varying *Tp* and initial mantle whole-rock $Fe^{3+}/\Sigma Fe$; the pentagram represents the average value under each *Tp*-$Fe^{3+}/\Sigma Fe$ condition; the values in the bar charts are composed of all the data points shown in (**F**–**J**).

is primarily controlled by high temperature (Fig. 1A), even with the same whole-rock $Fe^{3+}/\Sigma Fe$ of the mantle sources. However, does this phenomenon fully explain the differences in oxygen fugacity observed between Precambrian (particularly Archean) and modern basalts? We performed thermodynamic back-calculations on the compiled basalt samples (Methods, Fig. 3A; Supplementary Data 3). We first determined the $P$-$T$-$fO_2$ conditions during the melting extraction/formation of MORB-like basaltic rocks using V-based oxybarometers (Fig. 3B; Fig. S2A). This enabled us to apply our thermodynamic methods (alongside empirical methods)[27,28] to back-calculate the whole-rock $Fe^{3+}/\Sigma Fe$ of basaltic melts (Fig. 3C; Fig. S2B, C). Subsequently, thermodynamic modeling could estimate the oxygen fugacity and whole-rock $Fe^{3+}/\Sigma Fe$ in the mantle sources (Fig. 3D, E; Fig. S2D–F).

For the Archean samples, our results indicate that the average V-Ti (titanium) oxygen fugacity[13,18] recorded by basalt samples is −1.5 ± 0.2 (Fig. 3B; Supplementary Data 3), under average $P$-$T$ conditions of 2.2 GPa/1500 °C that align with results obtained from numerical simulations at mantle potential temperatures of 1500–1600 °C, with an average of 2.4 GPa/1502 °C (Figs. S1, S2). The V-Sc oxygen fugacity[13,18], based on a smaller number of samples than the V–Ti dataset, is reported as QFM-1.1 ± 0.3 (Fig. S4; V-Ti oxygen fugacity of the same samples is QFM-1.4 ± 0.2). Despite variations in filtering methods and data sources, the calculated primitive oxygen fugacity aligns with previous conclusions on basalts[6,7,11,13], with values of QFM-1.2 at 1.9 GPa/

1455 °C and QFM-1.3 at 2 GPa/1469 °C, both within error margins. This consistency reinforces the conclusion that the oxygen fugacity of Archean basaltic melts is relatively low.

The whole-rock $Fe^{3+}/\Sigma Fe$ of basaltic rocks at primitive $P$-$T$-$fO_2$ conditions of melting extraction was calculated using thermodynamic simulations, yielding an average value of 0.04 ± 0.01 (Fig. 3C; Fig. S2–S4), consistent with empirical methods (Fig. S2; 0.04). Additionally, our thermodynamic calculations for two previous basalt databases yielded a slightly higher whole-rock $Fe^{3+}/\Sigma Fe$ of 0.05 ± 0.01[11,13] (Fig. S3; Supplementary Data 3). Thermodynamic back-calculation of all databases provided consistent results, showing that the Archean mantle had an average whole-rock $Fe^{3+}/\Sigma Fe$ ratio of 0.02 (Fig. 3E; Fig. S3).

The back-calculation results closely match those of numerical simulations (Figs. 2–3; Supplementary Data 2). Specifically, if the Archean mantle had the same $Fe^{3+}/\Sigma Fe$ ratio as the modern mantle, it would produce a basaltic melt with a $Fe^{3+}/\Sigma Fe$ ratio of 0.07–0.08 with $Tp$ = 1500–1600 °C. However, actual records indicate an average $Fe^{3+}/\Sigma Fe$ of 0.04–0.05, aligning with simulation results corresponding to melts with a mantle $Fe^{3+}/\Sigma Fe$ ratio of 0.02[7]. This indicates that the redox state of the Archean mantle as proxied by its $Fe^{3+}/\Sigma Fe$ was significantly lower than the modern mantle (0.04–0.05).

We further evaluated the potential impact of valence state changes in carbon and sulfur on the mantle redox budget. In the modern

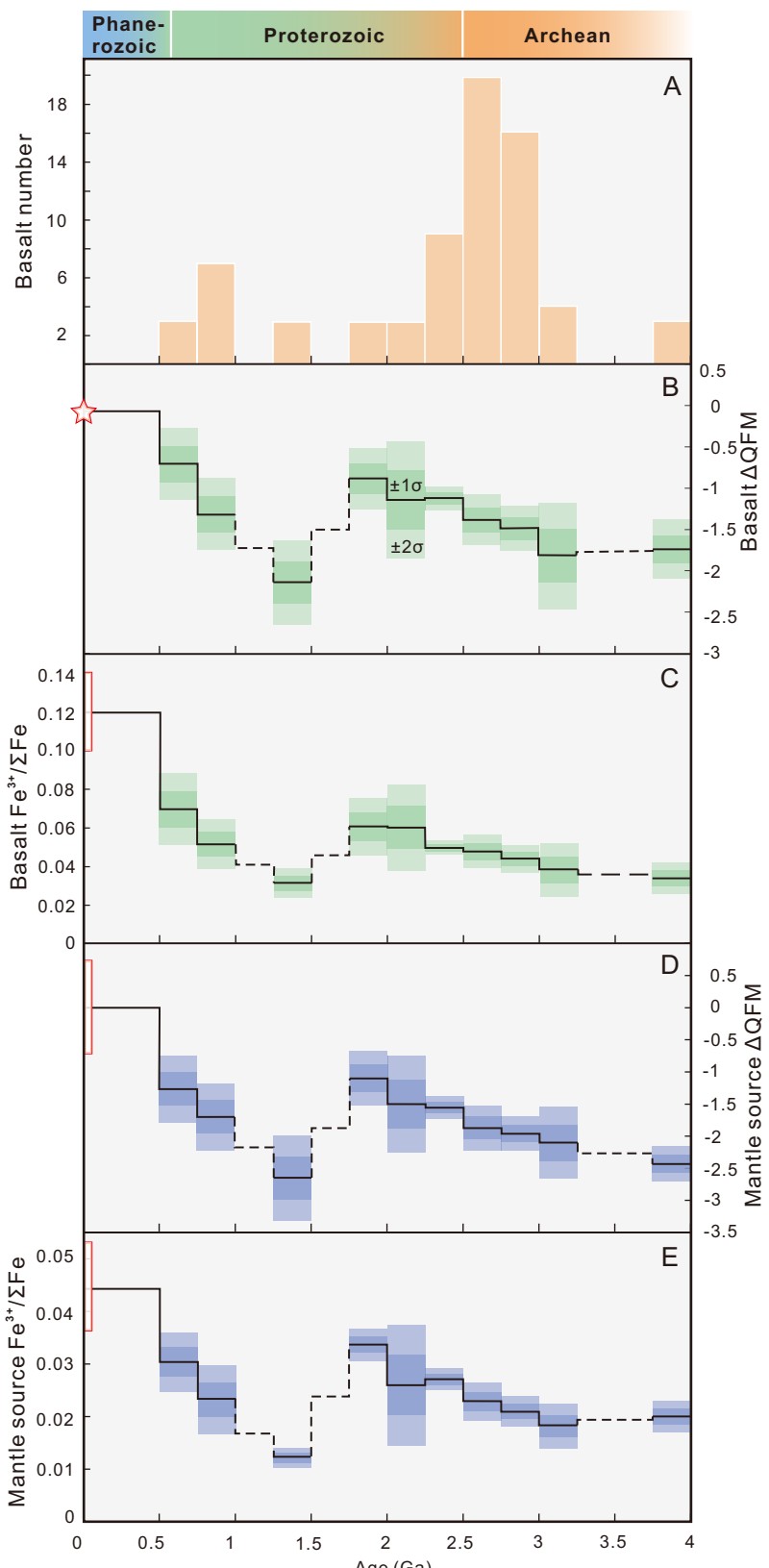

**Fig. 3 | The redox evolution of collected Precambrian MORB-like basalt samples since the Archean. A** Age distribution of basalt samples. **B** Source oxygen fugacity of basalts. **C** Whole-rock $Fe^{3+}/\Sigma Fe$ when the basaltic melts formation from thermodynamic calculations (see Methods). **D** Mantle source oxygen fugacity from thermodynamic calculations based on $P$-$T$-$Fe^{3+}/\Sigma Fe$ of basaltic melts (see Methods). **E** Whole-rock $Fe^{3+}/\Sigma Fe$ of the mantle source from thermodynamic calculations based on $P$-$T$-$Fe^{3+}/\Sigma Fe$ of basaltic melts (see Methods). Modern MORB (star and red rectangle in **B**, **C**) and mantle (red rectangles in **D**, **E**) values are from references cited in the main text.

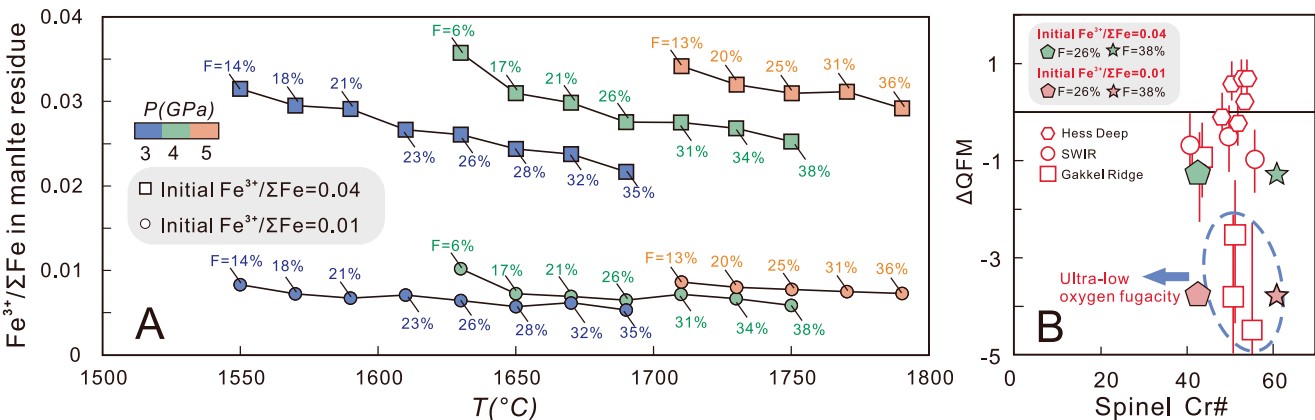

**Fig. 4 | Thermodynamic experiments on isobaric batch melting. A** Whole-rock $Fe^{3+}/\Sigma Fe$ of the residues after different melt fractionation from depleted mantles (initial $Fe^{3+}/\Sigma Fe$ = 0.04 and 0.01, respectively) at 3–5 GPa. **B** Comparison of the oxygen fugacity and spinel Cr# in the residues (F = 26% and 38% at 4 GPa) with the natural samples of refractory mantle residues; only when the initial whole-rock $Fe^{3+}/\Sigma Fe$ ratio is very low (e.g., 0.01) can it fall into the low oxygen fugacity range. The data for natural samples from Hess Deep, Gakkel Ridge and Southwest Indian Ridge (SWIR) was collected by ref. 15.

mantle, carbon in MORB source is typically assumed to exist as $C^{4+}$, serving as a reference state[22]. Thermodynamic calculations suggest that $C^{4+}$ can remain stable down to QFM-1.5 to −2 buffer at 2–3 GPa[23,24], corresponding to the oxygen fugacity of the Archean mantle. Furthermore, given the low abundance of C in typical MORB mantle sources[22], its contribution to the redox budget is limited, even if valence transitions occur. Sulfur in the modern MORB mantle already exists in its reduced form as $S^{2−}$, and it is unlikely to have been in a more reduced state in the Archean[22]. Thus, sulfur valence states are negligible in terms of their effect on the redox budget of the MORB mantle.

In summary, our findings suggest that the mantle $Fe^{3+}/\Sigma Fe$, as a proxy for redox state and budget, has doubled. A deeper onset of melting owing to higher Archean mantle $Tp$ alone is insufficient to explain the differences in oxygen fugacity recorded in rocks. This stands in stark contrast to recent reports[11,15] but is consistent with the conclusions of some previous studies[6,7,13,14].

### Ultra-reduction unattainable through deep and hot melting
Some highly reduced mantle domains (down to QFM-4 to -5, and spinel $Fe^{3+}/\Sigma Fe$ ratios can be lower than 0.03) identified in present-day oceanic lithosphere are thought to represent residual material after partial melting of the Archean mantle[15]. These reduced mantle domains were interpreted as residues that had the same redox state as the modern mantle but underwent extensive partial melting (>20%) at high $P$-$T$ conditions that could generate komatiites[15].

A key implication of our study is that it does not support this inference. As shown in Fig. 4, we modeled the melt extraction from a modern oxidized mantle ($Fe^{3+}/\Sigma Fe$ = 0.04) along an isobaric trajectory at 3–5 GPa until the melt fraction (F) exceeded 35%, then we calculated the $Fe^{3+}/\Sigma Fe$ and other redox parameters (e.g., $fO_2$ and $Fe^{3+}$ in minerals) of the residual material (Fig. 4A, B). After undergoing very high degrees of partial melting, the $Fe^{3+}/\Sigma Fe$ in the residues decreases, but the degree is insufficient to match the observed values (Fig. 4A, B). For example, at 4 GPa and 1750 °C, the mantle experiences high melting degrees (F = 38%). The residual mantle is harzburgite with a very high Cr# in spinel, consistent with observations[15] (Fig. S5). However, the in-situ residual mantle oxygen fugacity is QFM-1.3 at this stage, and the $Fe^{3+}/\Sigma Fe$ in spinel exceeds 0.2, both of which are much higher than the observed values[15] (Fig. S5).

Therefore, the observed ultra-reduced mantle residues cannot be explained by high melting degrees at deeper depth during Archean. Such residue formation can only be attributed to a mantle source that initially had a very low $Fe^{3+}/\Sigma Fe$ (even <0.01) before melting (Fig. 4B;

Fig. S5B, D). This further suggests that Archean mantle was more reduced than the modern mantle.

### The tortuous evolution of Mantle $Fe^{3+}/\Sigma Fe$ ratio
Our study, in addition to comparing the redox states of the Archean and modern mantle, reveals the evolutionary trajectory of mantle $Fe^{3+}/\Sigma Fe$ since the Hadean (Figs. 3 and 5), particularly for MORB-like mantle source regions, which is tortuous, not a monotonic process and likely reflects several major events in Earth's geological history.

Our discussion is based on the trends reflected in the current database. However, it should be noted that, despite the application of stringent screening criteria (e.g., $(Nb/La)_{PM} \geq 1$) to ensure comparability with modern MORB-like environments, the precise provenance and tectonic setting of the basalt samples cannot be fully constrained. Moreover, preservation bias potentially induced by the supercontinent cycle and variations in tectonic regime strength may have influenced the spatiotemporal distribution of basaltic records (Fig. 3). In combination with limited statistical coverage in certain geological periods, these factors warrant caution when interpreting the compiled data. Future discoveries and reporting of additional rock records may help mitigate the underrepresentation of specific time intervals, such as the Mesoproterozoic and the Eoarchean.

Specifically, Hadean mantle-derived zircons suggest that the redox state of ~4.4 to 4.0 Ga upper mantle may have been close to or even exceeded modern levels[31] (Fig. 5A, C). By the time a melt becomes saturated in zircon, its composition has typically been significantly modified or evolved from the original mantle-derived melt. Therefore, whether $fO_2$ recorded by such melts truly reflects mantle conditions remains questionable. Nevertheless, we still performed $Fe^{3+}/\Sigma Fe$ estimations on the relevant samples for comparison with previously established redox trends. Our modeling suggests that these mantle source regions may have contained an average $Fe^{3+}/\Sigma Fe$ of 0.07 (Methods). Some studies proposed that this Hadean-mantle redox state resulted from the redox disproportionation of $Fe^{2+}$ into $Fe^{3+}$ and $Fe^0$ within a magma ocean stage, a process that may have extended to the top of lower mantle[32,33]. The high-density metallic iron would have sunk into the lower mantle or core[2,33,34], leaving the upper mantle magma ocean increasingly oxidized (Fig. 5A).

Hadean zircon analyses further suggest that the mantle became progressively more reduced during the late Hadean[31,32]. This is consistent with findings from our basalt database and other studies[6,11,13], which indicate that the Archean mantle exhibited highly reduced conditions (Fig. 5C). A plausible explanation for this reduction lies in a late accretion event before early Archean, during which chondritic

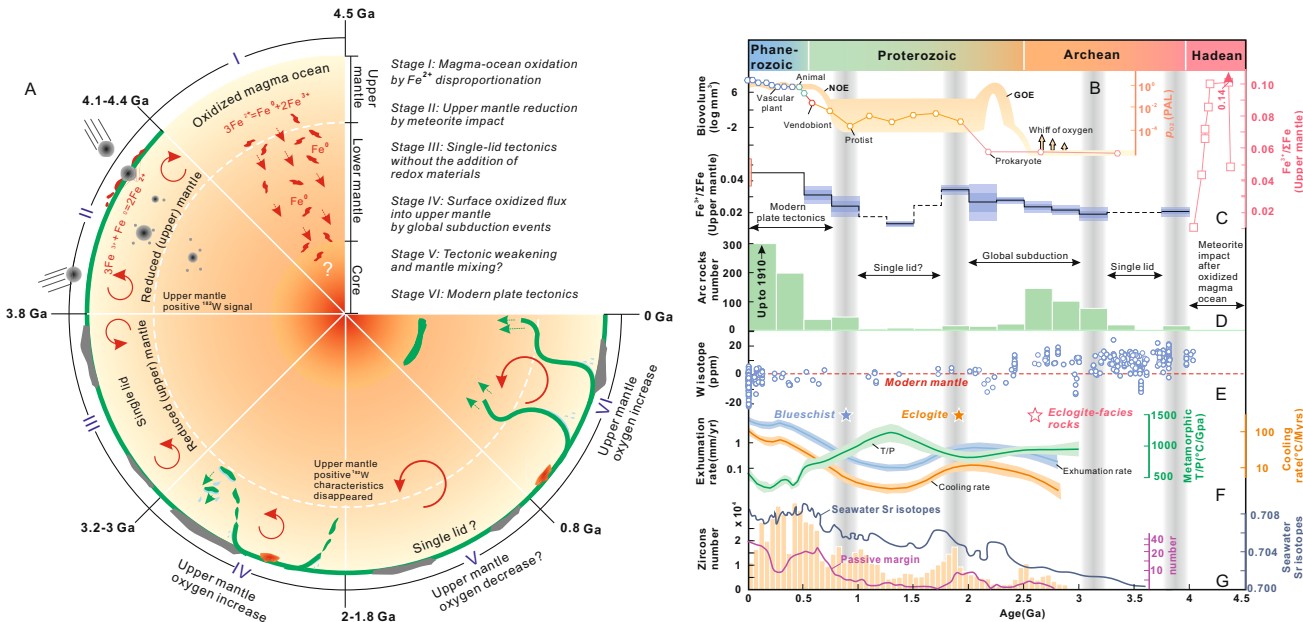

**Fig. 5 | The tortuous evolution of the mantle redox state during Earth history and its correlation with major geological events. A** A cartoon diagram showing the coupled evolution of mantle redox state and major geological events. **B** The trends in atmospheric and biological evolution[70,71]. **C** The record of mantle whole-rock $Fe^{3+}/\Sigma Fe$ since the Hadean (this study). **D** Age distribution diagram of arc

(basaltic) magma samples since the Archean[66,67]. **E** Trends in W isotopes since the Archean[35–38]. **F** Metamorphic *T/P* ratio, cooling rate, exhumation rate, and the first occurrence of characteristic metamorphic rocks (marked with stars)[49,50,70,71]. **G** Trends in zircon distribution, seawater Sr isotopes, and the number of passive margins[50,70,71].

meteorites comprising 0.5–1% of Earth's mass delivered metallic iron to the mantle (Fig. 5A), triggering a reduction event[32,33]. This hypothesis suggests that the accretion of meteorites with negative tungsten (W) isotopic signatures (–190 ppm) lowered the W isotopic composition of the silicate mantle, which initially had a positive W isotope composition (e.g., 20–30 ppm), in the wake of core formation and evolution at high Hf/W[35–38] (Fig. 5E). Mass balance calculations indicate that this impact event at the Archean–Hadean boundary could have significantly altered the mantle's redox state into observational results (Fig. 5), even if the magma ocean initially had a high $Fe^{3+}/\Sigma Fe$ ratio (>0.2)[32]. If this hypothesis is correct, the surface oxidizing environment was interrupted for over a billion years due to this meteorite impact event, hindering the early development of Earth's habitability (Fig. 5B, C).

Following this event, the mantle redox state remained relatively reduced during the early Archean[31], with sparse samples suggesting the persistence of single-lid tectonics (Figs. 3 and 5). It was not until 3.0–3.2 billion years ago that the mantle began to oxidize progressively (Fig. 5A, C). One hypothesis suggests that this oxidation is contributed from an $Fe^{3+}$-rich-bridgmanite-bearing lower mantle layer through mantle plume activity[9]. From the perspective of mass balance and charge balance, this has been proven to be a reasonable assumption. However, the hypothesis still faces the following challenges:

The lower mantle is not necessarily more oxidized than the upper mantle. Recent thermodynamic simulations suggest that metallic $Fe^0$, produced during the disproportionation reaction associated with the formation of $Fe^{3+}$-rich bridgmanite, may have accumulated at the top of the lower mantle rather than sinking into the core[34]. This implies that the overall redox budget may not have changed.

Although recent studies suggest that a basal magma ocean is geodynamically plausible[39], the melt fraction at the core–mantle boundary at that time would have been close to 100%, making it difficult for $Fe^{3+}$-rich bridgmanite to crystallize and thus for the disproportionation reaction to proceed[39]. Therefore, from both thermodynamic and geodynamic perspectives, a new mechanism for

generating $Fe^{3+}$-rich bridgmanite may be required to further support the oxidized plume hypothesis.

Finally, considering the similarities to Earth, $Fe^{3+}$-rich bridgmanite should also be present on Venus, and the planet exhibits vigorous mantle plume activity. However, it does not show any evidence for a mantle oxidation mechanism[40]. Nonetheless, the associated hypothesis presents an appealing explanation. Future progress in experimental petrology and planetary science may provide critical insights to test and potentially validate this model.

Another plausible mechanism is the onset of global mobile-lid tectonics from 3.0–3.2 billion years ago and continued into the Archean–Proterozoic boundary[41–44], which initiated large-scale mantle oxidation (Fig. 5D–G). Multiple geological evidence supports this tectonic transition, including the increasing abundance of zircons, which suggests the growth of felsic rocks; changes in seawater strontium (Sr) isotopes and the proliferation of passive margins; and the emergence of bimodal metamorphism and eclogite-facies rocks, characteristic of subduction-driven tectonics[41–43] (Fig. 5D–G). Recent data also indicate large-scale craton movement associated with mobile-lid tectonics at least at ~2.7 Ga[44].

Our compiled database of arc rocks further highlights subduction processes and the associated development of arc magmatism since the late Archean (Fig. 5D; Supplementary Data 3). Subduction zones played a critical role in enhancing mantle oxidation by transporting accumulated surface redox budget into the mantle, which was generated through hydration reactions (e.g., formation of serpentinite and altered oceanic crust)[45], thereby increasing mantle redox state.

Before the onset of global subduction (e.g., the early Archean), the oceanic lithosphere accumulated redox budget through oceanic hydration reactions and released reduced gases such as hydrogen. During the Archean, the escape rate of hydrogen was particularly high[46], leaving behind a net redox budget in the oceanic lithosphere. In addition, these reduced gases likely acted as sinks for oxygen by reacting with $O_2$ produced by photosynthesis after entering the atmosphere. Notably, prior to subduction, the redox budget stored in

the oceanic lithosphere did not contribute oxygen to the atmosphere due to isolation by seawater.

With global subduction initiation at the Archean–Proterozoic boundary, a substantial portion of the redox budget was transferred into the mantle. This marked the emergence of new oxygen sources driven by subduction. On one hand, subduction generates oxidized arc magmas and their associated oxidized degassing. On the other hand, mantle oxidation induced by subduction can enhance the redox state of other tectonic degassing processes (e.g., mid-ocean ridges). These processes represent new mechanisms for transferring redox budget from Earth's interior to the atmosphere.

Geological evidence indicates that most Archean komatiites exhibit elevated high $Fe^{3+}$ contents as a result of serpentinization[45], and the earliest records of serpentinization date back to the early Archean[47]. In addition, banded iron formations (BIFs), which are unique to the Precambrian, also exhibit very high ferric iron $Fe^{3+}$ ratios, all of which indicate the early accumulation of oxidized materials in the lithosphere. Our latest quantitative results (D.W.Y., G.T., & Z.X., in preparation) reveal that the global subduction system transfers a net redox budget of $36 \pm 6 \times 10^{12}$ mol/yr to the mantle. This estimate is conservative, as it does not account for the additional redox budget contributed by serpentinized komatiites, BIFs, and the altered oceanic crust unique to modern settings (e.g., $Fe^{3+}/\Sigma Fe > 0.5$), but only considers the contributions from previously observed Archean altered oceanic crust with relatively low $Fe^{3+}/\Sigma Fe < 0.26$. Mass-balance calculation suggests that this redox budget is sufficient to oxidize the whole mantle to the modern levels (D.W.Y., G.T., & Z.X., in preparation).

Numerical simulations reveal that Archean-Paleoproterozoic subduction differed markedly from modern ones[48], as higher $Tp$ caused frequent slab break-off (Fig. 5A and Fig. S6). Due to the lack of negative buoyancy, some smaller slabs were retained in the mantle transition zone, where dense oceanic crust separated from lighter lithosphere[48]. Most subducted lithosphere returned to the upper mantle, driven by positive buoyancy, and recycling within it[48] (Fig. 5A). This ancient cycle style can accelerate the upper (or shallow) mantle oxidation due to the smaller mass of mantle.

Some studies suggested that the mantle redox state during the Paleoproterozoic approached modern levels[6,7,14]. Our and other previous data do reveal the presence of such oxidized mantle compositions[11], but also record relatively reduced compositions, likely reflecting mantle heterogeneity[7]. Thus, while our findings indicate further mantle oxidation during the Paleoproterozoic, the average redox state had not yet reached modern levels (Fig. 5C).

Whether plume-driven or subduction-driven mechanisms, the result is that mantle oxygen fugacity increased by more than 0.5 log units relative to the QFM buffer between 3 and 2.3 Ga (Fig. 3D), calculations of which suggest could still trigger GOE[4]. Limited Mesoproterozoic data from our and other studies[6,7,11,14] may indicate a stagnation or even a decrease in mantle oxidation during this period. Geological evidence[49,50], such as the decreasing convergent margins, an increase in metamorphic $T/P$ ratios, and the decreasing metamorphic cooling and exhumation rates (Fig. 5), suggests a weakening or stagnation of mobile-lid tectonics. Subduction could no longer transfer redox budgets into mantle. However, due to limited data, we remain cautious about the trend during Mesoproterozoic period.

Mantle oxidation resumed from the Neoproterozoic to now[13], overlapping with modern plate tectonics characterized by low-temperature metamorphism (e.g., blueschist; Fig. 5)[43,50]. During this period, accumulated surface redox budgets from the Mesoproterozoic could be rapidly transferred into the mantle via subduction (D.W.Y., G.T., & Z.X., in preparation). The simultaneous occurrence of Neoproterozoic oxidation events (NOE) suggests that these were likely not coincidental but driven by mantle oxidation processes (Fig. 5B, C), then further facilitated the emergence of complex life. ~4-billion-years accumulation of surface redox budgets[47], over 2-billion-years global

episodic subduction activities (Fig. 5) and mantle convection resulted in relatively uniform mantle redox conditions, leading to the more consistent redox states recorded in modern MORB[27–30], compared to the early Earth (Figs. 3 and 5).

Our analysis highlights that shifts in the mantle redox state closely track major tectono-magmatic events, serving as key indicators of Earth's dynamic evolution (Fig. 5). These redox changes, while primarily reflecting large-scale geodynamic reorganizations, also influence surface conditions by linking deep Earth processes to atmospheric oxygenation and planetary habitability. This underscores the dual role of mantle redox evolution, which both records and modulates the coupled geodynamic, geochemical, and biological systems that sustain Earth over geological time.

## Methods

### Basalt reflects the mantle redox state

During the early Archean, $Tp$ was 150–300 °C higher than that at present[19,20]. The preserved rocks from that period are exceedingly rare. These elevated temperatures have profound implications for the tectonic regimes, directly informing the selection of tectonic scenarios and rock types for subsequent redox simulations. Numerical simulation developed by ref. 51. can effectively reconstruct the tectonic regimes of that time. Here, we show a case of this model for the early Archean under the $Tp$ condition of 1600 °C (Fig. S6).

The early-Archean simulation indicates that the thick oceanic crust (~20–25 km), composed of cold and dense material, experienced subsidence due to negative buoyancy. This process triggered the formation of several mantle upwelling zones, where ascending, hotter, melt-bearing peridotite underwent additional decompression melting. This led to the generation of basaltic or picritic magma, which subsequently formed new hydrated basaltic lavas at the crust surface. The mantle became progressively depleted due to melt extraction.

A series of spreading centers formed, producing basaltic melts that generated a new basaltic crust (Fig. S6). Over time, this process facilitated the formation of oceanic slabs. A large spreading center emerged, with weak zones developing along its margins, eventually giving rise to Archean subduction events characterized by frequent slab break-off[48].

Notably, basaltic melts at the spreading centers remained the primary product of mantle melting and the dominant component of oceanic crust, consistent with geological observations, despite the high $Tp$ favored the frequent komatiite formation. In comparison, the $P$-$T$ conditions for basalt formation are relatively constrained[11], leading to smaller errors for redox-state calculation. Thus, MORB-like basalts are the more suitable candidates for tracking mantle redox states since the Archean, based on the simulation and observation. Under modern geodynamic regimes, if the upper and lower mantle can mix effectively and their composition becomes relatively homogeneous, basalts can represent the average state of the mantle.

It seems difficult to determine the redox properties of the lower mantle in the Archean. However, a reasonable hypothesis is that if magmatic redox states could only persist to the top of the lower mantle[32], and meteorite impacts also affected the lower mantle[35], then theoretically, the lower mantle in the Archean would have been reduced. The oxygen stagnation during the Mesoproterozoic might also indicate that more reducing materials from the lower mantle were brought into convection during mantle mixing (Fig. 5).

### Thermomechanical modeling

In this study, we employed the two-dimensional (2D) thermomechanical simulation code I2VIS[52] to solve the governing equations for momentum, mass continuity, and heat conservation. The numerical approach is based on the marker-in-cell method combined with a finite difference scheme, ensuring accurate spatial and temporal discretization[52]. The fundamental equations governing the system are

expressed as follows:

$$\frac{\partial \sigma'_{ij}}{\partial x_j} - \frac{\partial P}{\partial x_j} + \rho \mathbf{g}_i = 0 \tag{1}$$

$$\mathrm{div}(\vec{\mathbf{v}}) = \frac{\partial v_x}{\partial x} + \frac{\partial v_y}{\partial y} = 0 \tag{2}$$

$$\rho C_P \left(\frac{DT}{Dt}\right) = \frac{\partial}{\partial x_j}\left(\kappa \frac{\partial T}{\partial x_j}\right) + H_a + H_r + H_s, \tag{3}$$

where $\partial \sigma'_{ij}$ represents the deviatoric stress tensor, $P$ denotes the pressure, $\rho$ is the density, and $\mathbf{g}_i$ corresponds to the gravitational acceleration. The velocity components in the horizontal and vertical directions are $v_x$ and $v_y$, respectively. The parameter $C_P$ stands for the isobaric heat capacity, while $\frac{DT}{Dt}$ refers to the material derivative of temperature $T$. The coefficient $\kappa$ characterizes the thermal conductivity, and the terms $H_a$, $H_r$, and $H_s$ represent the contributions from adiabatic, radiogenic, and shear heat production, respectively. The latent cooling and heating effects related to respectively melting and crystallization of rocks is taken into account implicitly by correcting $C_P$ and $H_a$ values in the temperature equation[53].

**Rheology.** We adopt a visco-plastic rheology to describe rock deformation. The viscous creep of rocks is formulated based on deformation invariants and is influenced by temperature, pressure, and strain rate. Rocks exhibit both plastic and viscous behavior, transitioning into a slowly creeping fluid-like state over long timescales. The plastic rheology follows the Drucker–Prager yield criterion[54], which defines the yield stress as:

$$\sigma_{\mathrm{yield}} = C + P \sin(\varphi), \tag{4}$$

where $C$ represents compressive strength (cohesion), $P$ denotes pressure, and $\varphi$ is the internal friction angle. For dry rocks, the friction angle is given by $\varphi_{\mathrm{dry}}$, and the influence of pore fluid pressure is accounted for by the pore fluid pressure factor $\lambda$, modifying the frictional term as:

$$P \sin(\varphi) = P \sin\left(\varphi_{\mathrm{dry}}\right)(1 - \lambda). \tag{5}$$

The plastic viscosity is determined using the relationship:

$$\eta_{\mathrm{plastic}} = \frac{\sigma_{\mathrm{yield}}}{2\dot{\varepsilon}_{\mathrm{II}}}, \tag{6}$$

where $\dot{\varepsilon}_{\mathrm{II}}$ represents the second invariant of the strain rate tensor. For the viscous rheology, the effective ductile viscosity follows a power-law dependence on stress:

$$\eta_{\mathrm{ductile}} = A_D^{-\frac{1}{n}} \sigma^{\frac{1-n}{n}} e^{\frac{E_a + PV_a}{nRT}}, \tag{7}$$

where: $E_a$ is the activation energy, $V_a$ is the activation volume, $n$ is the stress exponent, $R$ is the universal gas constant, $A_D$ is a material constant. Finally, the effective viscosity is determined as the lower bound of plastic and ductile viscosities:

$$\eta_{\mathrm{eff}} = \min\left(\eta_{\mathrm{plastic}}, \eta_{\mathrm{ductile}}\right). \tag{8}$$

**Melt.** the melt fraction is assumed to follow a linear dependence on temperature. For a given rock type and pressure condition, the volumetric melt fraction $M$ is determined using the following formulation:

$$M = \begin{cases} 0, & T \leq T_{\mathrm{solidus}} \\ \frac{T - T_{\mathrm{solidus}}}{T_{\mathrm{liquidus}} - T_{\mathrm{solidus}}}, & T_{\mathrm{solidus}} < T < T_{\mathrm{liquidus}} \\ 1, & T \geq T_{\mathrm{liquidus}} \end{cases} \tag{9}$$

where $T_{\mathrm{solidus}}$ and $T_{\mathrm{liquidus}}$ represent the solidus and liquidus temperatures for the specific lithology (see Supplementary Data 4). The effective density of partially molten rocks, $\rho_{\mathrm{eff}}$, varies as a function of the melt fraction and pressure-temperature (P-T) conditions:

$$\rho_{\mathrm{eff}} = \rho_{\mathrm{solid}} - M(\rho_{\mathrm{solid}} - \rho_{\mathrm{molten}}) \tag{10}$$

where $\rho_{\mathrm{solid}}$ and $\rho_{\mathrm{molten}}$ denote the densities of the solid and molten phases, respectively. Additionally, the dependence of density on pressure and temperature is described as:

$$\rho_{P,T} = \rho_0 [1 - \alpha(T - T_0)][1 + \beta(P - P_0)], \tag{11}$$

where $\rho_0$ is the reference density at standard conditions ($P_0 = 0.1$ MPa and $T_0 = 298$ K), $\alpha$ is the thermal expansion coefficient, $\beta$ is the compressibility coefficient. Once partial melting initiates ($M > 0$), the effective heat capacity ($C_{\mathrm{pe}}$) and thermal expansion coefficient ($\alpha_e$) are modified accordingly:

$$C_{\mathrm{pe}} = C_P + Q_L \left(\frac{\partial M}{\partial T}\right)_{P = \mathrm{const}} \tag{12}$$

$$\alpha_e = \alpha + \frac{\rho Q_L \left(\frac{\partial M}{\partial P}\right)_{T = \mathrm{const}}}{T}, \tag{13}$$

where $Q_L$ represents the latent heat of fusion.

The thermomechanical models simulate mid-ocean ridge extension experiments under different mantle potential temperatures, including modern conditions (1350 °C and 1400 °C) and Archean conditions (1500 °C, 1550 °C, and 1600 °C)[19,20], based on an area of 1000 km × 300 km (Figures. S7). The rectangular grid is composed of 501 × 151 nodes with a uniform resolution of 2 km × 2 km.

The model includes oceanic crust composed of basalts and gabbros, with thickness varying depending on the mantle potential temperature. The lithospheric depleted mantle and asthenospheric mantle are composed of dry olivine. In the modern models, oceanic crust and lithosphere follow the half-space cooling model, where temperature evolution as a function of time and depth follows the classical heat conduction equation, simulating the cooling and thickening process over time. In the Archean, higher mantle potential temperatures result in a thicker lithospheric mantle, but it is unknown whether the oceanic lithosphere follow the half-space cooling model. Based on simulation results and observations of cratonic lithospheric mantle thickness (up to 250 km)[48,51], we thus assume that at $Tp = 1600$ °C, the lithospheric mantle reaches a maximum thickness of approximately 250 km. A 50 °C decrease in potential temperature corresponds to an approximate 20 km reduction in lithospheric mantle thickness. The thicknesses provided in the Supplementary Data 2. Detailed material physical properties can be found in Supplementary Data 4. In the reference model (Fig. 2), we present information after 6.5 Ma of mid-ocean ridge spreading, because mid-ocean ridge activity at this time has become mature under all $Tp$ conditions. In addition, we simulated different extension rates (half-spreading rates of 1 and 5 cm/year) and the impact of melt extraction at different times on the melt extraction P-T conditions and $Fe^{3+}/\Sigma Fe$. According to reports, we also tested different melt extraction thresholds (2–10%)[48,55]. The results are displayed in the Supplementary Data 2. The results indicate that the $Fe^{3+}/\Sigma Fe$ of

the melt extracted has very little effect from these parameters of melt extraction thresholds and extension rates (<0.01).

## Thermodynamic modeling

Thermodynamic modeling under the $Na_2O \cdot CaO \cdot FeO \cdot MgO \cdot Al_2O_3 \cdot SiO_2 \cdot H_2O \cdot O \cdot Cr_2O_3$ (NCFMASHOCr) system was carried out with the software GeoPS[56], using the ds633 thermodynamic database[57]. Based on previous studies and numerical simulation results (Fig. S6)[51,58], we use the reported dry depleted MORB mantle whole-rock composition[58] as the initial source of MOR-basaltic melt (Supplementary Data 1). We also tested the effect of the evolution from a primitive mantle end-member[58] to a depleted MORB mantle end-member on the $Fe^{3+}/\Sigma Fe$ ratio in basaltic melts (Supplementary Data 1). The results indicate that the evolution of major element compositions from primitive to depleted mantle has a negligible impact on the modeling results (Fig. S8). The thermodynamic solution models employed including olivine (Ol), spinel (Sp), garnet (Gt), orthopyroxene (Opx), clinopyroxene (Cpx), and redox-sensitive melts (Melt) are from refs. 59,60. These updated models are applicable under conditions up to 6–8 GPa and align well with experimental petrology, particularly in predicting redox states, a capability that has been extensively validated in recent studies[59–63]. We also validated our approach against experimental petrology data[64] before modeling. Specifically, previous studies have indicated that (Alpha-)Melt software tends to overestimate the oxygen fugacity (up to 1.2 orders of magnitude) of the residual system due to an unsuitable spinel model of the Melt software[63] (Fig. S9). By comparing experimental petrology results[64] for the relevant compositions with our thermodynamic outputs, we found that the predicted $Fe^{3+}$ behavior in spinel matches experimental observations, indicating no bias in the model predictions. We further modeled the $Fe^{3+}/\Sigma Fe$ -$fO_2$ relationship of the rock system, and that predicted by the thermodynamic model for mantle rocks under given $P$-$T$ conditions aligns well with experimental observations[26].

For the melt $P$-$T$ conditions derived from thermomechanical numerical simulations, thermodynamic modeling will be called to conduct iterative modeling of depleted mantle compositions under open-system conditions with two initial whole-rock $Fe^{3+}/\Sigma Fe$ values (0.04 and 0.02). 0.04 falls within the range of modern measured values, while 0.02 is used as a reduced end-member—essentially half of the modern value—for comparison experiments. The thermodynamic simulation adheres to the melt extraction thresholds. When the output $P$-$T$ conditions surpass this threshold, the residual system after extracting melt is used for iterative calculations, and this process is repeated cyclically. The thermodynamic calculations will finally output the melt $Fe^{3+}/\Sigma Fe$ corresponding to each data from the thermomechanical simulation.

To simulate the influence of melting degree on the redox evolution of residual refractory components at hot and deep conditions, we conducted isobaric melt fractionation simulations using an initial depleted mantle at 3–5 GPa. Results were output at 20 °C intervals until the cumulative melt fraction (F) exceeded 35%. The results of initial $Fe^{3+}/\Sigma Fe$ for the depleted mantle at 0.04 and 0.01 are shown, respectively. Because we conducted a series of calculations (ranging from 0.04 to 0.01), we found that only $Fe^{3+}/\Sigma Fe$ as low as 0.01 are comparable to the observational data[15].

## Data collection and filtration

To calculate the mantle redox state since the Archean and facilitate comparison with previous studies[11,13], we collected the oxygen fugacity ($fO_2$) using V/Ti and V/Sc redox proxies[13,18] by compiling the Precambrian whole-rock composition dataset of Phanerozoic-MORB-like basalts since 3.8 Ga (Supplementary Data 3). The whole-rock geochemical data of the basalts were assembled from the EarthChem rock database and refs. 13,65–67. Some samples with imprecise age ranges have been corrected by tracing back to the original publications by refs. 66,67 and this study.

To precisely calculate the oxygen fugacity and $P$-$T$ conditions of basalts, we first selected rocks with $SiO_2$ contents ranging from 45 to 54 wt%. Then, rocks with MgO contents below 8 wt% were filtered out to avoid the potential influence of clinopyroxene and magnetite fractionation and contamination from continental material[11,13]. Similarly, komatiite samples with MgO contents exceeding 18 wt% were excluded, as discussed earlier, and highly altered samples (e.g., loss on ignition greater than 6 wt%) were also excluded although some studies suggest that V is not affected by later alteration or metamorphism.

Previous studies used $(Nb/La)_{PM} \geq 0.75$ as a criterion to identify MORB-like basalts[11,13]. We applied stricter filtering criteria of $(Nb/La)_{PM} \geq 1$ to enhance MORB-like characteristics and minimize the influence of continental contamination (the continental crust typically has low Nb concentrations). Furthermore, none of the samples exhibit significant Nb and Ta (if present) negative anomalies, which are commonly interpreted as signatures of subduction-related influence.

Additionally, we examined other elements such as REEs; these samples display a slightly LREE-depleted or nearly flat pattern after chondrite normalization. This helps to exclude samples derived from enriched mantle sources such as modern plume-related ocean island basalts, continental flood basalts, continental intraplate continental basalts, and continental rift basalts, characterized by light REE enrichment. Additionally, we compared the temperature and pressure conditions associated with mantle plumes from numerical models and modern observations[68] to rule out plume-related influence (Fig. S10). The results show that, regardless of whether in the modern or Archean conditions, the extraction temperatures and pressures of plume-derived rocks are significantly higher than those recorded in our rock database (Fig. S10).

Finally, the V/Ti oxybarometer is specifically designed for peridotite-derived melts; therefore, we exclusively included lavas that satisfy geochemical criteria (e.g., CaO>13.81−0.274*MgO) to ensure their formation from peridotite sources[11]. Additionally, we included a previously reported and rigorously filtered set of MORB-like metamorphic rock samples in the database[6].

Other geological data were collected from the literature. Specifically, the arc magmatism frequency is based on refs. 66,67 the isotopic ratios of W were compiled from refs. 35–38 while zircon quantities were sourced from D.W.Y., G.T., & Z.X. (in preparation); the metamorphic T/P ratios and cold subduction rock types were sourced from ref. 69. Metamorphic cooling rates and folding rates were derived from ref. 49 data on the number of convergent margins, seawater Sr isotopes, and trends in biological evolution were summarized from refs. 50,70,71.

## $P$-$T$-$fO_2$ calculation of Basalt

To accurately determine the whole-rock $Fe^{3+}/\Sigma Fe$ in the basalt, it is essential to strictly constrain the $P$-$T$-$fO_2$ conditions[59]. Like the recent studies[11,13,18], we estimated the oxygen fugacity of basaltic melts using the V/Ti and V/Sc redox proxies. The V/Ti method is advantageous because it is more sensitive to mantle redox conditions and is not influenced by residual garnet or volatile degassing[11,13]. Since the partition coefficients of V and Ti in silicate minerals are $P$-$T$-dependent, we calculated the melting (extraction/formation/in-situ) $P$-$T$ conditions of MORB-like basalts using Fractionated-PT thermobarometer[68] before determining final $fO_2$: olivine fractionation is corrected for by incrementally adding equilibrium olivine back into the magma until the magma is in equilibrium with olivine having a Mg# equivalent to that of the average mantle residuum; the Fo of the mantle source was set to 0.9, consistent with the original reference. [68]. A key parameter affecting $P$-$T$ condition estimation is the $Fe^{3+}/\Sigma Fe$ ratio[68]. We used a bracketing approach (or squeeze theorem) to iteratively converge on the $P$-$T$-$fO_2$ conditions. Specifically, we first assigned an initial $Fe^{3+}/\Sigma Fe$ ratio, from which the corresponding $P$-$T$-$fO_2$ conditions were derived. This allowed us to calculate a thermodynamically consistent $Fe^{3+}/\Sigma Fe$ ratio.

The new ratio was then used as input for the next iteration, and the process was repeated until the $P$-$T$-$Fe^{3+}/\Sigma Fe$ values stabilized. The partition coefficients of V, Sc, and Ti for olivine, orthopyroxene, clinopyroxene, garnet, and spinel were also sourced from the literature[11,13,18].

Given that sodium content is susceptible to alteration effects, titanium content is used as a proxy to constrain the degree of partial melting in the calculation[11,13,18]. During the calculation process, for samples with melting pressures within the stability range of spinel peridotite, we applied a spinel peridotite partial melting model to determine $fO_2$. For samples within the stability range of garnet peridotite, a garnet peridotite melting model was used (Fig. 1). Most of the samples fell on the spinel field, with only a small number of samples falling on the garnet field. As a result, Archean basalts are modeled and interpreted as highly reduced, which contrasts with a previous empirical model suggesting that many Archean rocks may not have been reduced if their melting primarily occurred within the spinel stability field[15]. The further details for calculating $fO_2$ can be found in refs. 11,13.

Additionally, the $fO_2$ values were also calculated using the V/Sc oxybarometer for comparison[13], which is less affected by the melt fraction degree, especially under reduced conditions (Fig. 1). The oxygen fugacity values of melts from different time periods calculated by the two V-related methods are consistent within the range of uncertainty. For example, V-Sc oxygen fugacity of the Archean samples is QFM-1.1 ± 0.3, while the V-Ti oxygen fugacity in the same samples is QFM-1.4 ± 0.2; V-Sc oxygen fugacity of the Proterozoic samples is QFM-1 ± 0.2, while the V-Ti oxygen fugacity in the same samples is QFM-1.2 ± 0.2. All oxygen fugacity and $Fe^{3+}/\Sigma Fe$ calculations are averaged at intervals of 250 Ma to display historical trends. Smaller intervals would result in more divisions with no samples. Periods without samples used dashed lines to transition to the average values of adjacent periods. We also conducted a binning sensitivity test, and the results show that regardless of whether the binning is based on the episodic mobile-lid or single-lid tectonic regime (Fig. 5), or on broader divisions as the Archean and Proterozoic, the conclusion that the mantle $Fe^{3+}/\Sigma Fe$ ratio has doubled since the Archean remains unaffected.

## Whole-Rock $Fe^{3+}/\Sigma Fe$ calculation for Basalt

If the whole-rock composition of the melt, its formation $P$-$T$ conditions, and the $Fe^{3+}/\Sigma Fe$ are known, its oxygen fugacity can be easily calculated[59], and vice versa, once the $P$-$T$-$fO_2$ conditions are strictly constrained, we can calculate the whole-rock $Fe^{3+}/\Sigma Fe$ in the basaltic melts. Please note that the calculated $Fe^{3+}/\Sigma Fe$ here refers to the initial value of melt formation, as V-related oxybarometers record the source redox conditions, unaffected by degassing[11,13,18].

Since 1991, the empirical formula proposed by ref. 72. (Kress and Carmichael) has been widely used to relate the redox state of Fe to $fO_2$. However, recent studies have revealed that this method significantly overestimates the proportion of $Fe^{3+}/\Sigma Fe$ in whole-rock samples[27,28]. Specifically, extensive datasets indicate that the oxygen fugacity of modern MORB is approximately at the QFM buffer, with a $Fe^{3+}/\Sigma Fe$ ranging from 0.10 to 0.14[26–29]. The method from ref. 72, however, overestimates the $Fe^{3+}/\Sigma Fe$ by up to 0.07. This overestimation is also observed in Precambrian samples in the database[11] (Fig. S11).

Therefore, we do not rely on the method from ref. 72. to estimate the $Fe^{3+}/\Sigma Fe$ in basalt. Instead, we use thermodynamic methods for calculation. A recently developed tool ($fO_2$melt) enables the calculation of the relationship between $Fe^{3+}/\Sigma Fe$ and $fO_2$ under known $P$-$T$ conditions[59], embedding the latest thermodynamic model validated by experimental data.

Our computational approach utilizes $fO_2$melt to reverse-calculate iteratively the $Fe^{3+}/\Sigma Fe$ in the melt under specific $P$-$T$-$fO_2$ values and record the results. Subsequently, the recorded whole-rock composition is forward-simulated using thermodynamic modeling, and the

oxygen fugacity calculated is compared with the measured $fO_2$ values of the rocks. The results show that the two methods validate each other.

In addition, we applied the recently revised Fe-$fO_2$ empirical formula to calculate the $Fe^{3+}/\Sigma Fe$ in the melt[27,28]. This formula has been extensively validated against previous datasets. The results obtained from the revised empirical formula and the thermodynamic method are close to each other, but they are significantly lower than those obtained using the earlier method[72] (Fig. S11). The whole-rock $Fe^{3+}/\Sigma Fe$ corresponding to the $P$-$T$-$fO_2$ conditions recorded in the basalt database of previous studies[11,13] was also calculated for comparison to our results. It is worth noting that we respected these previously established $P$-$T$-$fO_2$ results and calculated the $Fe^{3+}/\Sigma Fe$ ratio based solely on these conditions. The variations in $P$-$T$-$fO_2$ estimates among different studies do not affect the overall conclusions.

## Whole-Rock $fO_2$ and $Fe^{3+}/\Sigma Fe$ in mantle source

When the $P$-$T$ conditions of the melt and the corresponding whole-rock $Fe^{3+}/\Sigma Fe$ are obtained, we can readily determine the redox characteristics of its mantle source using thermodynamic modeling (e.g., Fig. 1A). This allows us to calculate both the oxygen fugacity and the whole-rock $Fe^{3+}/\Sigma Fe$ of the mantle source. We iteratively assign different whole-rock $Fe^{3+}/\Sigma Fe$ of the mantle source to generate corresponding melt recorded from $P$-$T$- $Fe^{3+}/\Sigma Fe$. The redox state of the mantle source corresponding to each basaltic sample is determined and systematically recorded. The $Fe^{3+}/\Sigma Fe$ in the mantle source corresponding to zircons in mantle-derived melts of the Hadean can also be determined through thermodynamic modeling because their $P$-$T$-$fO_2$ conditions are already known (Supplementary Data 3).

## Data availability

The simulation data in this study have been deposited in the Zenodo database under https://doi.org/10.5281/zenodo.17577714. The rock sample data in this study are provided in the Supplementary Information.

## Code availability

The GeoPS software is available at http://www.geops.org/zh-cn/. The I2VIS code can be obtained at https://doi.org/10.5281/zenodo.10426375.

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

## Acknowledgements
We are grateful to Sanzhong Li, Paolo Angelo Sossi, and Jintuan Wang for valuable discussions on redox calculation. We also thank Chuntao Liu for the discussion for data filtering. W.Y.D. and X.X.Z. sincerely thank Guochun Zhao for his support and acknowledge funding from the National Natural Science Foundation of China Major Project (41890831) and the Hong Kong Research Grants Council (RGC) grants (JLFS/P-702/24 and 17307918). T.G. acknowledges the support from SNSF Research Grants 200021_192296 and 200021–231594, from ILP Task Force "Bio-geodynamics of the Lithosphere" and COST Grant CA23150 pan-EUROpean BIo-Geodynamics network (EUROBIG).

## Author contributions
W.Y.D. conceived the study. W.Y.D. and X.X.Z. conducted redox calculations, collected and processed the data, and wrote the original manuscript. W.Y.D., X.X.Z., T.G., and X.Z. processed the numerical model code and calculations. J.C.T. assisted in collecting W isotopes and wrote the corresponding discussion sections. T.G., X.Z. reviewed the original manuscript.

## Funding

## Competing interests
The authors declare no competing interests.
