## [Transparent Peer Review file · Nature Communications]

The Mantle $\text{Fe}^{3+}/\Sigma\text{Fe}$ Ratio Has Doubled Since the Early Archean

Corresponding Author: Dr Wenyong Duan

Version 0:

Reviewer comments:

Reviewer #1

(Remarks to the Author)

The authors developed a novel approach to extract information on mantle $f\text{O}_2$ through time, which combines numerical simulations and thermodynamic calculations using the latest models. This allows them to track $\text{Fe}^{3+}/\text{Fe}(\text{T})$ in basalts as a function of pressure and temperature in the sub- and suprasolidus regions of peridotite phase relations and during decompression melting, which are ultimately convertible to $f\text{O}_2$. I find this approach of forward modelling the most sophisticated yet, allowing to understand and quantify the complex relationships of these latter two parameters. In fact, the results are somewhat underexploited, as there are some interesting systematics (e.g. in Fig. 1A) that are not explored, though I understand that this is not the main thrust of the paper and there are limits to the length.

Overall, I think this will be of interest to a wide readership and would be publishable after moderate to major revisions. Nevertheless, my list of comments is very long, but if I see something that might be improvable, I say something. I hope the authors find these useful in order to craft a stronger manuscript or otherwise can easily rebut them.

Main concerns and suggestions:

1. I can see that the authors extract $\text{Fe}^{3+}/\text{Fe}(\text{T})$ of melts from the numerical experiments in Fig. 2F-J, and that the thermodynamic phase equilibria modelling can back-extract $\text{Fe}^{3+}/\text{Fe}(\text{T})$ from the V/Ti-based $f\text{O}_2$ of natural samples. Maybe I'm being thick, but can the authors better expose how the numerical experiments link to the natural samples? In general, the modelling, in particular the numerical one, could be described in more detail. Is it correct that the authors used forward thermodynamic modelling using PT of melting of natural samples to find the associated $\text{Fe}^{3+}/\text{Fe}(\text{T})$, from which they then extract $f\text{O}_2$. That is, their $f\text{O}_2$ estimate for the natural samples is completely unrelated to the V-based $f\text{O}_2$ estimate? The onus is on the authors to ensure their manuscript becomes accessible to a wide and non-specialised readership.

2. Related, the authors claim that their approach yields the most accurate and robust insights yet, but ultimately, constraints on mantle redox evolution rest with the natural samples and their shortcomings: (a) unclear tectonic setting (in fact, unlikely MOR-setting for the vast majority of Archean basalts (Kamber 2015 Precambr Res; Puchtel and Arndt 2025 TOG; Brown+24 JGeolSoc) – this is important because we know that mantle $f\text{O}_2$ varies as a function of tectonic setting (Cottrell+22 in Geophys Monogr 266); (b) uneven sampling through time, with some sparsely sampled periods; (c) uncertainties related to uncertainties in PT estimates from natural samples. Thus, the information from the modelling, however sophisticated, cannot be more accurate and precise than that derived from natural samples.

3. There are quite a number of vague, sweeping or inaccurate statements, and some ad hoc statements, throughout the main part of the text that are not very informative and therefore leave more questions than answers. Instances are given below.

4. The authors variably refer to redox state, oxygen fugacity, oxidation degree, oxidation budget, $\text{Fe}^{3+}/\text{Fe}(\text{T})$ The mantle redox state reflects the sum of several multivalent elements, of which iron is certainly overall the most important one, but S, C

and H are not negligible (Evans 2006 Geology). I suggest to make sure fO_2 , redox state (or redox budget), and $Fe^{3+}/Fe(T)$ are accurately used throughout the text, and to refrain from using additional terms (e.g. oxidation degree, oxidation budget etc).

5. Given continued uncertainties regarding earliest Earth processes and the physicochemical-dynamic conditions attached to them, the authors are of course entitled to their preferred interpretation of why the fO_2 of Earth's mantle changed after the Hadean. However, the arguments against upward mixing of redox budget arising from disproportionated iron, and the arguments in favour of komatiite serpentinisation followed by subduction are substantially less clear-cut than presented, for reasons I detail below. This discussion could be a lot more balanced.

6. Furthermore, the authors take evidence from short-lived radiogenic isotopes to argue for separate upper and lower mantle reservoirs, but this seems too simplistic, as early-generated heterogeneities survive to the present day (as sampled by OIBs) and are thought to reside in geographically restricted regions that are not identical to the lower mantle as a whole.

7. The authors dismiss komatiites as "enigmatic". This is a misrepresentation of the field, as komatiites have extremely well-constrained petrogeneses, which may be complicated by their derivation from thermochemically anomalous plumes, and it is true that their pressures of formation are difficult to estimate. I would accept this as an argument in favour of focusing on spreading ridge-derived samples (MORB-like basalts and picrites). By the same token, the authors should acknowledge that few if any Archaean basalts are truly spreading ridge-derived.

8. The authors assume a modern subduction redox budget to estimate whether this could have oxidised the mantle to present-day levels. However, we know that the atmosphere was oxygenated to just 1% PAL as late as 2.4 Ga ago, and that oceanic bottom waters became oxygenated only in the Neoproterozoic (Stolper+Keller 18 Nature), before which time there would have been very little dissolved sulphate, which is the most potent oxidant (Tomkins+Evans 15 EPSL). The subducted redox budget surely must have been variable through time. Or maybe I misunderstood what the authors are aiming at. It also seems that they address this in some detail in a paper under consideration (ref. 38), this should be clarified.

9. There are some sentences with awkward English that would benefit from polishing.

10. The figures and their captions can be much improved. There is too much, in part easily overlooked, small font in the panels themselves. The captions are so terse as to be uninformative. IMO, readers should be able to understand what the diagrams show and what arguments they support without reading through the entire manuscript.

11. Throughout, I suggest to use italics for state variables, such as T , P , f (fugacity) etc.

In-line comments and suggestions

Title: what is oxidation degree? fO_2 ? Redox budget? $Fe^{3+}/Fe(T)$? Please be more precise, also in the abstract and elsewhere

11-13 It is clear what the authors want to express in the first sentence but the wording is inaccurate. What is oxygen-rich habitability? The atmosphere is now O_2 -rich. What the mantle regulates is the exchange of redox budget, which ultimately enabled O_2 to accumulate in the atmosphere, thereby affecting how life evolved

13 our planet should be capitalised, also should be "Earth's"

14 remains a subject of

16 they are not all basalts, but also picrites according to the supp table. As an aside, the mere fact that there are so many basalts implies that Archaean ambient mantle TP could not have been very hot. If the samples are basalts due to extensive differentiation, then they are not appropriate to estimate mantle fO_2 from them

16-17 when you are referring to a ridge setting specifically, there is no need to again mention "geodynamic settings". Perhaps "modern conditions"

20 demonstrate

20 clearly, measured $Fe^{3+}/Fe(T)$ are not reliable and if you estimated the values based on V/Sc or V/Ti , then your result cannot be more reliable than the information contained in the element ratios? Besides, Zhang et al. (2024, cited) already presented whole-rock $Fe^{3+}/Fe(T)$ thermodynamics-based estimates. If yours are significantly improved, or novel in the sense that you used a different approach, then this should be expressed differently – this relates to my main comment above

24 please be more specific here: what kind of geological evidence? Below you also refer to biological evolution

25-26 tectonic reorganisation - ok. But what do you mean by geological reorganisation?

I am aware that habitability is a catchword, but "oxygen-rich habitability" just is not expressed well, see comment above

27 I understand there is a word count limit, but this reference to biological evolution is too vague

37-38 exert... influence, or simply “have... effects”. Effects are not exerted, I think.

ref. 6 was updated in Aulbach+19 SciRep - and using both V/Sc and Fe³⁺/Fe(T)! - so you could cite this instead

here and throughout: QFM units don't exist. It is just a reference buffer. fO₂ was lower by ... orders of magnitude (or log units, if you will) – throughout manuscript

“tends”. Also, please be more specific as we know it is only the garnet-bearing mantle down to metal saturation at ~8-10 GPa where fO₂ decreases with increasing pressure. At P<3-3.5 GPa, the fO₂-depth profile is more complex, as demonstrated from phase equilibria and empirical models (Stolper+20 AmMiner; Birner+24 Nature)

58-61 This sentence does not reflect the state of the art:

1. Aulbach+Stagno 16 (cited) did correct for the fO₂ decrease with increasing pressure and did find significant differences between Archaean and modern samples.
2. Zhang+24 (cited) wrongly corrected their basalt V/Ti-derived fO₂ for the reduction of the melt with decreasing pressure after leaving the source. However, the V/Ti-derived fO₂ is already that of the source.
3. Both biased their Archaean, more deeply-derived samples (due to higher TP) to too-high fO₂ by assuming the garnet-peridotite fO₂-pressure trend could be extrapolated to 1 GPa, but we know since Stolper+20 that this is not the case.

these are “elemental ratios”. Furthermore, this statement is blatantly incorrect: neither ref. (6) nor refs. (11,13,14) used direct geochemical ratios, as (6) accounted for P (T effect not known at the time), (14) for T and (11) accounted for PTX (though made a mistake), and (13) accounted for TX. It is true only for (12) that V/Sc was used directly. (16) should not be cited here - it's a review paper and we were well aware of PTX effects

“the” depleted mantle is understood by many to be the MORB source. What your fig. 1A seems to show is mantle that is residual from extraction of various extents of melting in the suprasolidus region as a function of distance from the solidus (we later learn in the methods that the bulk composition of the model corresponds to a depleted MORB mantle)

You show the mantle at constant Fe³⁺/Fe(T), so what do you mean by “often”? where in the figure is it coupled vs. decoupled?

69-70 This is not shown in Fig. 1? Qualitatively, if Fe³⁺ behaves like a moderately incompatible element (O'Neill+18, cited), one would expect the ratio to decrease with increasing melt F, which is a function of T, all other things being equal. This seems to be shown in Fig. 2.

not just depleted mantle, but any mantle, or more specifically, peridotite

perhaps refer redox budget sensu Evans (2006) and then stick with it

72-76 I think you should refer to the phase eq. and empirical modelling of Stolper+20 (Am Miner) and Birner+24 (cited) here. Stolper highlighted how this decoupling is related to changing mineralogy across peridotite facies as a function of pressure

please don't refer to QFM units. Refer to fO₂ and several orders of magnitude (or log units)

I think “historical” is - historically - used for time since humans keep records. Perhaps “trends over geological time” or similar

82-84 I don't see how this is more accurate? Melt fO₂ also depends on PTX, and unlike elemental ratios, Fe³⁺/Fe(T) can be changed by redox interactions (due to redox melting, degassing etc). Besides, measured Fe³⁺/Fe(T) of ancient rocks are unreliable due to alteration. Therefore, ultimately, your Fe³⁺/Fe(T) modelling is novel and possibly more accurate, but it remains a model that, to be useful, you have to compare against a rock record

even if this is explained as part of the Methods later, this reference to “thermodynamic and geochemical methods” is much too vague. A few more sentences are needed here, what these methods are, so we understand broadly what you did without having to consult the Methods

yes, the modelling is supercool, novel and a major advance. But this does not make observations from the sparse rock record itself any more robust

please be precise regarding what it is that has doubled. I think you mean Fe³⁺/Fe(T), which may be the most abundant multivalent element, but does not alone fix the redox budget (see Evans 2006 Geology)

It is not the redox modeling that is modern or Archaean, but Redox modeling for modern and Archean conditions (what you call thermodynamic conditions)

98-99 This needs much better justification. You could say that komatiites are plume-related and therefore may sample

thermochemically anomalous sources, whereas basalts may sample an ambient mantle source representing a better-constrained, less variable system. You should also be honest and acknowledge in your manuscript that virtually no Archaean basalts are spreading ridge-derived (Puchtel and Arndt 2025 TOG; Brown+24 JGeoSoc). The samples you describe as "MORB-like" are actually basalts associated with melting in intraplate settings and erupted through attenuated continental mantle. The only true Archaean spreading-ridge derived material is now sampled as cratonic eclogite (no need to cite, but see Aulbach+Smart 23 AREPS if you want to know why)

there is no representative TP for the Archaean period, as TP estimates vary by >200 oC (e.g. Herzberg+2010, cited vs. Aulbach+Arndt 19 EPSL)

a totally expected result for all tectonic settings where decompression melting occurs...

Again, given what we know, this is a trivial finding. Rather than "highlights", perhaps it validates your modelling approach?

please mind language - what is melting T? T of solidus intersection? It is afterwards, when the melt takes with it heat, that cooling occurs. But this is not specific to MOR ridges, so I am not sure what you mean

please explain here the presence of cold drips and the mixing, or refer to a text/Methods where we can read up on it - is this related to a pre-plate tectonic regime?

122-123 not sure this sentence makes sense as written? What is an intricate "internal thermodynamic equilibrium"? Why internal? Aren't the physical (not geophysical, presumably?) parameters (PT-fO₂) dictating the equilibrium? Aren't the thermodynamic calculations based on experimental petrology, first and foremost?

imposed rather than provided?

"MORB with fO₂ near QFM"

You cite Hirschmann 2023 in a different context, but you should acknowledge here - and thereby support your choice of Fe³⁺/Fe(T) of 0.02 - that he suggested that this is the maximum ratio in case there was a mantle redox evolution since the Archaean

Please see Fig. 2b in Aulbach+19 SciRep - this is roughly similar to the ratios we reconstructed for Archaean eclogite xenoliths interpreted as metamorphosed subducted MOR-derived oceanic crust

140-141 There is no rationale provided here why it is increased T_p rather than an intrinsically lower fO₂ "promoted reduction of basaltic melts"? Maybe a few steps in the chain of arguments are missing here? (higher T_p = deeper melting = lower fO₂ in the garnet-bearing mantle source = lower fO₂ in the melt which separated and reduced)

A doubled Fe³⁺/Fe(T) does not equate doubly oxidised - please be more exact

144-146 Keeping your broad readership in mind, perhaps it would be better to say that a higher T_p results in a lower fO₂ via its effect on the pressure of melting (which is deeper), at least in the garnet peridotite facies - at least this is what you suggest? Also, we've known this before, so this is not a new finding from the present study. This is why AS16 and Zhang+24 applied a P correction (though both wrongly extrapolating the garnet peridotite fO₂-P profile to 1 GPa). AS16 actually estimated P of melting from their preferred TP evolution curve, so the link between TP-P-fO₂ is clear. Finally, it is necessary to be very careful with the terminology: here you refer to a lower mantle redox state despite constant Fe³⁺/Fe(T) - I think you mean fO₂

150-154 1st sentence: keeping your broad readership in mind, perhaps it would be better to say that a higher T_p results in a lower fO₂ via its effect on the pressure of melting (which is deeper), at least in the garnet peridotite facies - this is what you suggest? Also, we've known this before, so this is not a new finding from the present study. This is why AS16 and Zhang+24 applied a P correction (though wrongly extrapolating the garnet peridotite fO₂-P profile to 1 GPa). AS16 actually estimated P of melting from their preferred TP evolution curve, so the link is clear. Finally, it is necessary to be very careful with the terminology: here you refer to a lower mantle redox state despite constant Fe³⁺/Fe(T) - I think you mean fO₂

looking at Stolper+20 AmMiner, it is clear that 2.4 GPa is below P where the simple relationship between pressure and fO₂ in the garnet peridotite facies breaks down and fO₂ varies significantly and non-linearly as a function of changing modal abundances

"previous" - give reference. If Zhang+24, they overestimated Archaean fO₂ in two ways, as I explain elsewhere in this review). So should we be worried about the consistency with your approach?

what do you mean by in-situ? what do you mean by Mg-calibrated? Please note:

(1) Zhang+24 made the same 2 mistakes when applying their corrections to the metabasalts reported in AS16 that they did with the basalt database, and their Archaean metabasalt-based fO₂ estimates are also too high as a result

(2) I happen to be familiar with ref. 6: AS16 did more than just look at MgO content. We used major and trace element constraints to understand the samples' petrogenesis, which allowed us to exclude samples with cumulate protoliths, or those

that had been metasomatised or that were LREE-enriched suggesting derivation from an enriched source or melting in the presence of lithospheric lid - this was done to ensure that the comparison to forward-modelled melt V/Sc (to estimate source fO_2) and to modern MORB is justified. For many of the sample suites, AS16 used reported radiogenic isotope systematics to further demonstrate their derivation from a depleted mantle source, as is true for modern MORB, and given that at least mildly depleted sources existed in the Archaean. This was further corroborated in Aulbach+Arndt19 EPSL where many of the same eclogite suites for which AS16 estimated fO_2 are shown to have initial $^{87}Sr/^{86}Sr$ indicative of a protolith derived from depleted mantle.

(3) Your and other's basalt database, on the other hand, includes in particular Archaean basalt samples that are not spreading ridge-derived, with $(Nb/La)_N$ down to 0.75, and the onus is on you to demonstrate that these samples record the fO_2 of a convecting mantle source that can be usefully compared to the modern MORB source. That is, without complications that arise from mixing tectonic settings.

any back-calculated result would match some result from the numerical simulations?

redox state of the Archean mantle as proxied by its $Fe^{3+}/Fe(T)$

what is the oxidation degree and how do you double it?

184-185 again, the effect of TP is indirect, and I think it would be better to make it more explicit. "Deeper onset of melting owing to higher Archean mantle TP" alone might be misinterpreted

not only that, but it confirms earlier findings as per (6, 13, 14) – worth mentioning here

MORB with fO_2 near the QFM oxygen buffer

188-189 sure, but this last statement is hanging in the air, as it is not followed by some statement as to whether this needs further investigation, or whether it conflicts with some evidence. Perhaps you can just leave it out.

this is likely, but currently they are "thought to represent" Archaean - or were $^{187}Os/^{188}Os$ reported for these samples?

in Figure 4

then we calculated

which other redox parameters?

what kind of melting? Batch or fractional? It makes a difference for how concentrations of incompatible components in the residue evolve

high degrees of partial melting

suggests that Archean mantle was more reduced than the modern mantle

214-217 Please add a statement/caveat regarding uncertain provenance/tectonic setting of basalts, a potential preservation bias linked to the supercontinent cycle and poor statistic coverage of certain eras.

218-219 by the time a melt is able to saturate zircon, it is so far removed/processed from the original mantle-derived melt that it is questionable at best that the fO_2 they record reflect the mantle rather than some later process

232-233 I suggest to rewrite "... this hypothesis, as the accretion of meteorites with negative ... lowered the W isotopic composition of the silicate mantle with an initial positive W isotope composition (+200...) in the wake of core formation and evolution at high Hf/W ".

when did these meteorites arrive? Given how short-lived the system is, there must be an estimate for the timing

should be ref 29? Might need to check neighbouring references, too

235-237 please rewrite, this sentence is not well constructed

how do you define habitability? Life evolved even as the atmosphere contained $\ll 1\%$ PAL O_2 . So this change in mantle O_2 would have affected the course of evolution of life, but not hindered life per se.

246-249 this is much too simplistic an argument, and untenable. Even modern OIBs still contain components that must have been generated during earliest Earth evolution including W (e.g. Mundl+17 Science) and Nd (Horan+18 EPSL). The sources likely reside in long-lived heterogeneities (possibly atop the core) that cannot be framed in terms of upper vs. lower mantle

"Archean rocks have W and Nd isotopic signatures..." (what was fractionated were the parent and daughter elements

not today, but if iron disproportionation and extraction of metal to the core is real, then it would have been at the time of

late accretion.

Sure, but in modelling a lot depends on parameters, including some that are not well-constrained. I am not convinced that the last word has been spoken on whether or not metal formed by disproportionation can be extracted to the core. Hirschman 23 EPSL (cited, though not in this context), invoked a basal magma ocean to explain Earth's redox evolution, and recent work (Boukaré+25 Nature) suggests that this would have been inescapable on Earth. It seems difficult to argue that metal formed by Fe²⁺ disproportionation on top of the core (expected at this pressure) would not have been able to be efficiently extracted to the core even if Boukaré+ were not concerned with this particular aspect.

what do you mean by “not show mantle oxidation mechanism”? This is awkward English. Is it correct to simply say it has a more reduced mantle than Earth? Despite having a metallic core, a lower mantle composed of perovskite and despite having had a (basal) magma ocean, which are pre-requisites for this mechanism to work?

I don't agree that this is more plausible. In order to have permanent mantle oxidation you need to permanently sequester reducing power, which you can do by extracting metal to the core or having hydrodynamic escape of H₂ which surely would have been much more efficient during earliest planetary evolution (even if it still occurs at an exceedingly low level today). Taking oxidising power from the ocean-atmosphere system and sticking it into the mantle means you are reducing the surface reservoirs, thereby inhibiting the advent of the GOE. Besides, this subducted material and redox budget would eventually resurface (we see it in OIBs and even MORBs), resulting in no net change. To the extent that mantle fO₂ affects the fO₂ of magmatic gases, you would, after the onset of subduction of oxidised material, still have a shallow reduced mantle degassing reduced volatile species that are O₂ sinks. In fact, before the oxygenation of ocean bottom waters in the Neoproterozoic, the redox budget of what subducts might not have been hugely positive (see Stolper+Keller 18 Nature; Tomkins+Evans 15 EPSL), and you would need to subduct a great volume to achieve an effect on the mantle, which then must be reconciled with other evidence for the maximum mass of subducted material in the mantle.

that's 500 Myr later than what you call for at the beginning of the paragraph.

263-275 please clarify overlaps with the manuscript under consideration as ref. 38

they are transporting redox budget, not fluxes. Besides, this redox budget is then “missing” at the surface and unavailable to oxidise surficial reservoirs.

ref. 38 is miscited in this place. They showed that komatiite serpentinisation produces H₂ and that's it. They never mention subduction or mantle redox. Furthermore, I question that the H₂ so produced could be quantitatively lost from the Earth system, as hydrodynamic escape would have been highly inefficient by the time komatiites were emplaced

271-274 (1) It is insufficient that ultra-oxidised oceanic crust occurs. What matters is its mass. E.g. Cratonic eclogite xenoliths undoubtedly represented subducted oceanic crust, including having fractionated O isotopic compositions pointing to seawater alteration, have consistently much lower Fe³⁺/Fe(T) than what you state here (no need to cite, I am just referencing in support of my statement: Aulbach+Smart 23 AREPS, Aulbach+22 JPet, Aulbach+19 SciRep). (2) It is not admissible to assume a modern subduction redox budget when we know that the atmosphere was oxygenated to just 1% PAL as late as 2.4 Ga ago, and that oceanic bottom waters became oxygenated only in the Neoproterozoic (Stolper+Keller 18 Nature), before which time there would have been very little dissolved sulphate, which is the most potent oxidant (Tomkins+Evans 15 EPSL). (3) I wonder where the idea came from that komatiites have relatively high Fe³⁺? If measured Fe³⁺/Fe(T) were reliable, someone would have exploited that before, but they are not. Berry+ 08 Nature were able to make an estimate on a melt inclusion in komatiite and obtained 0.10, similar to modern MORB.

parent-daughter elements are fractionated not the isotopic compositions, which trace different sources with different isotope compositions owing to elemental fractionation at some time when the system was still alive. See my comment above regarding the inadmissibility of casting these different sources of ancient signatures in terms of upper vs. lower mantle.

283-284 seems an ad hoc statement and requires a little more info

287-288 this was already proposed by Aulbach+19 SciRep based on the eclogite and komatiite record.

288-289 please see my comment above. This is worded peculiarly and moreover falsely diminishes how ref. 6 or 11 scrutinised their database (although 6 went a step further to demonstrate derivation from a depleted mantle source) – see similar comment above

299 why would there have been separate upper and lower mantles? Where are isotopic anomalies from now extinct isotope systems in modern OIBs from when we know for sure we have whole-mantle convection? This paragraph is based on a series of highly uncertain findings and perhaps could be shortened to two sentences invoking the magmatic lull (which was not global, however) as a potential sign of sluggish tectonics that might explain the lack of mantle fO₂ evolution during that period

ref. 6 did not invoke a Neoproterozoic rise (though we were tempted)

hm. This was surface driven (increased photosynthesis and organic carbon burial, plus more; e.g. Och+12 EarthSciRev), which ultimately caused oxygenation of oceanic bottom waters. This increased the subduction flux of redox

budget, as seen by the advent of certain convergent-margin ore deposits (Tomkins+Evans 15 EPSL), then may have trickled down into the mantle

313-326 last paragraph is too wordy, repetitive and vague. Could be shortened to a crisp and strong final statement.

propose

315 except for large impacts, nothing of what follows is abrupt.

greater quantities of... fluxes makes no sense, the flux already comprises a quantity (referenced to a time unit). Please reword.

I disagree. Some estimates, including Aulbach+Arndt 19 EPSL, are lower (100-150 oC). Low-T estimates from basalts around 2.9 Ga that support a warm rather than hot Archaean ambient mantle (e.g. Zhang+24 Natcomms, cited), continue to be ignored. But that's a different story...

please specify somewhere the composition of the mantle that you modelled ($Fe^{3+}/Fe(T)$ is not enough to characterise the system).

remove first comma

337 how thick in km?

what do you mean by "melt-bearing peridotite"? The melt was similarly dense as the peridotite and failed to separate? Or simply the peridotite being above its solidus T?

basaltic or picritic?

341 why hydrated?

344-348 Are we still in Fig. S6? Can you refer to panels, additionally?

I think the concept of ambient and excess TP is useful in that can explain why komatiites form while ocean basins are made up of picrites, so there is no conflict at all. Syntax in last part of sentence needs fixing

354-357 It is fine to not consider komatiites in your work and to say that the petrogenesis of spreading-ridge derived basalts, but this is an inexact summary of the state of the art, and a weak way to dismiss the komatiite record. With the exception of a single komatiite (Commondale), where subduction MAY be a possibility, there is no evidence for komatiite in subduction zones, or from particularly wet sources. No one says komatiites form at the core-mantle boundary. Please duly distinguish between mantle sources and melts. Please refer to Waterton+Arndt 25 (In: Hofmann+ eds The Archaean Earth: Tempos and Events 2nd Edition of The Precambrian Earth), Puchtel+22 ChemGeol, Puchtel+Arndt 25 TOG, Herzberg 16 JPet etc. By the same token, why not acknowledge that virtually no Archaean basalt is strictly comparable to modern MORB (and hence mantle sources) because they were emplaced on continental margins?

where is the evidence for that? Sources such as Early Enriched Reservoir and similar that were sluggishly mixed throughout the mantle, are though to reside in geographically isolated parts of the (lower) mantle and do not require separate UM and LM compositions

please don't refer to komatiites as highly enigmatic. They are not.

how did accreted meteorites get into the lower mantle, if not through the upper mantle? Wouldn't they have equilibrated with an upper magma ocean more efficiently than with a partially solidified lower mantle?

370-372 In the discussion, you say that the mantles were already mixed and that plate tectonics are weak. Here, you want to bring up deep reduced mantle. Can you have it both ways?

ref. 55: this is a different citation from the two you referenced above when talking about the spread of Archaean TP

I am all for making thick Archaean mantle by decompression at high temperatures (in my reading, 1600 oC, require excess TP), but please note that these are present-day thicknesses (for intact cratons). Ancient diamond formation supports that cratons already reached thicknesses of 150-180 km.

447-448 fix this sentence.

not crystal-clear what "these" refers to

Please specify this is the depleted MORB mantle (lithosphere contains a lot of variably depleted mantle, too).

up to 1.2 orders of magnitude? Average 1.2? Surely this must depend on the modelled conditions??

please explain what motivates this choice of 0.02 and 0.04

and are these increments then aggregated?

why 0.04 and 0.01 here, but 0.04 and 0.02 above?

what kind of age correction? Why?

fix materia

"low degree of alteration". How was this assessed? Did you use LOI?

As importantly, Nb/LaPM <1 points to a continental influence, either via crustal assimilation, or by formation in convergent margins

One would hope that you retained samples with an arc magmatism likelihood <<10%. I think this is expressed wrongly?

how many of how many is "most"?

what are zircon quantities?

528-529 Maybe, but a recent study finds that V/Sc is less sensitive to mantle source and melt fraction (Liu+25 GPL). For the P you report, garnet-present melting is subordinate, anyways (as per your next paragraph).

what is the P interval? Is it itself T-dependent?

more specifically a previous empirical model

awkward sentence, please rewrite

545 following ref. 11 is bad news because they calculated fO_2 based on V/Ti and V/Sc, which records source fO_2 directly (forward-modelling of peridotite melts) rather than low-P crystallisation, but then applied a correction assuming that the melt self-reduces during decompression after separating from the source. This reduction is relevant to olivine-melt V partitioning, and to $Fe^{3+}/Fe(T)$ recorded in the emplaced melt, but not to forward-modelled V/Sc or V/Ti. I gather from a later paragraph that you did not apply this wrong correction.

It is very important that you clearly expose how you estimated uncertainties of individual fO_2 estimates, which will have contributions from uncertainties in PT estimates and on source composition

562-563 See comment to line 545. The statement itself is inaccurate because olivine-melt V partitioning is a V-based oxybarometer that does not record fO_2 of melt formation (in the mantle source), but of low-pressure saturation of liquidus olivine when the melt may have self-reduced.

last sentence is a bit strange. Perhaps say it is also observed in the database assembled in this study, or similar

I don't clearly understand how this works. You want to learn about $Fe^{3+}/Fe(T)$ in natural samples by comparison of sample-derived V/Ti-based fO_2 to modelling results. You only have one independent $Fe^{3+}/Fe(T)$ estimate (from the model). What am I missing? Please make it more explicit.

the (inherited) zircons, but not the melts (or lavas) are in the record. The melts were likely SiO_2 -rich and not mantle-derived, but either remelted or strongly differentiated from a mantle-derived melt, so as to be able to saturate in zircon. Either would imply that zircon fO_2 does not record that of the mantle-derived melt and much less that of the mantle source.

I attach my comments to figures and captions in the main text and in the supplement directly in the pdfs.

8 April 2025 Sonja Aulbach

Reviewer #2

(Remarks to the Author)

This manuscript explores the redox evolution of Earth's mantle since the early Archean, employing thermodynamic-thermomechanical numerical simulations and a comprehensive database of mid-ocean ridge basalt-like (MORB-like) samples. The authors perform advanced numerical simulations to calculate the redox state of basaltic melts under varying mantle potential temperatures and mantle Fe^{3+}/Fe_{Total} ratios. The study suggests that the mantle's average oxidation degree has approximately doubled since the early Archean, reflecting significant geological and tectonic reorganization events.

The main innovation lies in the integrated approach combining empirical observations with high-resolution numerical simulations, highlighting the intrinsic coupling between Earth's oxygen-rich habitability, biological evolution, and tectono-

magmatic processes.

However, this study has some shortcomings in the selection and organization of MORB-like basalt samples, which undermine the robustness of its conclusions. These issues include: (1) There are many repeated samples with consistent chemical compositions in the dataset, but they display different ages, P-T-fO₂ conditions and Fe³⁺/Fe^{Total} values. (2) The machine learning (ML) method used in this study may have several drawbacks, leading to some uncertainties in MORB-like sample filtration. (3) The quantity of MORB-like samples (if they are) varies significantly over time and is insufficient overall. Therefore, the trend of basalt oxygen fugacity presented in Figure 3 may have some problems. As the following discussions are all dependent on this trend, I suggest that the authors should deal with the above-mentioned comments first.

1. The descriptions of Figure 1 are overly simplistic and lack the necessary details to fully convey the key information (After reading the whole manuscript, most of the figure captions have similar problems). How could you obtain this P-T diagram? What is the meaning of the five-pointed stars in Figure 1A? How about the white dotted lines? All of these should require detailed introductions in both manuscript and figure captions.
2. Data Collection and Filtration: The authors used a ML method published by Liu et al. (2024) to filter Archean MORB-like basalt data. However, the training dataset listed in Liu et al. (2024) was subdivided into subduction and non-subduction groups, respectively. The latter contains not only many MORBs, but also a lot of oceanic island basalts (OIBs), oceanic plateau basalts (OPBs), continental flood basalts (CFBs), continental intraplate basalts (CIBs) and continental rift basalts (CRBs) and so on. Therefore, samples with the prediction results > 90% do not necessarily originate from the mid-oceanic ridge magmatism. My main concern is how the authors determine that these Archean basalts are MORB-like, rather than OIB-like, within plate-like or others.
3. After detailed examination of the listed dataset in Liu et al. (2024), the ML method may have some drawbacks: (1) Nearly two thousand lines of basalt data are older than Cenozoic (65 Ma), which might not be suitable for using as training data; (2) The generation of CFBs, CIBs and CRBs was commonly accompanied by various degrees of continental contamination. However, the influences of continental contamination were not precluded. Obviously, it is confusing to use samples that are strongly affected by continental materials to discuss their origins and tectonic settings; (3) The listed arc basalts are composed of many samples from back-arc basin areas. However, the simultaneous influences of back-arc spreading and mantle plume were recently proposed for the tectonic evolution of the West Philippine Basin. In essence, the chemical compositions of back-arc basin basalts (BABBs) vary in a large scale, including MORBs, normal arc basalts, and OIBs and so on, suggesting that the generation of BABBs may be complicated. Therefore, labeling the BABBs simply as subduction origins may be unreasonable; and (4) The elements of U, Pb, Ba, Sr and Rb (and the ratios made up by these elements) were also applied to the ML model. But these elements can be easily affected by post-magmatic alterations. In fact, I am concerned about the impact of element mobility on the predicted results of Archean basalts, which were mainly implemented in the 'Black Box'. Therefore, I suggest that the authors should apply a more reasonable method to screen the Archean MORB-like samples.
4. P-T-fO₂ Calculation of Basalt: The calculated parameters for Fractionated-PT thermobarometer should be introduced. I notice that Table S2 contains many repeat samples (e.g., samples Z2967-49 and S z-1) in lists of 'Data in Zhang et al. (2024)' and 'Data in Gao et al. (2022)'. However, the same sample displays distinct P-T-fO₂ conditions and calculated Fe³⁺/Fe^{Total} values in each data list. In addition, many repeat samples in list of 'MORB-like samples in this study' have different ages. Therefore, I consider that the organization of basalt dataset in this manuscript should be significantly improved.
5. The quantity of MORB-like samples fluctuates dramatically over time, and the overall sample quantity is insufficient. With these points in mind, I am skeptical about the authenticity of the trend shown in Figure 3. As the following discussions in this manuscript are all dependent on this trend, I suggest that the authors should deal with the above-mentioned comments first.

Reviewer #3

(Remarks to the Author)

Review on "The Mantle Oxidation Degree Has Doubled Since the Early Archean" by Zhu et al.

The secular redox evolution of the mantle attracts broad interests because it may influence the long-term evolution of the atmosphere. The redox evolution of the mantle could be dictated by oxygen fugacity (fO₂) or redox capacity (Fe³⁺/Σ Fe). Whether the fO₂ has increased in Archean remains hotly disputed (Zhang et al., 2024; Li and Lee, 2004; Aulbach and Stagno, 2016). In this study, the authors inferred the redox evolution of the mantle from the perspective of redox capacity (Fe³⁺/Σ Fe). They estimated circuitously the Fe³⁺/Σ Fe of the mantle from calculated T, P and fO₂. Despite that the authors performed arduous thermodynamic-thermomechanical simulations, the results are not novel, and even be wrong for the following reasons.

1. In estimation of Fe³⁺/Σ Fe in the mantle, the authors used fO₂ estimated by Zhang et al. (2024) and Gao et al. (2022). Both of the studies show that the fO₂ of mantle-derived basalt increased around 2.5 Ga. Therefore, it's foreseeable that the estimated mantle Fe³⁺/Σ Fe will increase. I wonder what the result will be if the authors use a constant fO₂ (as proposed by Li and Lee, 2004) in their calculations.
2. If the increase of mantle Fe³⁺/Σ Fe at stage 4 (3.2-1.8 Ga) was caused by subduction of oxidized surficial materials into the deep mantle, then from the point of mass-balance, we should expect a decrease in O₂ of the atmosphere. However, in the contrary, the O₂ content of the atmosphere increased at this stage. In addition, if the mantle was remoulded in Fe³⁺/Σ Fe, the bulk composition of the mantle should also change due to subduction. That means the authors cannot use a constant mantle composition in the simulations.
3. The estimated mantle Fe³⁺/Σ Fe has a large uncertainty, which may lead to misleading rather than clarity.

Minor points:

The title is pompous, the word “oxidation degree” is unclear. If the authors mean “ $\text{Fe}^{3+}/\Sigma \text{Fe}$ ratio, they should use $\text{Fe}^{3+}/\Sigma \text{Fe}$.”

Lines 11-13 The expression is vague, how mantle’s redox properties enables cycling of redox sensitive materials? Or the redox state of the mantle regulated by material cycling? What’s the definition of “redox-sensitive materials”

Lines 33-34: Yes, the mass of the mantle is large, however, the MORBs are derived from the very shallow part of the mantle, how could the authors use MORB represent the whole mantle?

Line 39-41: It’s very abrupt to compare the redox state of Earth with other terrestrial bodies, what’s the point here?

Line 62: change “the mantle and derived-melt redox state” to “the redox state of the mantle and mantle-derived melts”

Line 70: According to which study, the authors alleged “The $\text{Fe}^{3+}/\text{Fe}^{\text{tot}}$ in melts decreases with increasing temperature”?

Line 78-79: That’s also the reason why Aulbach and Stagno (2016) normalized the V/Sc ratio to constant P of 1 GPa.

Version 1:

Reviewer comments:

Reviewer #1

(Remarks to the Author)

I have read the authors’ rebuttal letter and the manuscript version with tracked changes. I am impressed with the detailed, careful and mostly well-argued responses they have given. Although I don’t agree with all of the authors’ reasonings, this concerns fine details that have no significant influence on the main outcomes of the study, which is novel and robust as far as I can tell.

Here, I first discuss remaining instances where I am not in complete agreement with the authors with respect to their rebuttal, the consideration of some of which is at the authors’ discretion. Then, I make detailed comments on the revised text, many of which do require the authors’ attention.

1. For example, to my main point #4, they say that “although at greater depths, carbon may exist as C^0 . Upon ascent, the transformation of C^0 to C^{4+} would consume Fe^{3+} (i.e., cause reduction), but the overall redox budget remains conserved in MORB mantle source”. I think this is only true so long as mantle remains subsolidus, but if you generate CO_2 , then you have redox melting, in which case the redox budget in the residue should decrease, CO_2 being basically perfectly incompatible, Fe^{3+} mildly incompatible.

2. They further say a few lines later that “oxygen fugacity from QFM-2 to 0 likely reflect changes in the oxidation states of both Fe and S”, whereas I would say that there is no appreciable S_4 at those conditions.

3. In their reply to my comment #8, they write that “sulphide minerals (e.g., FeS_x) within subducted lithologies can be oxidized to sulphate in situ via internal redox reactions, driven by the reduction of Fe^{3+} within the rock ($\text{S}_2 + 8\text{Fe}^{3+} = \text{S}_6 + 8\text{Fe}^{2+}$).” I only know of iron-reducing sulphate-producing bacteria. Where do you have $f\text{O}_2$ favouring appreciable S_6 over Fe^{3+} in subduction zones? I can envision that highly oxidised surficial material (e.g. sediments) releases oxidising fluids and S_6 upon subduction that are then reduced by the mantle (via oxidation primarily of Fe), but not that the reaction is shifted to the right. E.g. Tomkins and Evans 2015 EPSL model early anhydrite breakdown and later pyrite breakdown releasing H_2S , while Maffei et al. 2024 SciAdv suggest that pyrite might disproportionate to generate also sulphate, using modelling relevant to metacarbonate sediments.

4. Reply to my comment to L271-274: The authors “We are not sure whether the $\text{Fe}^{3+}/\Sigma\text{Fe}$ values in the cratonic eclogites mentioned here refer to the original magma composition or the post-alteration state ... it is worth noting that eclogites likely underwent dehydration. Assuming the altered rocks originally had $\text{Fe}^{3+}/\Sigma\text{Fe}$ ratios of 0.2–0.3 ... This implies that the RB remaining in exhumed eclogites may not reflect the total redox budget acquired during seafloor alteration prior to subduction.” Sure, but dehydration and subduction-related reduction of the redox budget would also apply to the ancient oceanic crust that the authors infer subducted and increased the mantle redox.

5. Reply to Reviewer 3 comment on L78-79: I did not point out that applying a constant pressure of 1 GPa to V/Sc ratios is invalid, but that the V/Sc-derived $f\text{O}_2$ must be corrected to a common pressure (we chose 1 GPa) in order to make valid comparisons between melts generated at different average pressures, because differences in $f\text{O}_2$ may arise from the “intrinsic” (at constant bulk Fe^{3+}/Fe) change in mantle $f\text{O}_2$ as a function of pressure alone. Our correction (assuming an increase of $f\text{O}_2$ by 0.4 orders of magnitude per GPa) was wrong because the complex redox profile of the mantle below 3 GPa was not known in 2016, but that’s a different issue.

6. Regarding a top-down oxidation of the mantle:

I am still not convinced that subduction of positive redox budget could have brought about the observed increase in mantle fO_2 , for the same reason I explained in the original review (also given by Reviewer 3). Here, the authors argue that “seafloor oxidized materials formed through oceanic hydration reactions subsequently accumulate in the oceanic lithosphere and remain isolated from the atmosphere by seawater”. However, the ocean-atmosphere-uppermost crust form part of the exosphere and in reservoir modelling, also those concerned with volatiles including oxygen, and are considered as an entity (e.g. Hirschmann 23 EPSL) that is interacting with the mantle on geological timescales. Taking redox budget out of the deep ocean water therefore means removing redox budget from the exosphere, thereby acting against oxygenation of the atmosphere. To then argue that “after global subduction (During Stage 4: ~3.2–1.8 billion years ago), through processes like arcs volcanic degassing, these subducted oxidized oceanic rocks are capable to deliver redox budget to the mantle and then to the atmosphere by volcanic degassing” illustrates that subduction of positive redox budget generates a short-circuit that diverts at least a fraction of it back to the atmosphere.

If the subduction mechanism worked, the question also arises as to why the mantle fO_2 has started to increase before 3.2 Ga (in the single-lid phase according to Fig. 5) and why it has not continually increased after the onset of global subduction and before the Proterozoic tectonic lull. Instead, the Fe^{3+}/Fe evolution ends several 100 Ma before the tectonic lull. I suggest this shows that after the transition to plate tectonics the exosphere and mantle are in some type of equilibrium on geologic scales, with the subducted redox budget being more or less balanced by release of redox budget via magmatism.

It should be noted that the increase in the mantle’s Fe^{3+}/Fe also carries petrologic consequences, as investigated by Asimow (2022 in Book: Magma Redox Geochemistry) with respect to the argument on the modern MORB Fe^{3+}/Fe : “Oxidizing conditions predict cold, low-MgO primary aggregate magmas that have difficulty crystallizing the most magnesian olivine phenocryst compositions found in MORB, that imply potential temperatures too cold to generate the traditionally assumed typical thickness of oceanic crust, and that are so close to the erupted basalt composition that it becomes difficult to explain the origin of a thick, cumulate lower crust as the complementary product of fractional crystallization”.

7. Regarding the inferred role of komatiite:

I am not sure that komatiitic oceanic crust could have subducted in the modern sense. Once metamorphosed, this lithology should convert to dense pyroxenite that is prone to delamination, as discussed by Foley et al 2003 Nature who also emphasised that continental crust production would have required partial melting of mafic, not ultramafic oceanic crust.

Regarding measured Fe^{3+}/Fe in komatiites, how do you distinguish oceanic, quasi syn-volcanic serpentinisation from post-emplacment (near)surface weathering-related serpentinisation?

Comments on the manuscript with tracked changes:

There are some formatting issues related to Fe^{3+}/Fe (superscript missing in L25, bold font in L33) – other instances later in text

L28 “especially when” – ambiguous, “given that” or “assuming that”

L37 “tectonic activity”, “tectonic reorganization events” – vague, perhaps “Earth’s evolving tectonic regime” and “a transition from single-lid to mobile-lid tectonics” or something more concrete as per your Fig. 5. I’m not sure about the abstract word count, but you could specifically mention the Proterozoic tectonic lull or refer to the relatively well-known “boring billion”

L58 “log units than QFM” should be “log units relative to the QFM”

L71 sentence needs rewriting. The garnet-bearing mantle extends down further than 8-10 GPa, but metal saturation buffers the mantle fO_2 at higher pressure. The word “where” could be misunderstood to imply that it is above 8-10 GPa that fO_2 decreases

L86-87 Birner et al. (ref 15) should be quoted here

I note that the f in fO_2 is not italicized consistently, other state variables (P , T) also not italicized

L94 fO_2 is not measured directly in any of the studies. It is related to Fe^{3+}/Fe and some thermodynamic equilibrium; not sure what you mean by “corrected”

L98 reference 18 is Canil et al. 1994 and is not recent. Ref 17 would be appropriate – please check all your references! [I later see that the references in the ms with tracked changes don’t agree with the clean ms so I am no longer checking]

L98-99 not sure what you mean by “using direct ratios... even greater discrepancies” – Ref 17 showed specifically the great impact of temperature (while also considering variations in source compositions, with minor effects on V-derived fO_2 estimates). The preceding lines already establish that other effects, such as pressure, were considered, so direct ratios were not used.

“under high-pressure conditions” – give the pressure, as high pressure means different things to different readers

L129 which source pressure is given here? There is no single source pressure...The average pressure of melt extraction?

L129 it is not the Archean melt that is extrapolated, but some parameter – formatting off in this and following sentence

L159 “under uppermost mantle conditions” – the mantle is vast and most carbon is not in the 4+ state...

L188 it would be better to list some attributes that you used to identify MORB-like basalts (e.g. REE patterns, Nb/La)

L196 “mutually” is redundant with “each other”

L213 and because its source and the melt extraction processes leading to MORB generation are relatively well-constrained

L276 what is “melt-extracted temperature”?

L301 unclear what you mean by “within the fewer samples”?

L303 what is “primitive fO_2 ”? That of un/little differentiated melts?

L333-335 this begs the question how the alternative assumption affects the result. Here, I am particularly interested in the effect of redox melting, occurring as shallow as 120 km for a more reduced Archean uppermost mantle source (Aulbach and Stagno 16, cited)? My feeling is that even at higher TP, the onset of partial melting in Archean ridges was (dominantly) shallower than 120 km, so that no complication from redox melting is expected.

L340 the reference to subarc depths comes a bit out of the blue, as up until here you seem mostly concerned with ridge settings and decompression melting

L342-344, L350 I strongly disagree. For the solid-melt system combined, redox melting has of course no consequence, but if you oxidise C to C4+, you induce redox melting, whereby CO₂ behaves like a highly incompatible component that is extracted. At the same time, you have a residue the Fe³⁺/ Fe of which was reduced to make C4+. Moreover, the remaining Fe³⁺ behaves like a moderately incompatible component. Combined, this implies a reduction of redox budget in the residue. However, as I note in the comment to L333, it may not be an issue for melting even of Archean ambient mantle.

L353-355 this is an odd way to end this section. Either delete, or give your estimate for the Archean mantle for comparison.

L388 “redox state”

L388-440 this section still fails to convince, there are too many unknowns and no rock record. Given the uncertainties regarding disproportionation, the comparison to Venus is also not warranted. This is really a topic for a different paper in my opinion and distracts from the main points in the present manuscript.

L457-460 what do you mean by “making it difficult for bridgmanite to crystallize” – obviously, the magma ocean does not still exist?

L474 it is not highly plausible for reasons I explain in point 6 above. At the very least, please moderate the language.

L490-491 clearly, alteration/weak metamorphism has had a major effect on the Fe³⁺/ Fe of komatiites, the compositions of which we can measure today, such that e.g. fluid-mobile elements are typically not used to constrain komatiite petrogenesis.

L529 typo

L1101-1102 syntax off – needs fixing (“indicating that”)

L1154 I would delete “historical” as this adjective is typically used in the context of human history. The word “trend” already implies the evolution

10 August 2025 Sonja Aulbach

Reviewer #2

(Remarks to the Author)

This is a much improved version of the original submission of this manuscript, and all of my previous comments have been addressed satisfactorily. I am happy to recommend this manuscript for publication in its current form.

Reviewer #3

(Remarks to the Author)

I have read the rebuttal letter and the revised manuscript carefully. My major concerns are: 1) A close check of Fig. S3b and Fig. S4c shows that the $Fe^{3+}/\Sigma Fe$ for both the peridotitic mantle and the basalt whole rocks are highly variable. I suspect the alleged "doubled $Fe^{3+}/\Sigma Fe$ " of the mantle by the authors. It shows that no clear secular evolution of $Fe^{3+}/\Sigma Fe$ of the mantle. 2) If exists, the explanations for the tortuous evolution of the mantle are mainly qualitative. No figures were used to explain the possible causal relation between the elevated mantle $Fe^{3+}/\Sigma Fe$ and tectonic parameters.

Minor points:

- 1) The authors used many vague nouns, such as "oxidized characteristics, L.44" and "redox characteristics, L752."
- 2) L.45 "the evolution trends of mantle that forms mid ocean ridge basalt" could be "the secular redox evolution of the MORB mantle"
- 3) L.47 what's the meaning of "exceeding 0.5 log units than QFM", do you mean "0.5 log unit increase in fO_2 "? the same for L.435
- 4) The last paragraph of the main text is really confusing, is the redox state an indicator or a driving force? In the first sentence, the authors claimed that "shifts in mantle redox state track major tectono-magmatic events", but in the last sentence, they said "this underscores the fundamental role of mantle redox evolution in driving the coupled ..."

Hope the above suggestions will be helpful to the authors.

Responses to Reviewers' Comments Point by Point

REVIEWER COMMENTS

Reviewer #1 (Remarks to the Author):

The authors developed a novel approach to extract information on mantle fO_2 through time, which combines numerical simulations and thermodynamic calculations using the latest models. This allows them to track $Fe^{3+}/Fe(T)$ in basalts as a function of pressure and temperature in the sub- and suprasolidus regions of peridotite phase relations and during decompression melting, which are ultimately convertible to fO_2 . I find this approach of forward modelling the most sophisticated yet, allowing to understand and quantify the complex relationships of these latter two parameters. In fact, the results are somewhat underexploited, as there are some interesting systematics (e.g. in Fig. 1A) that are not explored, though I understand that this is not the main thrust of the paper and there are limits to the length.

Answer: Thank you for your detailed and constructive comment. We greatly appreciate your recognition of the complexity and sophistication of our forward modeling approach, which indeed provides valuable insights into the intricate relationship between $Fe^{3+}/\Sigma Fe$ and fO_2 . The systematic trends you pointed out (e.g., as shown in Fig. 1A) are certainly worth further investigation, and we have expanded on this aspect within the space allowed by Nature Communications -- please see Lines 85-112 for details.

Overall, I think this will be of interest to a wide readership and would be publishable after moderate to major revisions. Nevertheless, my list of comments is very long, but if I see something that might be improvable, I say something. I hope the authors find these useful in order to craft a stronger manuscript or otherwise can easily rebut them.

Answer: We would like to once again express our sincere gratitude for your detailed and insightful suggestions. They have been extremely helpful in improving our manuscript, particularly in clarifying some previously ambiguous sections. We have carefully addressed all of your comments and made corresponding revisions. Please find our detailed responses below.

Main concerns and suggestions:

1. I can see that the authors extract $Fe^{3+}/Fe(T)$ of melts from the numerical experiments in Fig. 2F-J, and that the thermodynamic phase equilibria modelling can back-extract $Fe^{3+}/Fe(T)$ from the V/Ti-based fO_2 of natural

samples. Maybe I'm being thick, but can the authors better expose how the numerical experiments link to the natural samples? In general, the modelling, in particular the numerical one, could be described in more detail. Is it correct that the authors used forward thermodynamic modelling using PT of melting of natural samples to find the associated $\text{Fe}^{3+}/\text{Fe(T)}$, from which they then extract $f\text{O}_2$. That is, their $f\text{O}_2$ estimate for the natural samples is completely unrelated to the V-based $f\text{O}_2$ estimate? The onus is on the authors to ensure their manuscript becomes accessible to a wide and non-specialised readership.

Answer: In the natural sample workflow, $f\text{O}_2$ is estimated using V-based oxybarometers (e.g., V/Ti or V/Sc ratios; Fig. 3). In contrast, the numerical model (Fig. 2) derives melt $\text{Fe}^{3+}/\Sigma\text{Fe}$, $f\text{O}_2$, and other redox parameters solely from known P-T conditions and initial mantle compositions, without the need for V-based constraints.

The numerical experiments presented in Figure 2 are fundamentally independent of the calculations based on natural samples; however, they serve as mutual validation of each other. The modeling results in the figure are derived from forward simulations based on prescribed geodynamic conditions and rock compositions within a coupled thermomechanical-thermodynamic framework. These simulations do not require V-based oxybarometers to obtain the $\text{Fe}^{3+}/\Sigma\text{Fe}$ ratio in basaltic melts.

In contrast, the calculations for natural samples are inversions. When pressure, temperature, $f\text{O}_2$, and bulk composition are known, thermodynamic modeling can be used to back-calculate the $\text{Fe}^{3+}/\Sigma\text{Fe}$ ratio at the time of melt formation, which can in turn be used to estimate the $\text{Fe}^{3+}/\Sigma\text{Fe}$ ratio of the mantle source. Therefore, the numerical and natural-sample-based approaches are conceptually and methodologically independent. The only commonality is their reliance on thermodynamic calculations.

We have expanded the relevant sections of the manuscript to better clarify these distinctions and to improve accessibility for a broader, non-specialist readership (e.g., Lines 173-176; 228-233).

2. Related, the authors claim that their approach yields the most accurate and robust insights yet, but ultimately, constraints on mantle redox evolution rest with the natural samples and their shortcomings: (a) unclear tectonic setting (in fact, unlikely MOR-setting for the vast majority of Archaean basalts (Kamber 2015 Precambr Res; Puchtel and Arndt 2025 TOG; Brown+24 JGeolSoc) – this is important because we know that mantle $f\text{O}_2$ varies as a function of tectonic setting (Cottrell+22 in Geophys Monogr 266); (b) uneven sampling through time, with some sparsely sampled periods; (c) uncertainties related to uncertainties in PT estimates from natural samples. Thus, the information from the modelling, however sophisticated, cannot be more accurate and precise than that derived from natural samples.

Answer: We agree that this section may benefit from clearer wording and it has been done. Our original intent was to emphasize that oxygen fugacity is inherently complex -- it reflects redox conditions in P-T-X space and is often relative in nature. To address this, we transform fO_2 into a redox budget (expressed as $Fe^{3+}/\Sigma Fe$), which is governed by mass and charge balance and therefore independent of pressure, temperature, and composition. This transformation enables a broader audience to understand:

How varying mantle compositions, along with their geodynamic and thermodynamic contexts, quantitatively control the redox evolution of melts (as shown in Figure 2); How we can better quantify the oxidation state of the mantle in a thermodynamically consistent way. The integration of both numerical modeling and natural sample analysis is thus essential for quantitatively constraining the redox budget of the mantle over time. This dual approach marks a methodological advancement over previous studies.

We fully acknowledge the limitations of natural samples:

(1) Uncertain tectonic settings- We agree that the majority of Archean basalts may not be from MOR-settings. This is precisely why, among the hundreds of thousands of entries in the database, only a few dozen are usable. We are well aware of this issue and have taken steps to address it. In particular, we have applied multiple geochemical screening methods to identify samples most analogous to modern MORBs. Trace element patterns in these samples closely resemble those of present-day MORB. In the revised version, we have followed the suggestions from you and other reviewers, such as using a threshold of $(Nb/La)_{PM} \geq 1$ to minimize crustal contamination. In addition, we included new calculations using MORB-derived metamorphic rocks (from Aulbach and Stagno, 2016) and their mantle source regions to reinforce and validate our previous interpretations.

(2) Uneven temporal distribution-this is a well-recognized challenge not only in our study but across nearly all research dealing with long-term geochemical evolution (e.g., Herzberg et al. 2010; Zhang et al., 2024; Aulbach and Stagno, 2016; Aulbach and Arnd 2019). The particularly sparse sampling in the early Archean and Mesoproterozoic may reflect a global reduction in tectonic activity. While the number of preserved samples is beyond our control, we do not believe this justifies halting efforts to understand early Earth processes. Fortunately, the available datasets from both the whole Archean and modern times are sufficiently abundant to support the key conclusions of our study. We have also noted in the manuscript that these trends should be approached with caution. In addition, we conducted two sets of sensitivity tests on the relationship between time and mantle $Fe^{3+}/\Sigma Fe$, and the results indicate that our conclusions are robust (see our response to Reviewer 2' Comment No. 5 for details).

(3) Uncertainty in P-T conditions of natural samples-we have taken deliberate steps to minimize related

errors. For example, we focus on basalts instead of komatiites, which are associated with greater uncertainties. When estimating P-T conditions, $\text{Fe}^{3+}/\Sigma\text{Fe}$ itself becomes a key variable influencing the outcome. To resolve this, we applied an iterative bracketing strategy (e.g., bracketing criterion): we start by assigning an initial $\text{Fe}^{3+}/\Sigma\text{Fe}$ within a reasonable range, use this to calculate P-T, derive $f\text{O}_2$ and a new $\text{Fe}^{3+}/\Sigma\text{Fe}$, then repeat the process until convergence is achieved-i.e., until both P-T and $\text{Fe}^{3+}/\Sigma\text{Fe}$ stabilize. This method ensures reliable estimates of the thermodynamic conditions.

Finally, we completely agree that no matter how sophisticated the modeling is, its accuracy and precision can never exceed the empirical constraints provided by natural samples. This is precisely why we adopted a dual approach: to cross-validate our findings. Numerical forward modeling is not constrained by empirical records, allowing us to explore how thermodynamic and thermomechanical conditions influence the $\text{Fe}^{3+}/\Sigma\text{Fe}$ ratio in the mantle. In contrast, natural samples provide direct geological records. We firmly believe that modeling is the extension of observation, and observation is the foundation of modeling.

3. There are quite a number of vague, sweeping or inaccurate statements, and some ad hoc statements, throughout the main part of the text that are not very informative and therefore leave more questions than answers. Instances are given below.

Answer: Thank you for pointing that out. Due to word count limitations in the initial draft, many parts were not explained as clearly as we would have liked. However, within the word limit allowed by Nature Communications, we have now expanded these sections to help readers better understand our logic and content.

4. The authors variably refer to redox state, oxygen fugacity, oxidation degree, oxidation budget, $\text{Fe}^{3+}/\text{Fe(T)}$ The mantle redox state reflects the sum of several multivalent elements, of which iron is certainly overall the most important one, but S, C and H are not negligible (Evans 2006 Geology). I suggest to make sure $f\text{O}_2$, redox state (or redox budget), and $\text{Fe}^{3+}/\text{Fe(T)}$ are accurately used throughout the text, and to refrain from using additional terms (e.g. oxidation degree, oxidation budget etc).

Answer: Overall, what we refer to as the “redox state” essentially corresponds to the redox budget (Evans, 2006), which we simply express using the $\text{Fe}^{3+}/\Sigma\text{Fe}$ ratio. As you rightly point out, in addition to iron, other multivalent elements such as C, H, and S also contribute to the mantle’s redox state at MORB formation depths. However, their concentrations are generally low, and in typical MORB-source mantle, their valence states are often assumed to be relatively constant. For example, sulfur is mainly present as S^{2-} , hydrogen as H^+ , and carbon

as C^{4+} under these conditions (Evans 2006, 2012). By the way, although at greater depths, carbon may exist as C^0 . Upon ascent, the transformation of C^0 to C^{4+} would consume Fe^{3+} (i.e., cause reduction), but the overall redox budget remains conserved in MORB mantle source.

Thus, by assuming fixed valence states for C-H-S species under MORB-source conditions, changes in the $Fe^{3+}/\Sigma Fe$ ratio can be used as a reliable proxy for changes in the mantle redox state. Furthermore, variations in oxygen fugacity from QFM-2 to 0 likely reflect changes in the oxidation states of both Fe and S, but not the C, especially under shallow mantle conditions as C^{4+} can be stable down to QFM-1.5 to QFM-2 at 2-3 GPa (Maffei et al., 2024; Walters et al., 2020).

Therefore, we accept your suggestion and will standardize our terminology throughout the manuscript, limiting ourselves to terms such as redox state, redox budget, and oxygen fugacity, and avoid less well-defined expressions like “oxidation degree” or “oxidation budget”. We also discussed the influence of other redox-sensitive elements (such as carbon and sulfur); please see Lines 118-132; 260-276.

5. Given continued uncertainties regarding earliest Earth processes and the physicochemical-dynamic conditions attached to them, the authors are of course entitled to their preferred interpretation of why the fO_2 of Earth’s mantle changed after the Hadean. However, the arguments against upward mixing of redox budget arising from disproportionated iron, and the arguments in favour of komatiite serpentinisation followed by subduction are substantially less clear-cut than presented, for reasons I detail below. This discussion could be a lot more balanced.

Answer: We have provided a more balanced discussion on this part, including the possibilities and weaknesses of both scenarios, and a more detailed explanation of the calculation process. Please see Lines 353-443 and our response below.

6. Furthermore, the authors take evidence from short-lived radiogenic isotopes to argue for separate upper and lower mantle reservoirs, but this seems too simplistic, as early-generated heterogeneities survive to the present day (as sampled by OIBs) and are thought to reside in geographically restricted regions that are not identical to the lower mantle as a whole.

Answer: Some of the more uncertain discussions have been removed, as they are not directly relevant to the main content. A detailed response to this issue is provided below (see comments on Lines 246–249); here, we briefly summarize the key points:

We agree that the mantle should not be oversimplified into distinct “upper” and “lower” compartments. A more accurate view is that mantle reservoirs exhibiting different short-lived isotope signatures have largely mixed into an approximately homogeneous state over prolonged geological timescales. That said, we fully acknowledge that the mantle is not perfectly homogeneous even today and retains localized anomalous domains, such as those sampled by ocean island basalts (OIBs).

Recent studies combining W–Ru isotopic systematics (Fischer-Gödde et al., 2020; Messling et al., 2025) provide strong support for the incorporation of late accreted material, as different mantle reservoirs and processes carry distinct isotopic fingerprints:

1. Core interaction (sampled by modern OIBs): negative ^{182}W and positive ^{100}Ru
2. Late accreted material: negative ^{182}W and negative ^{100}Ru
3. Archean mantle: positive ^{182}W and positive ^{100}Ru

The transition from the positive isotopic signature of the Archean mantle to the near-zero values observed in the modern mantle cannot be explained by ongoing core–mantle interaction alone. Instead, it necessitates the addition and mixing of late accreted material. This indicates that core–mantle interaction, as inferred from OIB compositions, is unlikely to be the primary control on Archean mantle isotopic characteristics.

Regarding mantle mixing, the progressive decline in mantle ^{182}W after the Archean still requires a reservoir with negative ^{182}W that was not sampled during the Archean—most likely residing in the deep mantle. However, the detailed mixing mechanisms remain difficult to constrain, due to the lack of information on the scale and distribution of this negative ^{182}W reservoir, as well as sparse data coverage in the ^{182}W record—particularly after ~2.5 Ga.

For these reasons, we have retained the discussion of late accretion in the manuscript, as it is consistent with both W–Ru isotopic system constraints and the oxidation state inferred in this study. However, we have removed the broader discussion of time-dependent mantle mixing due to the greater associated uncertainties.

Below is Figure 3a from Messling et al. (2025), illustrating the effects of late accretion and core–mantle

interaction:

7. The authors dismiss komatiites as “enigmatic”. This is a misrepresentation of the field, as komatiites have extremely well-constrained petrogeneses, which may be complicated by their derivation from thermochemically anomalous plumes, and it is true that their pressures of formation are difficult to estimate. I would accept this as an argument in favour of focusing on spreading ridge-derived samples (MORB-like basalts and picrites). By the same token, the authors should acknowledge that few if any Archaean basalts are truly spreading ridge-derived.

Answer: The related statements have been removed. We now only retain the point that the formation pressures of komatiites are indeed difficult to constrain, which may result in larger uncertainties in the derived results. Additionally, we have included a discussion acknowledging that few Archaean basalts are truly derived from spreading ridge environments.

8. The authors assume a modern subduction redox budget to estimate whether this could have oxidised the mantle to present-day levels. However, we know that the atmosphere was oxygenated to just 1% PAL as late as 2.4 Ga ago, and that oceanic bottom waters became oxygenated only in the Neoproterozoic (Stolper+Keller 18 Nature), before which time there would have been very little dissolved sulphate, which is the most potent oxidant (Tomkins+Evans 15 EPSL). The subducted redox budget surely must have been variable through time. Or maybe I misunderstood what the authors are aiming at. It also seems that they address this in some detail in a paper under consideration (ref. 38), this should be clarified.

Answer: The redox budget we used in our calculations represents a very conservative estimate. It is

constrained by petrological records, ocean drilling data, and seismic observations since ~3 Ga. Specifically, we did not include the highly oxidized lithologies unique to the Phanerozoic, such as those with $\text{Fe}^{3+}/\Sigma\text{Fe}$ ratios exceeding 0.5 even up to 0.8 as reported in Stolper and Keller (2018). Instead, we based our calculations on altered rocks with $\text{Fe}^{3+}/\Sigma\text{Fe}$ values of ~0.20 for serpentinites, ~0.21 for gabbros, and ~0.26 for basalts -- values consistent with observations from the Precambrian (Fig. a below; Precambrian data from Gard et al., 2019, Stolper and Keller 2018).

It is worth noting that we also did not account for the additional redox budget carried by highly oxidized lithologies such as serpentinized komatiitic rocks (Fig. b below; data from Tamblyn and Hermann 2023) or banded iron formations (BIFs). Therefore, we consider our calculation to be conservative.

These moderately oxidized lithologies are sufficient to carry a redox budget capable of oxidizing the mantle to present-day levels. It is also important to emphasize that none of the rocks we considered contain oxidants like seawater-derived sulphate, which are characteristic of the Phanerozoic (Tomkins and Evans 2015). In our model, sulphide minerals (e.g., FeS_x) within subducted lithologies can be oxidized to sulphate in situ via internal

redox reactions, driven by the reduction of Fe^{3+} within the rock ($\text{S}^{2-} + 8\text{Fe}^{3+} = \text{S}^{6+} + 8\text{Fe}^{2+}$). Part of S^{6+} goes into fluids (Duan et al., 2024). The released oxidized fluids then contribute to the oxidation of the mantle wedge, while the remaining Fe^{3+} is transported into the deep mantle, incrementally raising its redox budget.

It is also worth noting that the redox fluxes we report represent net fluxes, after accounting for the oxygen sinks associated with MORB, arc, and OIB melting.

Our goal in presenting these calculations is to provide a simulation- and observation-based perspective on the long-standing debate over whether subduction can oxidize the mantle to its present-day state. We have expanded the relevant discussion in the revised manuscript to clarify this reasoning.

9. There are some sentences with awkward English that would benefit from polishing.

Answer: We have carefully revised the manuscript to improve clarity and fluency throughout. Sentences with awkward or unclear phrasing have been reworded to ensure smoother and more precise English expression.

10. The figures and their captions can be much improved. There is too much, in part easily overlooked, small font in the panels themselves. The captions are so terse as to be uninformative. IMO, readers should be able to understand what the diagrams show and what arguments they support without reading through the entire manuscript.

Answer: We have revised the figures and their captions accordingly. Specifically, we have: Increased the font size within figure panels to improve readability; Simplified and clarified visual elements to reduce clutter; Expanded the captions to provide sufficient context, including clear descriptions of what each figure shows and how it supports the main arguments of the manuscript. Our goal is to ensure that readers can grasp the key messages of the figures independently, without needing to refer back to the main text.

11. Throughout, I suggest to use italics for state variables, such as T , P , f (fugacity) etc.

Answer: We have carefully revised the manuscript to ensure that all state variables such as T , P , and f (fugacity) are consistently formatted in italics throughout the text.

In-line comments and suggestions

Title: what is oxidation degree? $f\text{O}_2$? Redox budget? $\text{Fe}^{3+}/\text{Fe}(T)$? Please be more precise, also in the abstract and elsewhere

Answer: We have revised that as suggested to use $\text{Fe}^{3+}/\Sigma\text{Fe}$, and have provided a detailed explanation in both

the abstract and the main text. The oxidation degree we refer to represents changes in the redox budget of the system. In other words, it reflects shifts in the distribution of elements among different valence states (e.g., Fe²⁺ vs. Fe³⁺), see also Lines 119-133; 260-266.

11-13 It is clear what the authors want to express in the first sentence but the wording is inaccurate. What is oxygen-rich habitability? The atmosphere is now O₂-rich. What the mantle regulates is the exchange of redox budget, which ultimately enabled O₂ to accumulate in the atmosphere, thereby affecting how life evolved

Answer: We have rewritten the sentence.

13 our planet should be capitalised, also should be "Earth's"

Answer: Yes, has been changed.

14 remains a subject of

Answer: Has been revised.

16 they are not all basalts, but also picrites according to the supp table. As an aside, the mere fact that there are so many basalts implies that Archaean ambient mantle T_p could not have been very hot. If the samples are basalts due to extensive differentiation, then they are not appropriate to estimate mantle fO₂ from them.

Answer: The basaltic melt here includes both traditional basalt and picritic basalt. In the simulation, the composition depends on the temperature and pressure conditions. In the initial version, we did not specifically distinguish picritic melts. It has already been indicated in the relevant parts of the table and text.

We have conducted a series of mantle potential temperature (T_p) experiments that cover both lower T_p values (1350-1400 °C) and higher T_p values (1500-1600 °C), encompassing the range predicted by both Herzberg et al. (2010; 1500-1600 °C) and Aulbach and Arnd (2019; 1500°C). Notably, our numerical models show that even under very high T_p conditions (up to 1600 °C), Archaean-style lithospheric extension, basaltic rocks generation, and subduction can still occur. This may support the plausibility of MORB-like rock formation under high T_p in the Archaean. We agree that basalts produced through extensive differentiation are not suitable for estimating mantle fO₂. Therefore, we have made efforts to focus our study on basalts that are as close to primary melts as possible (see Methods).

16-17 when you are referring to a ridge setting specifically, there is no need to again mention "geodynamic settings". Perhaps "modern conditions"

Answer: Yes, has been revised.

20 demonstrate

Answer: Yes, has been revised.

20 clearly, measured $\text{Fe}^{3+}/\text{Fe(T)}$ are not reliable and if you estimated the values based on V/Sc or V/Ti, then your result cannot be more reliable than the information contained in the element ratios? Besides, Zhang et al. (2024, cited) already presented whole-rock $\text{Fe}^{3+}/\text{Fe(T)}$ thermodynamics-based estimates. If yours are significantly improved, or novel in the sense that you used a different approach, then this should be expressed differently – this relates to my main comment above

Answer: Yes, we agree that the measurements are not entirely accurate. We also did not use the measured ferric iron (Fe^{3+}) values. However, the thermodynamic approach we refer to here involves cross-validation between forward numerical modeling and inverse estimation of Fe^{3+} proportions based on recorded P-T-V/Ti conditions. Our method incorporates internal thermodynamic consistency as well as geodynamic context, representing the first such attempt in related research. Although Zhang et al. also inferred $\text{Fe}^{3+}/\Sigma\text{Fe}$ ratios in rocks, they primarily relied on the Kress and Carmichael (1991, K91) model, which tends to overestimate ferric iron content. Moreover, their correction for oxygen fugacity appears to be flawed, likely introducing additional estimation errors. Moreover, Zhang et al. (2024) did not calculate the ferric iron (Fe^{3+}) proportion in the mantle. We have revised the phrasing to better highlight the novelty and uniqueness of our approach.

24 please be more specific here: what kind of geological evidence? Below you also refer to biological evolution.

Answer: We have revised the sentence to clarify that the observed changes in molar volume are broadly coupled with variations in the redox state of the mantle.

25-26 tectonic reorganisation - ok. But what do you mean by geological reorganisation?

Answer: Perhaps the wording was inappropriate; it has been revised.

27 I am aware that habitability is a catchword, but “oxygen-rich habitability” just is not expressed well, see comment above

Answer: Has been revised. But please see our response to “Line 240 how do you define habitability?” for more detail.

27 I understand there is a word count limit, but this reference to biological evolution is too vague

Answer: Has been revised.

37-38 exert... influence, or simply “have... effects”. Effects are not exerted, I think.

Answer: Has been revised.

ref. 6 was updated in Aulbach+19 SciRep - and using both V/Sc and $\text{Fe}^{3+}/\text{Fe(T)}$! - so you could cite this instead

Answer: Has been cited.

here and throughout: QFM units don't exist. It is just a reference buffer. fO_2 was lower by ... orders of magnitude (or log units, if you will) – throughout manuscript

Answer: Has been revised.

“tends”. Also, please be more specific as we know it is only the garnet-bearing mantle down to metal saturation at ~8-10 GPa where fO_2 decreases with increasing pressure. At $P < 3-3.5$ GPa, the fO_2 -depth profile is more complex, as demonstrated from phase equilibria and empirical models (Stolper+20 AmMiner; Birner+24 Nature)

Answer: Has been revised.

58-61 This sentence does not reflect the state of the art:

1. Aulbach+Stagno 16 (cited) did correct for the fO_2 decrease with increasing pressure and did find significant differences between Archaean and modern samples.
2. Zhang+24 (cited) wrongly corrected their basalt V/Ti-derived fO_2 for the reduction of the melt with decreasing pressure after leaving the source. However, the V/Ti-derived fO_2 is already that of the source.
3. Both biased their Archaean, more deeply-derived samples (due to higher TP) to too-high fO_2 by assuming the garnet-peridotite fO_2 -pressure trend could be extrapolated to 1 GPa, but we know since Stolper+20 that this is not the case.

Answer: Yes, perhaps the meaning was not clearly conveyed. Here, we intended to say that the use of direct elemental ratios follows the approach used in reference 12. We have tried to expand the sentence to more accurately reflect the current state of research. Has been revised. See Lines 59-74.

these are “elemental ratios”. Furthermore, this statement is blatantly incorrect: neither ref. (6) nor refs. (11,13,14) used direct geochemical ratios, as (6) accounted for P (T effect not known at the time), (14) for T and (11) accounted for PTX (though made a mistake), and (13) accounted for TX. It is true only for (12) that V/Sc was used directly. (16) should not be cited here - it's a review paper and we were well aware of PTX effects

Answer: Yes, perhaps the meaning was not clearly conveyed. Here, we intended to say that the use of direct elemental ratios follows the approach used in reference 12. We have tried to expand the sentence to more accurately reflect the current state of research.

“the” depleted mantle is understood by many to be the MORB source. What your fig. 1A seems to show is mantle that is residual from extraction of various extents of melting in the suprasolidus region as a function of distance from the solidus (we later learn in the methods that the bulk composition of the model corresponds to a depleted MORB mantle)

Answer: Yes, the initial bulk composition corresponds to a depleted MORB mantle, not the residual material itself. This is consistent with the view held by many that the source of MORB-like basalts is a depleted mantle, and this is also reflected in the numerical simulations shown in the supplementary figures. In the revised version, we have also included the bulk composition of the primitive mantle. We demonstrate that the transition in whole-rock composition from the primitive mantle to a depleted mantle does not affect the simulation results (see answer and figure for Reviewer 3 comments “In addition, if the mantle was remoulded in $\text{Fe}^{3+}/\Sigma\text{Fe}$ …”).

You show the mantle at constant $\text{Fe}^{3+}/\text{Fe}(\text{T})$, so what do you mean by “often”? where in the figure is it coupled vs. decoupled?

Answer: This is described in detail in the manuscript (please see lines 85-112).

69-70 This is not shown in Fig. 1? Qualitatively, if Fe^{3+} behaves like a moderately incompatible element (O'Neill+18, cited), one would expect the ratio to decrease with increasing melt F, which is a function of T, all other things being equal. This seems to be shown in Fig. 2.

Answer: It is shown -- the decreasing trend of $\text{Fe}^{3+}/\Sigma\text{Fe}$ is indicated by the different colors. In a closed system, the decrease in $\text{Fe}^{3+}/\Sigma\text{Fe}$ is correlated with the degree of partial melting (F). However, in an open system, due to batch melt extraction, the solidus of the residual mantle shifts toward higher temperatures. When the residue undergoes subsequent melting, F starts from zero again, but the $\text{Fe}^{3+}/\Sigma\text{Fe}$ in the melt at a given temperature remains unaffected. Therefore, we would argue that $\text{Fe}^{3+}/\Sigma\text{Fe}$ is more a function of temperature or total cumulative melting, rather than the melt fraction (F) after batch extraction in an open system.

not just depleted mantle, but any mantle, or more specifically, peridotite

Answer: Yes, it has been revised.

perhaps refer redox budget sensu Evans (2006) and then stick with it

Answer: It has been revised and added.

72-76 I think you should refer to the phase eq. and empirical modelling of Stolper+20 (Am Miner) and Birner+24 (cited) here. Stolper highlighted how this decoupling is related to changing mineralogy across peridotite facies as a function of pressure

Answer: Yes, it has already been cited.

please don't refer to QFM units. Refer to $f\text{O}_2$ and several orders of magnitude (or log units)

Answer: Yes, it has been changed.

I think “historical” is - historically - used for time since humans keep records. Perhaps “trends over geological time” or similar

Answer: Yes, it has been changed.

82-84 I don't see how this is more accurate? Melt fO_2 also depends on PTX, and unlike elemental ratios, $Fe^{3+}/Fe(T)$ can be changed by redox interactions (due to redox melting, degassing etc). Besides, measured $Fe^{3+}/Fe(T)$ of ancient rocks are unreliable due to alteration. Therefore, ultimately, your $Fe^{3+}/Fe(T)$ modelling is novel and possibly more accurate, but it remains a model that, to be useful, you have to compare against a rock record

Answer: We reworded that. What we meant by saying our method is “more accurate” does not imply that existing fO_2 estimates are inherently inaccurate. Rather, the issue lies in the complexity of calibrating data across samples from different geological times, which can introduce significant uncertainties. For example, this has been discussed in the work of Zhang et al. Such calibration challenges may lead to the misleading impression that the redox state of the mantle has remained unchanged over time.

In our model, $Fe^{3+}/\Sigma Fe$ directly reflects the mantle redox budget. This is because, under the P-T conditions corresponding to MORB-like source regions, the reference state of C-S-H is well-constrained. As a result, in our simplified modeling framework, the $Fe^{3+}/\Sigma Fe$ of the melt at the time of generation is not influenced by complex external thermodynamic variables, nor does it require complicated calibration procedures.

We fully agree that measured $Fe^{3+}/\Sigma Fe$ in ancient rocks is often unreliable due to alteration. $Fe^{3+}/\Sigma Fe$ can indeed be affected by redox processes such as redox melting and degassing. However, these processes are not included in our modeling approach. As you know, the $Fe^{3+}/\Sigma Fe$ values in petrological databases are also inferred from elemental ratios and P-T conditions, rather than measured directly, and thus are likewise not affected by alteration or degassing.

Therefore, our aim is to define a redox state that is simple, intuitive, and constant, which helps make comparisons of redox evolution across geological time more accessible to a broader audience.

even if this is explained as part of the Methods later, this reference to “thermodynamic and geochemical methods” is much too vague. A few more sentences are needed here, what these methods are, so we understand broadly what you did without having to consult the Methods

Answer: Yes, we have added some sentences, please see Lines 136-147.

yes, the modelling is supercool, novel and a major advance. But this does not make observations from the sparse rock record itself any more robust

Answer: Yes, the rock record is what it is -- we cannot make it more precise. However, it is worth noting that our calculations on rock samples incorporate a relatively novel thermodynamic approach, which brings the

results closer to the actual record -- something that had not been achieved in previous studies. Moreover, earlier empirical methods for estimating ferric iron content often involve substantial uncertainties. See discussion for details.

please be precise regarding what it is that has doubled. I think you mean $\text{Fe}^{3+}/\text{Fe(T)}$, which may be the most abundant multivalent element, but does not alone fix the redox budget (see Evans 2006 Geology)

Answer: This point warrants further clarification: in our study, the $\text{Fe}^{3+}/\Sigma\text{Fe}$ ratio effectively represents the redox budget. This is because other redox-sensitive elements are either present in very low abundances or have reference states that are unlikely to change significantly in a typical depleted mantle setting. For example, sulfur generally exists as S^{2-} , hydrogen as H^+ , and carbon -- with an abundance of only about 50 ppm (Evans, 2006, 2012) -- occurs predominantly as CO_2 < ~2-3 GPa, or potentially as elemental carbon at greater depths.

Importantly, as deep-sourced materials ascend, any carbon that transforms into CO_2 draws its redox budget from $\text{Fe}^{3+}/\Sigma\text{Fe}$, ultimately yielding a $\text{Fe}^{3+}/\Sigma\text{Fe}$ ratio consistent with that of the shallow mantle that is the main focus of this study. Therefore, based on the definitions and reference states proposed by Evans (2012), $\text{Fe}^{3+}/\Sigma\text{Fe}$ can be used as a reliable proxy for tracking redox state variations in all our calculations. We will add a few sentences to the manuscript to clarify this relationship.

It is not the redox modeling that is modern or Archaean, but Redox modeling for modern and Archaean conditions (what you call thermodynamic conditions)

Answer: Yes, it has been changed.

98-99 This needs much better justification. You could say that komatiites are plume-related and therefore may sample thermochemically anomalous sources, whereas basalts may sample an ambient mantle source representing a better-constrained, less variable system. You should also be honest and acknowledge in your manuscript that virtually no Archaean basalts are spreading ridge-derived (Puchtel and Arndt 2025 TOG; Brown+24 JGeolSoc). The samples you describe as "MORB-like" are actually basalts associated with melting in intraplate settings and erupted through attenuated continental mantle. The only true Archaean spreading-ridge derived material is now sampled as cratonic eclogite (no need to cite, but see Aulbach+Smart 23 AREPS if you want to know why).

Answer: Regarding komatiites, we have decided to accept your suggestion -- we will remove the lengthy and potentially inaccurate discussion, and retain only the explanation regarding the large P-T uncertainties.

As for basalts, we are well aware that identifying definitively representative MORB samples is indeed challenging, and we also acknowledge the associated debates, as you pointed out. This is precisely why we use

the term "MORB-like" similar to how some other studies refer to "no-arc" samples.

However, what we can confidently confirm is that these samples do not originate from subduction settings or from sources similar to the modern enriched mantle. This distinction does not affect the validity of using them to assess the mantle redox state.

We have added a few sentences to clarify this point.

there is no representative TP for the Archaen period, as TP estimates vary by >200 oC (e.g. Herzberg+2010, cited vs. Aulbach+Arndt 19 EPSL)

Answer: We have carefully reviewed both papers and found that our simulations cover the range and average values of Archean samples reported in the two studies. Collectively, the Herzberg et al. (2010) suggests that the Archean mantle potential temperature (Tp) may have been around 1500-1600 °C, while Aulbach and Arndt (2019) suggests a warm Tp of 1500°C. We have added a sentence addressing the differences in these estimates accordingly.

a totally expected result for all tectonic settings where decompression melting occurs...

Answer: This is merely a descriptive statement.

Again, given what we know, this is a trivial finding. Rather than "highlights", perhaps it validates your modelling approach?

Answer: We appreciate the comment and agree with your assessment. The text has been revised to clarify that the result serves to validate our model, rather than to highlight a new finding.

please mind language - what is melting T? T of solidus intersection? It is afterwards, when the melt takes with it heat, that cooling occurs. But this is not specific to MOR ridges, so I am not sure what you mean

Answer: Has been done. It is melting extracted temperature.

please explain here the presence of cold drips and the mixing, or refer to a text/Methods where we can read up on it - is this related to a pre-plate tectonic regime?

Answer: Has been done. Please see also Fig. S6 and related refs.

122-123 not sure this sentence makes sense as written? What is an intricate "internal thermodynamic equilibrium"? Why internal? Aren't the physical (not geophysical, presumably?) parameters (P-T-fO₂) dictating the equilibrium? Aren't the thermodynamic calculations based on experimental petrology, first and foremost?

Answer: This reflects the thermodynamic equilibrium among complex mineral solid solution models, which is a key strength of our approach, as it captures realistic mineral behavior under varying P-T-fO₂ conditions.

imposed rather than provided?

Answer: Have revised.

“MORB with fO_2 near QFM”

Answer: Have revised.

You cite Hirschmann 2023 in a different context, but you should acknowledge here - and thereby support your choice of $Fe^{3+}/Fe(T)$ of 0.02 - that he suggested that this is the maximum ratio in case there was a mantle redox evolution since the Archaean

Answer: Yes, we have already cited this reference here.

Please see Fig. 2b in Aulbach+19 SciRep - this is roughly similar to the ratios we reconstructed for Archaean eclogite xenoliths interpreted as metamorphosed subducted MOR-derived oceanic crust

Answer: We have cited this paper at this point.

140-141 There is no rationale provided here why it is increased T_p rather than an intrinsically lower fO_2 “promoted reduction of basaltic melts”? Maybe a few steps in the chain of arguments are missing here? (higher T_p = deeper melting = lower fO_2 in the garnet-bearing mantle source = lower fO_2 in the melt which separated and reduced)

Answer: We have expanded the relevant explanation. However, our results show that the effect is largely temperature-dependent -- even at equal extraction pressures, similar $Fe^{3+}/\Sigma Fe$ ratios are observed.

A doubled $Fe^{3+}/Fe(T)$ does not equate doubly oxidised - please be more exact

Answer: We have revised the sub-title. Strictly speaking, the mantle redox state should be represented by the combined contributions of Fe^{3+} , C^{4+} , S^{6+} , and H^+ . However, as discussed earlier, the valence states of C-O-H components remain largely unchanged in the normal, shallow, depleted MORB mantle, and their overall concentrations are very low. Therefore, we simplify the redox state representation to $Fe^{3+}/\Sigma Fe$. We have provided a more detailed explanation in the manuscript regarding why a doubling of the ferric iron ratio can be interpreted as indicating at least a double increase in oxidation of the MORB mantle source. Please see Lines 119-133; 260-276.

144-146 Keeping your broad readership in mind, perhaps it would be better to say that a higher T_p results in a lower fO_2 via its effect on the pressure of melting (which is deeper), at least in the garnet peridotite facies - at least this is what you suggest? Also, we’ve known this before, so this is not a new finding from the present study. This is why AS16 and Zhang+24 applied a P correction (though both wrongly extrapolating the garnet peridotite fO_2 -P profile to 1 GPa). AS16 actually estimated P of melting from their preferred T_p evolution curve, so the link between T_p -P- fO_2 is clear. Finally, it is necessary to be very careful with the terminology: here you refer

to a lower mantle redox state despite constant $\text{Fe}^{3+}/\text{Fe(T)}$ - I think you mean $f\text{O}_2$

Answer: Yes, we have revised the text. However, what we intended to convey here is not the oxygen fugacity ($f\text{O}_2$) per se, but rather the Fe^{3+} proportion in the melt. As shown in Fig. 1a, this is primarily controlled by temperature, not pressure -- even at the same pressure, higher temperatures result in lower $\text{Fe}^{3+}/\Sigma\text{Fe}$ ratios in the melt.

When we referred to a "constant $\text{Fe}^{3+}/\Sigma\text{Fe}$," we meant in the initial mantle source itself. The lower redox state we describe refers specifically to the $\text{Fe}^{3+}/\Sigma\text{Fe}$ ratio in the basaltic melt, which decreases with increasing T_p .

150-154 1st sentence: keeping your broad readership in mind, perhaps it would be better to say that a higher T_p results in a lower $f\text{O}_2$ via its effect on the pressure of melting (which is deeper), at least in the garnet peridotite facies - this is what you suggest? Also, we've known this before, so this is not a new finding from the present study. This is why AS16 and Zhang+24 applied a P correction (though wrongly extrapolating the garnet peridotite $f\text{O}_2$ -P profile to 1 GPa). AS16 actually estimated P of melting from their preferred T_p evolution curve, so the link is clear. Finally, it is necessary to be very careful with the terminology: here you refer to a lower mantle redox state despite constant $\text{Fe}^{3+}/\text{Fe(T)}$ - I think you mean $f\text{O}_2$

Answer: See above answer.

looking at Stolper+20 AmMiner, it is clear that 2.4 GPa is below P where the simple relationship between pressure and $f\text{O}_2$ in the garnet peridotite facies breaks down and $f\text{O}_2$ varies significantly and non-linearly as a function of changing modal abundances

Answer: Stolper et al. (2020) is indeed an excellent reference. We cited this. However, it is important to note that their model is based on solid-solid phase relationships. Once the melt phase is taken into account, the oxygen fugacity and corresponding values can shift, even under identical P-T conditions and bulk compositions.

"previous" - give reference. If Zhang+24, they overestimated Archaean $f\text{O}_2$ in two ways, as I explain elsewhere in this review). So should we be worried about the consistency with your approach?

Answer: This has been clarified. Please note that we did not apply any corrections to the obtained $f\text{O}_2$ values- they directly reflect the source region's oxygen fugacity. Furthermore, all values we cite refer to source $f\text{O}_2$, not to corrected values such as the "potential oxygen fugacity" used in Zhang et al. (2024).

what do you mean by in-situ? what do you mean by Mg-calibrated? Please note:

Answer: Thank you for pointing this out. We have deleted the relevant discussion. Since rocks in Aulbach and Stagno (2016) can indeed represent MORB sources, we have also included the relevant rocks in the new version.

(1) Zhang+24 made the same 2 mistakes when applying their corrections to the metabasalts reported in AS16 that they did with the basalt database, and their Archaean metabasalt-based fO_2 estimates are also too high as a result.

(2) I happen to be familiar with ref. 6: AS16 did more than just look at MgO content. We used major and trace element constraints to understand the samples' petrogenesis, which allowed us to exclude samples with cumulate protoliths, or those that had been metasomatised or that were LREE-enriched suggesting derivation from an enriched source or melting in the presence of lithospheric lid - this was done to ensure that the comparison to forward-modelled melt V/Sc (to estimate source fO_2) and to modern MORB is justified. For many of the sample suites, AS16 used reported radiogenic isotope systematics to further demonstrate their derivation from a depleted mantle source, as is true for modern MORB, and given that at least mildly depleted sources existed in the Archaean. This was further corroborated in Aulbach+Arndt19 EPSL where many of the same eclogite suites for which AS16 estimated fO_2 are shown to have initial $^{87}Sr/^{86}Sr$ indicative of a protolith derived from depleted mantle.

Answer: We fully acknowledge that some samples in your study are of MORB origin. Therefore, in the new version, we have also included your sample as part of the database. We have made a concerted effort to demonstrate the MORB-like nature of the samples in our database. These samples exhibit characteristics of derivation from a depleted mantle source and show no evidence of arc-related influence. We also narrowed the $(Nb/La)_{PM}$ range (≥ 1) to further minimize the possibility of crustal contamination.

(3) Your and other's basalt database, on the other hand, includes in particular Archaean basalt samples that are not spreading ridge-derived, with $(Nb/La)_N$ down to 0.75, and the onus is on you to demonstrate that these samples record the fO_2 of a convecting mantle source that can be usefully compared to the modern MORB source. That is, without complications that arise from mixing tectonic settings.

Answer: Yes, in the revised version, we have further refined our basalt database by narrowing the sample selection criteria -- for example, by requiring $(Nb/La)_{PM} \geq 1$ -- to minimize the influence of crustal contamination and non-MORB-like geochemical signatures.

any back-calculated result would match some result from the numerical simulations?

Answer: Here we are talking about the average value, but since numerical simulations can produce thousands or even tens of thousands of results, we can always find some that are very similar among them.

redox state of the Archean mantle as proxied by its $Fe^{3+}/Fe(T)$

Answer: Have done.

what is the oxidation degree and how do you double it?

Answer: In this article, the oxidation state is simplified to the proportion of ferric iron (Fe^{3+}). Therefore, assuming other elements remain in their reference valence states, the oxidation state of the mantle has doubled compared to before. See also answer for comments of Line 143.

184-185 again, the effect of TP is indirect, and I think it would be better to make it more explicit. “Deeper onset of melting owing to higher Archean mantle TP” alone might be misinterpreted

Answer: We have rewritten this sentence.

not only that, but it confirms earlier findings as per (6, 13, 14) – worth mentioning here

Answer: Yes, has been mentioned.

MORB with $f\text{O}_2$ near the QFM oxygen buffer

Answer: Have done.

188-189 sure, but this last statement is hanging in the air, as it is not followed by some statement as to whether this needs further investigation, or whether it conflicts with some evidence. Perhaps you can just leave it out.

Answer: Yes, it has been deleted.

this is likely, but currently they are “thought to represent” Archean - or were $^{187}\text{O}/^{188}\text{O}$ reported for these samples?

Answer: To our knowledge, $^{187}\text{O}/^{188}\text{O}$ ratios were not reported for those samples; instead, the inference was made based on estimated degrees of melting. Accordingly, our interpretation in that section follows their assumption to reconstruct the nature of the corresponding initial mantle source.

in Figure 4

Answer: Have done.

then we calculated

Answer: Have done.

which other redox parameters?

Answer: $f\text{O}_2$, redox state of minerals, etc.

what kind of melting? Batch or fractional? It makes a difference for how concentrations of incompatible components in the residue evolve

Answer: It is from mafic to komatiitic melts, depending on pressure-temperature conditions. Melting is batch rather than fractional. The melt fraction (F) refers to the proportion of melting relative to the initial source composition, not the incremental fraction after each melting step. The different data points represent values

sampled at equal temperature intervals.

high degrees of partial melting

Answer: Have done.

suggests that Archean mantle was more reduced than the modern mantle

Answer: Have done.

214-217 Please add a statement/caveat regarding uncertain provenance/tectonic setting of basalts, a potential preservation bias linked to the supercontinent cycle and poor statistic coverage of certain eras.

Answer: Have done. Please see Lines 313-323

218-219 by the time a melt is able to saturate zircon, it is so far removed/processed from the original mantle-derived melt that it is questionable at best that the fO_2 they record reflect the mantle rather than some later process

Answer: Yes, we have added some sentences to clarify the reliability of the relevant records.

232-233 I suggest to rewrite "... this hypothesis, as the accretion of meteorites with negative ... lowered the W isotopic composition of the silicate mantle with an initial positive W isotope composition (+200...) in the wake of core formation and evolution at high Hf/W".

Answer: Have done.

when did these meteorites arrive? Given how short-lived the system is, there must be an estimate for the timing

Answer: At the Archean-Hadean boundary. Have done.

should be ref 29? Might need to check neighbouring references, too

Answer: Have done.

235-237 please rewrite, this sentence is not well constructed

Answer: Have done.

how do you define habitability? Life evolved even as the atmosphere contained $\ll 1\%$ PAL O_2 . So this change in mantle O_2 would have affected the course of evolution of life, but not hindered life per se.

Answer: Yes, "habitability" is a multidisciplinary concept, and we think its specific definition may vary depending on the research context. In our view, habitability is primarily determined by energy and matter. We think the central question is whether an environment can support the existence and sustained evolution of life. In our view, supporting the existence of life does not necessarily require an oxygen-rich environment, as cyanobacteria existed long before the Great Oxidation Event. However, when it comes to the sustained evolution

of life -- especially complex, carbon-based life as we know it -- oxygen-rich conditions are essential. Therefore, in our manuscript, we will add “oxygen-rich” before “habitability”, to refer specifically to environments capable of sustaining the long-term evolution of complex life. In the revised version, we have removed it.

246-249 this is much too simplistic an argument, and untenable. Even modern OIBs still contain components that must have been generated during earliest Earth evolution including W (e.g. Mundl+17 Science) and Nd (Horan+18 EPSL). The sources likely reside in long-lived heterogeneities (possibly atop the core) that cannot be framed in terms of upper vs. lower mantle

Answer: Indeed, this part of the content may not have been clearly expressed in the original version. We agree that the original ‘upper/lower mantle’ argument was oversimplified, and the evidence based solely on tungsten (W) and neodymium (Nd) isotopes is relatively weak. We have substantially revised this section to address these points.

For discussion about isotopes, in the revised version, we removed the discussion of neodymium isotope because discussion parts involving Nd are very limited. We instead incorporated recent findings (Messling et al., 2025) on ruthenium (Ru) isotopes to support the addition of late accretion material in the mantle in Archean in this answer. The key insight from (Messling et al., 2025) is that the ^{182}W anomalies observed in modern ocean island basalts (OIB) and early Earth rocks are likely originate from different mechanisms: modern OIB (which represents core leakage) show positive ^{100}Ru and negative ^{182}W , whereas Archean samples show both positive in ^{100}Ru and ^{182}W . Therefore, continuous core interaction couldn't fully explain the evolution of ^{182}W and ^{100}Ru in Archean, and this observation could be only explained by the incorporation of meteoritic material with negative isotopic signatures into the mantle (Fig. 5 and Fischer-Gödde et al. 2020).

Below is Figure 3a from Messling et al 2025, showing the effect of late accretion and core:

Therefore, for mantle mixing, if continuous core-input cannot explain the transition of ^{182}W from positive Archean values to around zero for modern MORB (otherwise the constraints from ^{100}Ru would be invalid, as

core input cannot decrease the mantle ^{100}Ru to present day value), a complementary negative ^{182}W reservoir that participates in later mantle mixing would still be required, which is likely located in the deeper mantle.

On the other hand, when discussing the dynamics of this mixing process, we agree that it is important to avoid overemphasizing a rigid “upper mantle versus lower mantle” division. However, we would like to emphasize that this view is not only based on isotopes but also based on our numerical modeling results (see figure and text in green box below), which show that Archean subduction differs significantly from modern subduction. This difference is due to phase transitions at 410km and 660km, thus causes differences in the upper/lower mantle.

We also should not assume that the mantle has become fully homogenized. As previous studies have pointed out, even today, some anomalous components still exist in the mantle. Therefore, we prefer to describe

the mantle as having undergone an “approximately homogeneous mixing” process -- meaning that although not perfectly uniform, it has reached a broadly equilibrated state over billions of years of geological time. This has been demonstrated in several studies.

Therefore, we kept the discussion about late accretion in the text as it's consistent with the constraints from W-Ru isotopic system and the inferred oxidation states in our model. However, for mantle mixing, we have removed the detailed discussion based on ^{182}W , given the uncertainties with mantle mixing due to unknown extent of the negative ^{182}W reservoir and lack of data during Proterozoic. A detailed discussion lies outside the scope of this study.

“Archean rocks have W and Nd isotopic signatures...” (what was fractionated were the parent and daughter elements

Answer: Have done.

not today, but if iron disproportionation and extraction of metal to the core is real, then it would have been at the time of late accretion.

Answer: Have done.

Sure, but in modelling a lot depends on parameters, including some that are not well-constrained. I am not convinced that the last word has been spoken on whether or not metal formed by disproportionation can be extracted to the core. Hirschman 23 EPSL (cited, though not in this context), invoked a basal magma ocean to explain Earth's redox evolution, and recent work (Boukaré+25 Nature) suggests that this would have been inescapable on Earth. It seems difficult to argue that metal formed by Fe^{2+} disproportionation on top of the core (expected at this pressure) would not have been able to be efficiently extracted to the core even if Boukaré+ were not concerned with this particular aspect.

Answer: We agree that the mantle plume model is a reasonable explanation for redox budgets, and we think we both agree that Fe^{2+} disproportionation requires the presence of a mineral, such as perovskite, that can accommodate Fe^{3+} . However, recent thermodynamic calculations (Stixrude and Lithgow-Bertelloni 2024) suggest that the Fe disproportionation reaction occurs only at the top of the lower mantle (near mantle transition zone), which was recently observed in natural samples (Pan et al., 2025). This suggests that iron can remain in place at least. If the thermodynamic results and this observational evidence are correct, then it implies that the mantle redox budget has not changed. This implies that FeO in the BMO (basal magma ocean) does not undergo disproportionation. The reason that this reaction only occurs at the top of the lower mantle is not due to the depth of the magma ocean, but is instead determined by the thermodynamic properties of the minerals.

For Boukaré et al., (2025): setting aside the thermodynamic results (i.e., Stixrude and Lithgow-Bertelloni 2024), if we assume that Fe²⁺ disproportionation forming perovskite can occur under the lowermost mantle conditions, then in Boukare paper, the melt fraction in the BMO region all reaches 100% (their Fig. 1j and Supplementary Video 1). If that is the case, it would be difficult for perovskite to crystallize from a fully molten BMO. Without crystallized Fe³⁺ perovskite, my understanding is that the disproportionation reaction may also hardly occur.

Anyway, we acknowledge that the mantle plume model is an appealing explanation, and we have revised the relevant discussion for both models of BMO-related mantle plume model and subduction model.

what do you mean by “not show mantle oxidation mechanism”? This is awkward English. Is it correct to simply say it has a more reduced mantle than Earth? Despite having a metallic core, a lower mantle composed of perovskite and despite having had a (basal) magma ocean, which are pre-requisites for this mechanism to work?

Answer: Have modified.

I don't agree that this is more plausible. In order to have permanent mantle oxidation you need to permanently sequester reducing power, which you can do by extracting metal to the core or having hydrodynamic escape of H₂ which surely would have been much more efficient during earliest planetary evolution (even if it still occurs at an exceedingly low level today). Taking oxidising power from the ocean-atmosphere system and sticking it into the mantle means you are reducing the surface reservoirs, thereby inhibiting the advent of the GOE.

Answer: We have revised the wording from “more plausible” to “another”. In order to achieve net oxidation of Earth's surface environment, the reducing flux must either be lost to space, sequestered into the core, or consumed by biological processes. The plate tectonic (or subduction-related) mechanism we now propose emphasizes the need to couple hydrogen escape and biological consumption over geological timescales.

However, as we noted in our previous discussion, when considering hydrogen escape (and biological consumption) as an oxidation mechanism, it is crucial to remember that the oxidation flux is not solely determined by the escape rate (and consumed rate), but also by the duration over which escape occurs. That is, the related redox budget = H₂ escape flux (+ H₂ consumed by microbes) = hydrogen escape rate × escape duration (+H₂ consumed rate × consumed duration) (i.e., the accumulation timescale of surface oxidation flux).

For example, during the Archean, the redox budget driven by hydrogen escape could reach $\sim 7 \times 10^{12}$ mol O₂/yr (Catling et al., 2001). Over a hydrogen escape timescale of 1.0-1.5 billion years (from ~ 3.9 to 2.4 Ga), the

total accumulated redox budget -- presumably stored in the oceanic lithosphere -- is sufficient to explain a global oxidation transition. These oxidation reservoirs could have been eventually recycled into the mantle during a single Wilson cycle (~200 Myrs) via subduction.

Furthermore, photosynthesis clearly existed well before the Great Oxidation Event (GOE), which suggests that a significant portion of the reduced flux was already being consumed biologically. The net long-term oxidative storage is likely preserved in hydrated, oxidized lithospheric rocks. Our mechanism highlights the quick features of global redox budgets input mantle via subduction.

Besides, this subducted material and redox budget would eventually resurface (we see it in OIBs and even MORBs), resulting in no net change. To the extent that mantle fO_2 affects the fO_2 of magmatic gases, you would, after the onset of subduction of oxidised material, still have a shallow reduced mantle degassing reduced volatile species that are O_2 sinks.

Answer: The global net subduction redox budget ($36 \pm 6 \times 10^{12}$ mol/year) we calculated has already subtracted the budget associated with rock and degassing related to arc, MORB, and OIB (see figure below; diagram on the right: RB input including RB in Fe^{3+} of rock; RB output including RB in Fe^{3+} , S^{6+} of rock and sulfur degassing; Since the input and output of C are balanced (diagram on the left), it is not included in the diagram on the right).

[REDACTED]

In fact, before the oxygenation of ocean bottom waters in the Neoproterozoic, the redox budget of what subducts might not have been hugely positive (see Stolper+Keller 18 Nature; Tomkins+Evans 15 EPSL), and you would need to subduct a great volume to achieve an effect on the mantle, which then must be reconciled with other evidence for the maximum mass of subducted material in the mantle.

Answer: Yes, we agree -- if you are referring to those Phanerozoic highly oxidized rocks (e.g., $Fe^{3+}/\Sigma Fe > 0.5$) such as those shown in the figure from Stolper and Keller (2018), they are indeed not suitable for past flux

estimations. We've taken that into account. To avoid overestimation, we based our flux calculations on moderately oxidized rocks, specifically those with $\text{Fe}^{3+}/\Sigma\text{Fe} < 0.26$, which is consistent with observation. For further details, please refer to our response to comment NO. 8.

that's 500 Myr later than what you call for at the beginning of the paragraph.

Answer: Yes, the timing of mobile-lid tectonics spans a broad interval, with many researchers suggesting it occurred between 3.2 and 2.5 Ga. Therefore, the event mentioned here at ~2.7 Ga falls within that timeframe. We have revised the wording.

263-275 please clarify overlaps with the manuscript under consideration as ref. 38

Answer: Only the calculation results from this reference are cited here; there is no duplication involved.

they are transporting redox budget, not fluxes. Besides, this redox budget is then "missing" at the surface and unavailable to oxidise surficial reservoirs.

Answer: We have already revised "flux" to "budget," and the net redox budget we refer to has already accounted for the portion that is missing from the surface.

ref. 38 is miscited in this place. They showed that komatiite serpentinisation produces H_2 and that's it. They never mention subduction or mantle redox. Furthermore, I question that the H_2 so produced could be quantitatively lost from the Earth system, as hydrodynamic escape would have been highly inefficient by the time komatiites were emplaced

Answer: Please note that our calculations ($36 \pm 6 \times 10^{12}$ mol/year) do not include the redox budget from serpentinization of komatiites. We only consider it as a potential additional budget. Moreover, since serpentinization readily occurs underwater, most serpentinized komatiites -- being part of the Archean oceanic crust -- were likely subducted long ago (former Ref. 40; Tamblyn and Hermann, 2023). In fact, another paper (Tamblyn et al., 2023) by the same authors of former Ref. 40 also discusses how the subduction of these serpentinized komatiites contributed to the formation of TTGs, indicating that they too recognize these rocks would have been subducted.

As for hydrogen escape during the Archean, as mentioned in our previous response, the escape rate could have been substantial. Furthermore, the total hydrogen escape flux is a function of both the escape rate and the duration over which it occurred.

271-274 (1) It is insufficient that ultra-oxidised oceanic crust occurs. What matters is its mass. E.g. Cratonic eclogite xenoliths undoubtedly represented subducted oceanic crust, including having fractionated O isotopic compositions pointing to seawater alteration, have consistently much lower $\text{Fe}^{3+}/\text{Fe(T)}$ than what you state here

(no need to cite, I am just referencing in support of my statement: Aulbach+Smart 23 AREPS, Aulbach+22 JPet, Aulbach+19 SciRep). (2) It is not admissible to assume a modern subduction redox budget when we know that the atmosphere was oxygenated to just 1% PAL as late as 2.4 Ga ago, and that oceanic bottom waters became oxygenated only in the Neoproterozoic (Stolper+Keller 18 Nature), before which time there would have been very little dissolved sulphate, which is the most potent oxidant (Tomkins+Evans 15 EPSL). (3) I wonder where the idea came from that komatiites have relatively high Fe³⁺? If measured Fe³⁺/Fe(T) were reliable, someone would have exploited that before, but they are not. Berry+ 08 Nature were able to make an estimate on a melt inclusion in komatiite and obtained 0.10, similar to modern MORB.

Answer: Regarding points (1) and (2), please note that we did not use highly oxidized AOC rocks with Fe³⁺/ΣFe > 0.26 even >0.5, which are characteristic of the Phanerozoic. The Fe³⁺/ΣFe values we used are 0.20 for serpentinite, 0.21 for gabbro, and 0.26 for basalt (see our response to comment NO. 8). At the same time, the Archean oceanic crust was thick (20–25 km), and we did not assume that the entire crust was altered. We only suggest that alteration occurred in the upper few kilometers, which would be sufficient to carry redox budget (RB). From this perspective, the estimate is conservative.

We are not sure whether the Fe³⁺/ΣFe values in the cratonic eclogites mentioned here refer to the original magma composition or the post-alteration state. The redox budget we refer to is based on the altered (i.e., hydrated or serpentinized) state. Moreover, it is worth noting that eclogites likely underwent dehydration. Assuming the altered rocks originally had Fe³⁺/ΣFe ratios of 0.2–0.3 (Figure in response to comment NO. 8), a portion of their redox budget (RB) could have been released into the sub-arc mantle during subduction, through reactions involving sulfur (see Duan et al., 2024). This implies that the RB remaining in exhumed eclogites may not reflect the total redox budget acquired during seafloor alteration prior to subduction.

The redox budget we calculated already subtracts the oxygen sinks associated with MORB, arc, and OIB components. It also does not consider the oceanic bottom-water sulfate unique to the Phanerozoic. Therefore, invoking a Great Oxidation Event or requiring widespread oxidation of ocean water is not necessary -- only water-induced alteration reactions (i.e., hydration) are needed.

Regarding point (3) The relatively high Fe³⁺ content in komatiites does not refer to their original (primary) ferric iron content, but rather to the Fe³⁺/ΣFe ratio after serpentinization alteration (Fig. b in comment NO. 8). Large datasets indicate that most komatiites have undergone serpentinization, and these serpentinized komatiites commonly exhibit elevated Fe³⁺ proportions.

parent-daughter elements are fractionated not the isotopic compositions, which trace different sources with

different isotope compositions owing to elemental fractionation at some time when the system was still alive. See my comment above regarding the inadmissibility of casting these different sources of ancient signatures in terms of upper vs. lower mantle.

Answer: The isotope-related discussion has been removed; our analysis is now based exclusively on numerical simulations.

283-284 seems an ad hoc statement and requires a little more info

Answer: The isotope-related discussion has been removed; our analysis is now based exclusively on numerical simulations.

287-288 this was already proposed by Aulbach+19 SciRep based on the eclogite and komatiite record.

Answer: It has already been cited.

288-289 please see my comment above. This is worded peculiarly and moreover falsely diminishes how ref. 6 or 11 scrutinised their database (although 6 went a step further to demonstrate derivation from a depleted mantle source) – see similar comment above

Answer: We have removed the relevant statement to avoid any misunderstanding.

why would there have been separate upper and lower mantles? Where are isotopic anomalies from now extinct isotope systems in modern OIBs from when we know for sure we have whole-mantle convection? This paragraph is based on a series of highly uncertain findings and perhaps could be shortened to two sentences invoking the magmatic lull (which was not global, however) as a potential sign of sluggish tectonics that might explain the lack of mantle fO_2 evolution during that period

Answer: As noted above, the anomalies observed in modern OIBs result from mechanisms clearly distinct from those in the Archean. Recent studies have attributed OIB signatures to possible core leakage. While present-day observations support whole-mantle convection, we agree that discussions related to the early Archean and Mesoproterozoic are uncertain and not central to the focus of this study. Therefore, we accept the suggestion to shorten the discussion, retaining only the point that the magmatic lull may reflect sluggish tectonic activity, which could help explain the apparent stagnation in mantle redox evolution during that period.

ref. 6 did not invoke a Neoproterozoic rise (though we were tempted)

Answer: Have modified.

hm. This was surface driven (increased photosynthesis and organic carbon burial, plus more; e.g. Och+12 EarthSciRev), which ultimately caused oxygenation of oceanic bottom waters. This increased the subduction flux of redox budget, as seen by the advent of certain convergent-margin ore deposits (Tomkins+Evans 15

EPSL), then may have trickled down into the mantle

Answer: We agree. However, as we've emphasized, even the net subduction budget of moderately oxidized rocks alone is sufficient to drive mantle oxidation. Please see answer for comment NO. 8.

313-326 last paragraph is too wordy, repetitive and vague. Could be shortened to a crisp and strong final statement.

Answer: We have condensed it into a clear and concise concluding statement.

propose

Answer: Have done.

except for large impacts, nothing of what follows is abrupt.

Answer: Have done.

greater quantities of... fluxes makes no sense, the flux already comprises a quantity (referenced to a time unit). Please reword.

Answer: Have done.

I disagree. Some estimates, including Aulbach+Arndt 19 EPSL, are lower (100-150 oC). Low-T estimates from basalts around 2.9 Ga that support a warm rather than hot Archaean ambient mantle (e.g. Zhang+24 Natcomms, cited), continue to be ignored. But that's a different story...

Answer: Following your suggestion, we have shortened this subsection to highlight that basalts are more suitable mainly due to their smaller P-T uncertainties. We have read the Aulbach and Arndt (2019) paper, which constrains a lower temperature range, and we have incorporated those constraints into our earlier discussion. However, we focus on the higher-end estimates in this paragraph because if similar spreading centers and Archean-style subduction could occur even under higher mantle potential temperatures, then such tectonic activity would likely persist as the mantle continued to cool.

please specify somewhere the composition of the mantle that you modelled (Fe³⁺/Fe(T) is not enough to characterise the system).

Answer: We have added this information; please refer to the citation below.

remove first comma

Answer: Have done.

how thick in km?

Answer: ~20-25km.

what do you mean by "melt-bearing peridotite"? The melt was similarly dense as the peridotite and failed

to separate? Or simply the peridotite being above its solidus T?

Answer: The term "melt-bearing peridotite" refers to a state in which the original solid peridotite transitions, upon exceeding a temperature threshold, into a mixture of peridotite and partial melt. It is simply the peridotite being above its solidus T.

basaltic or picritic?

Answer: Basaltic and picritic. It depends on pressure-temperature conditions and petrological definitions.

why hydrated?

Answer: This represents a secondary alteration process, in which the rock was hydrated through reaction with seawater at the surface.

344-348 Are we still in Fig. S6? Can you refer to panels, additionally?

Answer: Yes, have done.

I think the concept of ambient and excess TP is useful in that can explain why komatiites form while ocean basins are made up of picrites, so there is no conflict at all. Syntax in last part of sentence needs fixing

Answer: Yes, have done.

354-357 It is fine to not consider komatiites in your work and to say that the petrogenesis of spreading-ridge derived basalts, but this is an inexact summary of the state of the art, and a weak way to dismiss the komatiite record. With the exception of a single komatiite (Comondale), where subduction MAY be a possibility, there is no evidence for komatiite in subduction zones, or from particularly wet sources. No one says komatiites form at the core-mantle boundary. Please duly distinguish between mantle sources and melts. Please refer to Waterton+Arndt 25 (In: Hofmann+ eds The Archaean Earth: Tempos and Events 2nd Edition of The Precambrian Earth), Puchtel+22 ChemGeol, Puchtel+Arndt 25 TOG, Herzberg 16 JPet etc. By the same token, why not acknowledge that virtually no Archaean basalt is strictly comparable to modern MORB (and hence mantle sources) because they were emplaced on continental margins?

Answer: We have removed the potentially inaccurate summary regarding komatiites. For basalts, we have applied more stringent selection criteria to ensure that the selected samples most likely originate from MORB-like source regions. We agree that identifying true MORB samples is challenging -- this is precisely why, out of tens of thousands of samples, only a few dozen meet the criteria. In addition, we have corrected our previous oversight by including the metamorphosed MORB samples from Aulbach and Stagno (2016) in our dataset.

where is the evidence for that? Sources such as Early Enriched Reservoir and similar that were sluggishly mixed throughout the mantle, are though to reside in geographically isolated parts of the (lower) mantle and do

not require separate UM and LM compositions

Answer: Have deleted.

please don't refer to komatiites as highly enigmatic. They are not.

Answer: Have deleted.

how did accreted meteorites get into the lower mantle, if not through the upper mantle? Wouldn't they have equilibrated with an upper magma ocean more efficiently than with a partially solidified lower mantle?

Answer: Meteorites, being rich in iron and thus having higher density, tend to continue sinking into the lower mantle rather than remaining in the upper mantle.

370-372 In the discussion, you say that the mantles were already mixed and that plate tectonics are weak. Here, you want to bring up deep reduced mantle. Can you have it both ways?

Answer: Yes, we believe both can be true. Since the mantle was largely mixed (while not excluding the existence of chemically anomalous domains), a relatively reduced lower mantle would allow the mixed mantle to effectively suppress oxidation in the upper mantle -- or even reduce previously oxidized upper mantle regions. This is supported by rock records. Unless there was a continuous influx of redox budgets into the mantle during the Mesoproterozoic, further oxidation could not be sustained. However, due to weakened plate tectonics, such input would have been limited.

We suggest that, if applicable, isotopic data can only provide an upper limit for the timing of mantle mixing, rather than a lower limit. This is because, as more data become available, the lower limit may continue to shift downward. In addition, numerical modeling evidence indicates that subduction styles characteristic of the Archean-Proterozoic boundary -- namely, recycling of subducted materials within the upper mantle -- may have persisted until before the Neoproterozoic. Therefore, we propose that the final equilibration between the upper and lower mantle (excluding chemically anomalous domains) may not have been achieved until the Neoproterozoic.

Therefore, we propose that a reduced lower mantle, ongoing mantle mixing, and a decrease in redox budget input caused by diminished tectonic activity collectively contributed to the stalling of oxidation during the Mesoproterozoic.

ref. 55: this is a different citation from the two you referenced above when talking about the spread of Archean TP

Answer: We have added the previous references. The temperature of 1500 °C can encompass the Archean potential temperatures (Tp) reported both in Herzberg et al. (2010; 1500-1600 °C) and Aulbach and Arnd (2019;

1500°C).

I am all for making thick Archaean mantle by decompression at high temperatures (in my reading, 1600 oC, require excess TP), but please note that these are present-day thicknesses (for intact cratons). Ancient diamond formation supports that cratons already reached thicknesses of 150-180 km.

Answer: Yes, it is indeed.

447-448 fix this sentence.

Answer: Have done.

not crystal-clear what “these” refers to

Answer: Have done.

Please specify this is the depleted MORB mantle (lithosphere contains a lot of variably depleted mantle, too).

Answer: Yes, have done.

up to 1.2 orders of magnitude? Average 1.2? Surely this must depend on the modelled conditions??

Answer: Up to 1.2 orders of magnitude.

please explain what motivates this choice of 0.02 and 0.04

Answer: Yes, have done.

and are these increments then aggregated?

Answer: No, it is the actual value representing each temperature-pressure point.

why 0.04 and 0.01 here, but 0.04 and 0.02 above?

Answer: Yes, Has already explained.

what kind of age correction? Why?

Answer: Has already explained.

fix materia

Answer: Yes, have done.

“low degree of alteration”. How was this assessed? Did you use LOI?

Answer: Highly altered samples with a loss on ignition higher than 6 wt%.

As importantly, Nb/La_{PM} <1 points to a continental influence, either via crustal assimilation, or by formation in convergent margins

Answer: To ensure minimal crustal contamination, we further filtered the dataset using a stricter $(\text{Nb/La})_{\text{PM}} \geq 1$ threshold (see original text for details).

One would hope that you retained samples with an arc magmatism likelihood $\ll 10\%$. I think this is expressed wrongly?

Answer: The relevant criterion is no longer used due to concerns raised by Reviewer 2.

how many of how many is “most”?

Answer: The relevant criterion is no longer used due to concerns raised by Reviewer 2.

what are zircon quantities?

Answer: Eight grains are considered to be mantle-derived, based on their oxygen isotope characteristics.

528-529 Maybe, but a recent study finds that V/Sc is less sensitive to mantle source and melt fraction (Liu+25 GPL). For the P you report, garnet-present melting is subordinate, anyways (as per your next paragraph).

Answer: While their approach appears to rely on large-scale data analysis, our study emphasizes the thermodynamic behavior of elements.

what is the P interval? Is it itself T-dependent?

Answer: Basically, its interpretation is based on correlations, as seen in the garnet appearance line in Figure 1, rather than on a fixed pressure or temperature.

more specifically a previous empirical model

Answer: Have done.

awkward sentence, please rewrite

Answer: Have done.

following ref. 11 is bad news because they calculated fO_2 based on V/Ti and V/Sc, which records source fO_2 directly (forward-modelling of peridotite melts) rather than low-P crystallisation, but then applied a correction assuming that the melt self-reduces during decompression after separating from the source. This reduction is relevant to olivine-melt V partitioning, and to $Fe^{3+}/Fe(T)$ recorded in the emplaced melt, but not to forward-modelled V/Sc or V/Ti. I gather from a later paragraph that you did not apply this wrong correction.

Answer: Yes, we did not used. We have consistently emphasized that our fO_2 values represent the redox state of the mantle source, and that the $Fe^{3+}/\Sigma Fe$ ratios also reflect the source characteristics.

It is very important that you clearly expose how you estimated uncertainties of individual fO_2 estimates, which will have contributions from uncertainties in PT estimates and on source composition

Answer: Has done.

562-563 See comment to line 545. The statement itself is inaccurate because olivine-melt V partitioning is a V-based oxybarometer that does not record fO_2 of melt formation (in the mantle source), but of low-pressure

saturation of liquidus olivine when the melt may have self-reduced.

Answer: Already corrected.

last sentence is a bit strange. Perhaps say it is also observed in the database assembled in this study, or similar

Answer: Already corrected.

I don't clearly understand how this works. You want to learn about $\text{Fe}^{3+}/\text{Fe(T)}$ in natural samples by comparison of sample-derived V/Ti-based $f\text{O}_2$ to modelling results. You only have one independent $\text{Fe}^{3+}/\text{Fe(T)}$ estimate (from the model). What am I missing? Please make it more explicit.

Answer: The former approach uses the measured $f\text{O}_2$ to reverse-calculate the $\text{Fe}^{3+}/\Sigma\text{Fe}$ ratio and does not rely on the whole-rock composition. In contrast, the latter is a forward-modeling method that requires the input of the initial whole-rock composition and the $\text{Fe}^{3+}/\Sigma\text{Fe}$ ratio (obtained from the reverse calculation) to compute the $f\text{O}_2$ under the corresponding conditions. Although both methods utilize thermodynamic calculations, they differ slightly in approach.

the (inherited) zircons, but not the melts (or lavas) are in the record. The melts were likely SiO_2 -rich and not mantle-derived, but either remelted or strongly differentiated from a mantle-derived melt, so as to be able to saturate in zircon. Either would imply that zircon $f\text{O}_2$ does not record that of the mantle-derived melt and much less that of the mantle source.

Answer: We agree, and we have already acknowledged this possibility.

I attach my comments to figures and captions in the main text and in the supplement directly in the pdfs.

Answer: We have responded to or revised all the points raised. Finally, we would like to express our sincere gratitude for your thorough and thoughtful comments, which have significantly improved the quality of our manuscript!

8 April 2025 Sonja Aulbach

Reviewer #2 (Remarks to the Author):

This manuscript explores the redox evolution of Earth's mantle since the early Archean, employing thermodynamic-thermomechanical numerical simulations and a comprehensive database of mid-ocean ridge basalt-like (MORB-like) samples. The authors perform advanced numerical simulations to calculate the redox state of basaltic melts under varying mantle potential temperatures and mantle $\text{Fe}^{3+}/\text{Fe}_{\text{Total}}$ ratios. The study suggests that the mantle's average oxidation degree has approximately doubled since the early Archean,

reflecting significant geological and tectonic reorganization events.

The main innovation lies in the integrated approach combining empirical observations with high-resolution numerical simulations, highlighting the intrinsic coupling between Earth's oxygen-rich habitability, biological evolution, and tectono-magmatic processes.

However, this study has some shortcomings in the selection and organization of MORB-like basalt samples, which undermine the robustness of its conclusions. These issues include: (1) There are many repeated samples with consistent chemical compositions in the dataset, but they display different ages, P-T-fO₂ conditions and Fe³⁺/Fe^{Total} values. (2) The machine learning (ML) method used in this study may have several drawbacks, leading to some uncertainties in MORB-like sample filtration. (3) The quantity of MORB-like samples (if they are) varies significantly over time and is insufficient overall. Therefore, the trend of basalt oxygen fugacity presented in Figure 3 may have some problems. As the following discussions are all dependent on this trend, I suggest that the authors should deal with the above-mentioned comments first.

Answer: Thank you for these thoughtful and constructive comments. We sincerely appreciate your recognition of the integrated approach in this study, which combines thermodynamic–thermomechanical modeling with empirical data. In response to the specific issues you raised, we reply as follows:

Duplicate samples with inconsistent parameters: This issue mainly arose during the process of merging different databases. We acknowledge that some samples in the database have similar major element compositions, but they generally differ in trace element compositions, which is why they were retained in the initial draft. In the latest version, for the portion of the dataset compiled by us, we have re-examined data quality and adopted stricter filtering criteria to eliminate duplicates and ensure consistency across metadata entries. These corrections have been reflected in the updated analyses and figures.

Uncertainties in the machine learning-based filtering method: In this study, the machine learning method was used only as an auxiliary tool for sample classification. In the initial version, we first applied a series of geochemical selection criteria to conduct a preliminary filtering of the samples, followed by machine learning techniques to identify outliers. Since the initial filtering was already stringent, machine learning only excluded a very small number of samples. Following your suggestion, we have removed the machine learning method in the revised version and, incorporating feedback from both you and Reviewer 1, adopted stricter and more transparent geochemical criteria to enhance the robustness of MORB-like sample identification.

Temporal distribution and coverage of samples: We fully agree that the uneven distribution of samples across geological time is a common challenge in studies of long-term mantle evolution. Nevertheless, we point

out that both the Archean and modern time windows include a sufficient number of high-quality MORB-like samples to support our main conclusion: that the $\text{Fe}^{3+}/\text{total Fe}$ ratio of the modern MORB mantle source is approximately twice that of the Archean. In addition, we also conducted a binning sensitivity test (please see answer for comment NO.5 below). Following the suggestions from you and Reviewer 1, we have interpreted trends for time periods with fewer samples (e.g., the early Archean or Mesoproterozoic) with greater caution, and explicitly advise readers to treat such interpretations carefully.

In addition, we have updated the relevant figures, clarified our methodology, and expanded the discussion to better reflect these uncertainties (please see answers below).

1. The descriptions of Figure 1 are overly simplistic and lack the necessary details to fully convey the key information (After reading the whole manuscript, most of the figure captions have similar problems). How could you obtain this P-T diagram? What is the meaning of the five-pointed stars in Figure 1A? How about the white dotted lines? All of these should require detailed introductions in both manuscript and figure captions.

Answer: Thank you for your comment. Yes, due to the previous words limit, we had to describe these figures briefly. Based on your suggestion, we have now expanded this information within the format requirements of Nature Communications. Please refer to text and figure captions for details.

2. Data Collection and Filtration: The authors used a ML method published by Liu et al. (2024) to filter Archean MORB-like basalt data. However, the training dataset listed in Liu et al. (2024) was subdivided into subduction and non-subduction groups, respectively. The latter contains not only many MORBs, but also a lot of oceanic island basalts (OIBs), oceanic plateau basalts (OPBs), continental flood basalts (CFBs), continental intraplate basalts (CIBs) and continental rift basalts (CRBs) and so on. Therefore, samples with the prediction results > 90% do not necessarily originate from the mid-oceanic ridge magmatism. My main concern is how the authors determine that these Archean basalts are MORB-like, rather than OIB-like, within plate-like or others.

Answer: As previously noted, the machine learning method was only employed as an auxiliary tool after a series of geochemical screening steps had been completed. Our numerical modeling results demonstrate that MORB-like spreading centers can still form under high mantle potential temperatures (T_p), up to 1600 °C. In this study, we made every effort to identify rocks most likely formed in such or similar tectonic settings, ultimately retaining only a few dozen samples from an initial dataset of tens of thousands. Furthermore, in the revised version of the manuscript, we adopted more stringent geochemical criteria based on the suggestions

from both you and Reviewer 1.

With respect to plume-related basalts, we can reasonably exclude them based on trace element geochemistry, pressure–temperature (P–T) formation conditions, and supporting modeling results. (1) Plume-related ocean island basalts (OIBs), oceanic plateau basalts (OPBs), continental flood basalts (CFBs), continental intraplate basalts (CIBs), and plume-influenced continental rift basalts (CRBs) all exhibit geochemical signatures (LREE enrichment) that differ significantly from MORBs (see figure above; OIB, CRB, CFB, CIB data from Georoc database). In contrast, the samples selected in our study display typical MORB characteristics (see figure above).

Moreover, these basalt types differ in their formation P–T conditions. For instance, modern OIBs typically form at around 1540 °C and 3 GPa on average (Lee et al., 2009), and plume-related OPBs also originate at the high temperatures, up to 1564–1614 °C (Hastie and Kerr 2010). These temperatures are not only significantly higher than those of modern MORB-forming settings, but also exceed the average formation conditions observed in our dataset and in previous compilations of Archean MORB-like basalts. Our numerical simulations successfully reproduce these estimates: modern OIBs/OPBs correspond to modeled P–T conditions of ~3.2 GPa and 1568 °C, in good agreement with observational data (Fig. a below). When extrapolated T_p to elevated Archean mantle potential temperatures (e.g., ~1500 °C), plume-related types such as OIBs, OPBs, CFBs, CIBs, and CRBs may form under melting conditions as high as 1750–1800 °C and 5–6 GPa (Fig. b–c below), far beyond the P–T range of any Archean rocks in our dataset. Thus, the P–T conditions provide a robust basis for excluding these rock types.

We are currently focusing primarily on MORB-like rocks due to their smaller P-T uncertainties. Although trace element systematics and P-T conditions indicate that the samples in our database are not derived from plume-related magmatic sources (e.g., Archean OIBs or large igneous provinces, LIPs), we emphasize that even if some plume-related samples were inadvertently included, our conclusion—that the mantle $\text{Fe}^{3+}/\Sigma\text{Fe}$ ratio has doubled—remains robust, and may in fact be a conservative estimate.

Modern OIB/LIP-type basalts typically form under higher P-T conditions and exhibit oxygen fugacities equal to or greater than those of MORBs (commonly reaching up to QFM+2; see Nicklas et al., 2024 and Figure below). According to the trend shown in our Figure 1A, this implies that, at a given fO_2 , mantle sources of higher P-T OIBs/LIPs have higher $Fe^{3+}/\Sigma Fe$ values. Therefore, if even a small proportion of Archean samples included OIB/LIP components, the appropriate modern comparison would need to include modern OIBs as well—whose elevated $Fe^{3+}/\Sigma Fe$ values would further increase the modern reference level.

Given that our current comparison is based solely on MORB data, the actual extent of mantle redox evolution—if Archean OIB/LIPs are mixed in—may well exceed the doubled degree currently observed.

In addition, we think that MORBs and some potentially existing MORB-like CRBs derived from the shallow asthenospheric mantle are more difficult to distinguish geochemically and thermodynamically. This is because such CRBs essentially represent a precursor stage to MORBs, sharing similar geodynamic formation processes and forming under nearly identical P-T conditions—a trend that is also reflected in our numerical models (Fig. d of numerical models above). Therefore, this potential overlap does not compromise the robustness of our conclusions. In such cases, it is sufficient to exclude samples contaminated by continental crust. To this end, we applied stricter geochemical criteria to filter out such samples (e.g., excluding those with $(Nb/La)_{PM} < 1$).

3. After detailed examination of the listed dataset in Liu et al. (2024), the ML method may have some drawbacks:

(1) Nearly two thousand lines of basalt data are older than Cenozoic (65 Ma), which might not be suitable for

using as training data; (2) The generation of CFBs, CIBs and CRBs was commonly accompanied by various degrees of continental contamination. However, the influences of continental contamination were not precluded. Obviously, it is confusing to use samples that are strongly affected by continental materials to discuss their origins and tectonic settings; (3) The listed arc basalts are composed of many samples from back-arc basin areas. However, the simultaneous influences of back-arc spreading and mantle plume were recently proposed for the tectonic evolution of the West Philippine Basin. In essence, the chemical compositions of back-arc basin basalts (BABBs) vary in a large scale, including MORBs, normal arc basalts, and OIBs and so on, suggesting that the generation of BABBs may be complicated. Therefore, labeling the BABBs simply as subduction origins may be unreasonable; and (4) The elements of U, Pb, Ba, Sr and Rb (and the ratios made up by these elements) were also applied to the ML model. But these elements can be easily affected by post-magmatic alterations. In fact, I am concerned about the impact of element mobility on the predicted results of Archean basalts, which were mainly implemented in the 'Black Box'. Therefore, I suggest that the authors should apply a more reasonable method to screen the Archean MORB-like samples.

Answer: Yes, as you rightly pointed out, machine learning methods do have certain limitations. Therefore, prior to applying machine learning in the initial version of the manuscript, we first conducted a preliminary screening using several approaches previously adopted by other researchers. The samples selected through this process exhibited geochemical characteristics broadly consistent with MORBs.

In response to your suggestion and that of Reviewer 1, we have now reconstructed a new dataset for analysis. This time, we did not use machine learning but instead applied more stringent geochemical criteria to re-screen the samples from the database (see Lines 633–668 for details). We selected samples that show no negative Nb (with/without Ta) anomalies and have $(\text{Nb/La})_{\text{PM}} \geq 1$. These criteria help rule out an arc-related origin. Secondly, none of the samples exhibit geochemical or thermal characteristics associated with mantle plumes. Thirdly, we used high MgO (≥ 8 wt %) contents and $(\text{Nb/La})_{\text{PM}} \geq 1$ to avoid crustal contamination.

Our results show that even with this alternative screening approach, the selected samples do not change the conclusions of our study.

4. P-T-fO₂ Calculation of Basalt: The calculated parameters for Fractionated-PT thermobarometer should be introduced. I notice that Table S2 contains many repeat samples (e.g., samples Z2967-49 and S z-1) in lists of 'Data in Zhang et al. (2024)' and 'Data in Gao et al. (2022)'. However, the same sample displays distinct P-T-fO₂ conditions and calculated Fe³⁺/Fe^{Total} values in each data list. In addition, many repeat samples in list

of MORB-like samples in this study have different ages. Therefore, I consider that the organization of basalt dataset in this manuscript should be significantly improved.

Answer: The calculation parameters for the Fractionated-PT thermobarometer have already been described in the manuscript (Lines 676-686). We appreciate your careful review of the sample data used in our study. It should be noted that the P–T–fO₂ conditions for samples from Zhang et al. (2024) and Gao et al. (2022) were not calculated in this study, and only use their P–T–fO₂ conditions to calculate Fe³⁺/ΣFe ratio. We respect the thermodynamic results obtained by the original authors; however, as we are uncertain whether they adopted the same methods (e.g., squeeze theorem or equilibrium conditions) criteria when applying geothermobarometers, discrepancies between their results may exist. Therefore, we only highlight these inconsistencies without recalculating their P–T–fO₂ data. The two external databases from Zhang et al. (2024) and Gao et al. (2022) were used solely for comparison with our dataset and to evaluate the influence of different sample screening criteria and P–T estimation methods. Our results indicate that these differences have no significant impact on the estimation of mantle Fe³⁺/ΣFe (Lines 234-252).

In addition, we carefully examined the P–T conditions you noted as inconsistent. For example, for sample s_Z-1, Gao et al. (2022) reported 1.44 GPa and 1418 °C, and Zhang et al. (2024) reported 1.41 GPa and 1414°C—values that are very close. Our resulting Fe³⁺/ΣFe ratio (marked in green in Table S3) is identical in both cases. In another case, for sample Z2967-49, discrepancies in the calculated P–T–fO₂ conditions led to a difference of 0.01–0.02 in the Fe³⁺/ΣFe of the basalt. These results underscore that, although different P–T–fO₂ calculation methods may introduce minor uncertainties or inconsistencies, the final Fe³⁺/ΣFe ratios of Archean mantle sources remain lower than those of the modern mantle. This indicates that the computational method does not significantly affect the final outcome and, in fact, further supports the robustness of our conclusions.

In addition, there are differences in different petrological databases used by various studies. In the process of data compilation, we have removed some duplicated entries. A small number of samples remaining in the database had similar major element compositions but differed in trace elements and ages. This information was retained in the initial version but removed in the revised version after verification.

5. The quantity of MORB-like samples fluctuates dramatically over time, and the overall sample quantity is insufficient. With these points in mind, I am skeptical about the authenticity of the trend shown in Figure 3. As the following discussions in this manuscript are all dependent on this trend, I suggest that the authors should deal with the above-mentioned comments first.

Answer: As you noted, the sample distribution is indeed uneven, particularly with a lack of samples from the Mesoproterozoic or early Archean. We have already addressed this issue in our manuscript. In fact, this should not be viewed as a limitation of our study per se, but rather as a reflection of the relatively quiescent tectonic activity of the Earth during these stages. As such, we must base our discussions on the limited data available for this time period.

This is not a challenge unique to our study, but a common issue faced by the entire field (e.g., Herzberg et al. 2010; Zhang et al., 2024; Aulbach and Stagno, 2016; Aulbach and Arnd 2019). For example, in studies constraining potential mantle temperatures, Herzberg (2010) had only four samples from the Mesoproterozoic, compared to dozens from the Archean. Similarly, in studies on oxygen fugacity, Aulbach and Stagno (2016) included 1 Mesoproterozoic sample. In Zhang (2024), only a few samples were available for Mesoproterozoic period. By contrast, we included a dozen samples. Moreover, our discussion is not solely based on our own dataset, but also involves a comparative analysis of trends reported in previous studies (e.g., Gao et al., 2022; Zhang et al., 2024; Aulbach and Stagno 2016; Nicklas et al., 2019).

In addition, we conducted two sets of sensitivity tests for the database of this study. The first set used tectonic regimes—episodic mobile lid and stagnant lid—as temporal boundaries, dividing the data into four intervals: Early to Middle Archean ($\text{Fe}^{3+}/\Sigma\text{Fe}$ in mantle = 0.02 ± 0.002 ; $n = 19$), Late Archean to Paleoproterozoic (0.025 ± 0.002 ; $n = 39$; specifically, 0.022 for the Neoproterozoic [$n = 24$] and 0.028 for the Paleoproterozoic [$n = 15$]), and Mesoproterozoic (0.02 ± 0.006 ; $n = 10$). This binning strategy increases the number of samples per group. The results show that the overall trend remains consistent with that derived from the original 250 Ma interval-based classification.

The second set of tests divided the data more broadly into: Archean ($\text{Fe}^{3+}/\Sigma\text{Fe}$ in mantle = 0.021 ± 0.002 ; $n = 43$), Proterozoic (0.025 ± 0.003 ; $n = 28$), and compare them with the modern era (0.04-0.05). Although this binning masks some finer details, it still clearly captures the overall doubling trend in mantle $\text{Fe}^{3+}/\Sigma\text{Fe}$.

We would like to emphasize once again that this is not a flaw in these studies, including our own, but rather a discussion based on the best available data. The absence of samples from specific geological periods due to reduced tectonic activity should not be seen as a reason to hinder the advancement of related research.

Furthermore, due to the availability of whole Archean samples, which can be compared with modern counterparts, the overarching conclusions of our study—as reflected in the title—remain valid. In the revised version, we further talk about this issue and discuss the reliability of the observed trends accordingly. Please see Lines 313-323; 707-714 for our additions.

Finally, we sincerely thank you for your comments.

Reviewer #3 (Remarks to the Author):

Review on “The Mantle Oxidation Degree Has Doubled Since the Early Archean” by Zhu et al. The secular redox evolution of the mantle attracts broad interests because it may influence the long-term evolution of the atmosphere. The redox evolution of the mantle could be dictated by oxygen fugacity (fO_2) or redox capacity ($Fe^{3+}/\Sigma Fe$). Whether the fO_2 has increased in Archean remains hotly disputed (Zhang et al., 2024; Li and Lee, 2004; Aulbach and Stagno, 2016). In this study, the authors inferred the redox evolution of the mantle from the perspective of redox capacity ($Fe^{3+}/\Sigma Fe$). They estimated circuitously the $Fe^{3+}/\Sigma Fe$ of the mantle from calculated T, P and fO_2 . Despite that the authors performed arduous thermodynamic-thermomechanical simulations, the results are not novel,

Answer: Thank you for your comments. However, we disagree with the assessment that “the results are not novel”. As you noted, the question of whether the mantle’s redox state has increased since the early Archean is a very important and widely debated scientific issue. Our study is the first to quantitatively address this mantle redox question by integrating a thermodynamic–thermomechanical modeling framework with a global geochemical database of MORB-like rocks. In doing so, we are **the first to quantitatively derive the evolution of mantle $Fe^{3+}/\Sigma Fe$** —moving beyond the largely qualitative inferences presented in some previous studies. This approach and its results speak directly to, and advance, the ongoing debate you referenced.

In particular, recent studies have produced highly divergent, and in some cases, significantly biased conclusions that risk perpetuating misunderstanding and stagnation in the field (please see also Reviewer 1’s comments, e.g., for line 58-61, points 1-3). Against this backdrop, our work contributes to resolving key controversies, introduces a novel framework for investigating redox evolution, and aims to stimulate further research and discussion within the broader geoscience community.

While earlier studies have primarily focused on either direct or indirect fO_2 estimates and indirect geochemical proxies (e.g., V/Sc), our study adopts a redox budget perspective and introduces a systematic and quantitatively robust framework. We believe this represents a clear innovation in both theoretical approach and methodological implementation. Indeed, this novelty has been explicitly recognized by the other two reviewers (e.g., Reviewer 1: *The authors developed a **novel** approach.....this approach of forward modelling the most sophisticated yet, allowing to understand and quantify the complex relationships of these latter two parameters*).

Finally, we note that you describe the topic as “attracts broad interests” and “a widely debated issue,” while

simultaneously suggests that our **first quantitative results of mantle $\text{Fe}^{3+}/\Sigma\text{Fe}$** , which show a **doubling** from the Early Archean, are “not novel”. We respectfully suggest that this reflects an internal inconsistency. Precisely because the question remains unresolved and prior studies have lacked systematic, data-driven quantification, we developed our integrative modeling approach. In doing so, we aim to shift the discussion from qualitative speculation to quantitative evaluation. We therefore maintain that our study makes both a substantive and methodological contribution to the field.

and even be wrong for the following reasons.

Answer: We disagree. We welcome critical feedback and look forward to any specific points you may raise. However, we wish to clarify that our estimation of $\text{Fe}^{3+}/\Sigma\text{Fe}$ in natural rocks is neither arbitrary nor excessively indirect. It is derived through thermodynamic inversion of oxygen fugacity ($f\text{O}_2$), temperature (T), and pressure (P), using a self-consistent model grounded in redox equilibrium principles.

To ensure the robustness of our results, we employed an iterative bracketing strategy to achieve convergence and cross-validated our estimates with experimental constraints and petrological observations. This approach provides a theoretically sound and empirically supported framework for reconstructing the redox state of the mantle. It is likely more accurate than the oxygen fugacity estimates obtained via the direct elemental ratios used in Li and Lee (2004) that you suggested, which overlook the effects of pressure, temperature, and degree of melting on elemental partitioning.

1. In estimation of $\text{Fe}^{3+}/\Sigma\text{Fe}$ in the mantle, the authors used $f\text{O}_2$ estimated by Zhang et al. (2024) and Gao et al. (2022). Both of the studies show that the $f\text{O}_2$ of mantle-derived basalt increased around 2.5 Ga. Therefore, it's foreseeable that the estimated mantle $\text{Fe}^{3+}/\Sigma\text{Fe}$ will increase. I wonder what the result will be if the authors use a constant $f\text{O}_2$ (as proposed by Li and Lee, 2004) in their calculations.

Answer: We disagree. First, we would like to clarify that in estimation of $\text{Fe}^{3+}/\Sigma\text{Fe}$ in the mantle, our calculation is based on **our own rock dataset** that was independently constructed through rigorous filtering from large geochemical databases. We clearly stated this in the “Methods” section (Lines 625-661). The method we used for calculating $f\text{O}_2$ in natural samples is based on the experimental petrology results of Wang et al. (2019). **The $f\text{O}_2$ results in our own rock dataset were also calculated by ourselves** (this was clearly noted by the other two reviewers). The $\text{Fe}^{3+}/\Sigma\text{Fe}$ values derived from the $f\text{O}_2$ estimates of Zhang et al. (2024) and Gao et al. (2022) **do not** form the basis of our main conclusions and were included solely for comparison with our own

results. The fO_2 estimates of Zhang et al. (2024) and Gao et al. (2022) are only to be used to demonstrate that even with different filtering methods, our conclusion remains unaffected. It should be clarified that all calculations are based on well-established experimental petrology and thermodynamic constraints, therefore it is not appropriate to arbitrarily impose a constant fO_2 value.

Second, an increase in oxygen fugacity does not necessarily correspond to an increase in the $Fe^{3+}/\Sigma Fe$ ratio, and vice versa. We explicitly stated this point in the initial version of our manuscript (please see also Figure 1A and new Lines 85-112). **Under the same mantle composition and $Fe^{3+}/\Sigma Fe$, variations in oxygen fugacity can exceed 1.5 log units at the P-T space of Figure 1A.**

Regarding your suggestion to adopt the approach of Li and Lee (2004) (i.e., directly using the V/Sc elemental ratio to infer oxygen fugacity), we would like to respectfully point out that while their study made an early contribution to the development of V-based oxybarometers, **their method is relatively simplified and ignores the great impact of temperature and pressure.** This approach of Li and Lee (2004) relies primarily on elemental ratios and does not incorporate thermodynamic constraints from experimental petrology (as also noted by Reviewer 1). The limitations of their method were already acknowledged in our initial submission: as shown in Figure 1B of our manuscript, a single fO_2 value can correspond to a wide range of V/Sc ratios under varying P–T conditions. **Your comments on lines 78-79 below** appear to agree with these thermodynamic effects like that from varying P–T conditions. If so, you would likely recognize the limitations in the methodology and findings proposed by Li and Lee (2004).

Subsequent studies, such as Wang et al. (2019), have built upon this early work **by developing experimentally calibrated models that more comprehensively capture P–T dependencies.** These refinements are further supported by observations from modern arc and MORB lavas. From our perspective, and that of more recent research (e.g., Wang et al., 2019; Aulbach and Stagno 2016; as well as your comments on Line 78-79 below), **the model proposed by Li and Lee (2004) does not account for the significant methodological and experimental progress made in this field over the past two decades.**

Additionally, the Archean dataset cited in Li and Lee (2004) appears not to have excluded other basalts like that in arc and other settings, which typically do not represent MORB mantle conditions. Although they may have considered using titanium content to track the degree of partial melting as a basis for samples comparable to MORB, this logic is not valid. The degree of partial melting cannot substitute for or represent tectonic setting. Moreover, most of the titanium contents of Archean basalts and modern MORB in their database do not overlap. Therefore, if a comparison is to be made, it should be with modern basalts in all settings, rather

than only with modern MORB. Moreover, they did not report the detailed ages of the Archean rocks, making it impossible to determine the specific evolutionary trend of V/Sc during the Archean. Therefore, for the purposes of establishing a redox baseline in our study, we believe that **more stringent sample screening is necessary**.

Finally, we respect your suggestion and have included a V/Sc evolution trend using the Li and Lee (2004)'s method in this reply. With the expansion and refinement of global geochemical databases in recent years, the updated sample set is broader and more representative (all data from Keller and Schoene, 2018, Liu et al, 2024a and b). We first selected global MORB and applied the same screening criteria as Li and Lee (2004) (45–54 wt% SiO₂, 8–12 wt% MgO, n = 494), yielding a V/Sc ratio of 7.2 ± 0.2 (see figure below). In addition, we compiled modern basalts from all tectonic settings within the same Si-Mg range (45–54 wt% SiO₂, 8–12 wt% MgO, n = 2278), which produced an average V/Sc of 8.3 ± 0.3 (see figure below). This highlights the necessity of considering the geological t tectonic settings.

A compilation of Archean basalts from all tectonic settings within the same Si-Mg range (45–54 wt% SiO₂, 8–12 wt% MgO; this method is same as that in Li and Lee, 2004; n = 393) shows that their V/Sc ratios evolved as follows: from 3.25 to 3 Ga, the average V/Sc was 5.5 ± 0.3 ; from 3 to 2.75 Ga, it increased to 6.5 ± 0.3 ; and from 2.75 to 2.5 Ga, it reached 7.1 ± 0.8 (see figure below). This suggests a general increase in V/Sc ratios of basalts throughout the Archean. The average V/Sc is still noticeably lower than that of modern basalts from all tectonic settings (8.3 ± 0.3).

As noted above, when applying the method of Li and Lee (2004)—which does not distinguish tectonic settings or consider thermodynamic effects (e.g., P, T, T_p), but focuses solely on constraints from MgO and SiO₂

contents—the V/Sc ratios of Archean basalts remain significantly lower than those of modern basalts (i.e., 8.3 ± 0.3), by approximately 2.8 to 1.2 units (see figure above). This suggests the overall increasing trend of fO_2 is robust even when using only V/Sc of basalts and aligns with findings from other studies (e.g., Nicklas et al., 2019; Aulbach and Stagno 2016). Therefore, basaltic V/Sc ratios and their directly derived fO_2 are not constant either from the Archean, which is consistent with the V/Sc trend of mafic rocks in Aulbach and Stagno (2016) (see figure above). However, the specific values of Archean basalts (from all tectonic settings) are somewhat higher, which is due to the use of the Li and Lee (2004) method without filtering for tectonic setting. In contrast, Aulbach and Stagno (2016) focused on MORB samples. This discrepancy between the two approaches is also observed in modern samples (please see the figure above; upper left area). This further highlights the methodological defficiency of this method.

Thus, we would like to reiterate that while elemental ratios may provide useful first-order insights, they are inherently limited in precision. Incorporating the newest experimentally calibrated thermodynamic constraints is essential for improving the accuracy and quantitative robustness of redox proxy models, which is why such models have been continuously refined in recent years. Furthermore, **we emphasize that our forward numerical simulations (e.g., Fig. 2) do not rely on any V-based oxybarometers**; yet they still allow cross-validation with empirical observations, which represents one of the key lines of supporting evidence in our study. This is the first time such a numerical modelling-based approach has been developed and applied.

2. If the increase of mantle $Fe^{3+}/\Sigma Fe$ at stage 4 (3.2-1.8 Ga) was caused by subduction of oxidized surficial materials into the deep mantle, then from the point of mass-balance, we should expect a decrease in O_2 of the atmosphere. However, in the contrary, the O_2 content of the atmosphere increased at this stage.

Answer: Thank you for your important comment. However, we think that the mass-balance reasoning you presented does not accurately reflect the geological and geochemical process that we wish to emphasize. In our model, formation of Fe^{3+} in the oceanic lithosphere is predominantly controlled by oceanic hydration reactions (e.g., serpentinization reactions), which produce Fe^{3+} and H_2 . Therefore, the oxidized materials (e.g., Fe^{3+}) in the ocean-hydrated rocks are not the result of reactions with the atmosphere. Significant part of the formed hydrogen can then escape from the atmosphere to outer space during the early Earth atmospheric evolution (Catling et al., 2001). The seafloor oxidized materials formed through oceanic hydration reactions subsequently accumulate in the oceanic lithosphere and remain isolated from the atmosphere by seawater. As a result, these oxidized materials neither consume atmospheric oxygen nor release oxygen to the atmosphere. Only when they

are subducted into the mantle—where redox transfer processes are activated by slab dehydration and mantle melting—can they potentially contribute to the formation of atmospheric oxygen. It is only after global subduction (During Stage 4: ~3.2–1.8 billion years ago), through processes like arcs volcanic degassing, these subducted oxidized oceanic rocks are capable to deliver redox budget to the mantle and then to the atmosphere by volcanic degassing. To avoid misunderstanding, we have elaborated on this logic further in the main text (e.g., Lines 393-427).

Our detailed logic is as follows:

We emphasize that the oxidized materials carried by the subducting oceanic lithosphere had already been fixed into rocks through hydration and alteration reactions while still on and below the seafloor (Fig. a). This is analogous to the serpentinization of modern mantle peridotite, during which Fe³⁺ is incorporated into mineral structures and hydrogen gas is released (see reaction in Fig. a; white marked). The oldest known serpentinized rocks date back to approximately 4.0 billion years ago (e.g., Sleep et al., 2011), indicating that the accumulation of oxidized components on the seafloor had already begun in the early Archean. Once fixed and isolated from the atmosphere by seawater, these oxidized materials ceased interacting with the atmosphere and remained stored in the lithosphere on geological timescales (Fig. a), until they were eventually subducted into the mantle. Therefore, the transport of such materials into the mantle would not result in a decrease in atmospheric oxygen.

In addition, during the Archean, Earth experienced high rates of hydrogen escape (the redox budget in lithosphere driven by hydrogen escape could reach $\sim 7 \times 10^{12}$ mol O₂/yr, Catling et al., 2001). Meanwhile, microbial activity—such as that of photosynthetic bacteria—continuously reacted with reducing gases. These processes may have contributed to the persistently low levels of atmospheric oxygen, as the oxygen produced

by photosynthesis was consumed by reducing gases generated through water–rock interactions.

During Stage 4 (3.2–1.8 billion years ago; Fig. b below), the globally subducted oxidized materials—including serpentinized peridotite or komatiite, altered oceanic crust, and iron-rich sediments such as banded iron formations (BIFs)—had already reached oxidation states prior to subduction, as supported by multiple lines of evidence (e.g., Tamblyn and Hermann, 2023; Sleep et al., 2011; Gard et al., 2019; Stolper and Keller, 2018; see also in figures from Comment 8 of Reviewer 1).

At the same time, the enhancement of mantle oxidation because of subduction during this stage promoted the generation of more oxidized magmas, such as the widespread occurrence of arc-related volcanic rocks (Fig. b) and increasingly oxidized mid-ocean ridge basalts. These magmas, originating from increasingly oxidized source regions, released more oxidized volcanic gases during degassing, thereby increasing the oxygen input to the atmosphere (Fig. b). Meanwhile, the evolution of continental crust enhanced chemical weathering, further providing oxidants to the atmosphere. These subduction-driven processes together brought new sources of oxygen to the atmosphere. Meanwhile, ongoing hydration reactions continuously locked Fe³⁺ into the lithosphere that had not yet been subducted (Fig. b). Once these rocks were eventually subducted, the atmosphere was supplied with a sustained source of oxygen at 3.2–1.8 billion years ago.

Therefore, the rise in atmospheric oxygen during this stage is not only consistent with subduction-driven mantle oxidation but was likely caused to a large extent by it. The logic we emphasize is: (1) in the Archean, oxidized materials accumulated in the lithosphere, and the associated reducing gases (H₂) were partly consumed and partly escaped to outer space, thereby locking a net redox budget within the lithosphere; (2) after the onset of global subduction, this net redox budget was transferred into the mantle, triggering mantle oxidation and

facilitating redox exchange with the atmosphere through increasingly oxidized magmatic degassing, ultimately contributing to atmospheric oxidation.

Although we have demonstrated the plausibility of subduction mechanism here, it should be noted that the discussion related to the subduction process is a supplemental interpretation based on previous studies (This supplemental explanation has been further refined; please see Lines 393-427; 434). It is intended to provide a plausible geological context for the main conclusion of this study—that the $\text{Fe}^{3+}/\sum\text{Fe}$ of the mantle has doubled—within the framework of existing knowledge. **This part of the discussion does not affect the robustness of the study's main conclusion concerning significant oxidation of the mantle through geological time.**

In addition, if the mantle was remoulded in $\text{Fe}^{3+}/\sum\text{Fe}$, the bulk composition of the mantle should also change due to subduction. That means the authors cannot use a constant mantle composition in the simulations.

Answer: We appreciate your important comment; however, we disagree with this line of reasoning. Furthermore, we did not assume a constant mantle composition in our simulations.

First, the increase in the $\text{Fe}^{3+}/\sum\text{Fe}$ ratio in the mantle occurs through electron transfer among redox-sensitive elements, like Fe^{3+} , rather than the addition of total iron mass. This process takes place during the subduction of slabs that carry an excess redox budget into the mantle (e.g., Evans, 2012). It neither requires nor results in significant changes to the major element composition of the mantle (except the oxygen content and the $\text{Fe}^{3+}/\sum\text{Fe}$ ratio). Considering the cyclic nature of plate tectonics and subduction (a Wilson cycle of ~200 million years; during this time, crustal generation and subduction were approximately in balance), it is highly unlikely that subducted materials could significantly alter the mantle's major element composition (for example, total iron). In this context, the increase in the mantle's redox budget is achieved through charge transfer and the periodic subduction cycles.

This conclusion is also supported by observational data: the major element compositions of the modern depleted MORB mantle and the primitive upper mantle are remarkably similar (Workman and Hart, 2005; McDonough and Sun, 1995; e.g., both mantle compositions include ~8 wt% $\text{FeO}_{\text{total}}$). This shows that even after over a billion years of subduction, the mantle composition has not significantly changed in the way suggested in the comment.

We performed an additional simulation to address your comment (see also Lines 590-593; new Fig. S8), which explicitly demonstrates that the transition from primitive to depleted mantle has only a minimal effect

(<0.005) on the basaltic melt $\text{Fe}^{3+}/\Sigma\text{Fe}$ ratio (please see the figure below: the white isopleths show the melt $\text{Fe}^{3+}/\Sigma\text{Fe}$ trend in equilibrium with the primitive upper mantle, assuming the same melt $\text{Fe}^{3+}/\Sigma\text{Fe}$ in the depleted MORB mantle indicated by different colors. The melt $\text{Fe}^{3+}/\Sigma\text{Fe}$ trends of two mantle composition are nearly indistinguishable along with the evolution of P–T conditions, indicating that whether primitive mantle or depleted MORB mantle, as whole-rock compositions, do not significantly affect the estimation of the melt $\text{Fe}^{3+}/\Sigma\text{Fe}$ ratio.

Finally, we reiterate that we did not use a constant mantle composition in our modeling. **Our simulations are based on mass balance within an open-system framework**, as described in detail in the Methods section.

3. The estimated mantle $\text{Fe}^{3+}/\Sigma\text{Fe}$ has a large uncertainty, which may lead to misleading rather than clarity.

Answer: Thank you for your comment although it is unfortunately not detailed which uncertainty you mean. It remains unclear whether you are referring to the uncertainty associated with individual data or simulation points, or to the overall uncertainty of our findings, or both. It is also unclear whether the concern pertains to the empirical dataset or the modeling framework or to both of them. Therefore, our answer and clarifications to this short comment will be sufficiently lengthy.

First, regarding the empirical data: if your comment refers to the uncertainty of individual samples, we note that the associated uncertainties in $\text{Fe}^{3+}/\Sigma\text{Fe}$ are directly tied to those of the corresponding oxygen fugacity ($f\text{O}_2$). These uncertainties are primarily governed by the pressure–temperature (P–T) conditions used in the calculations, which can be effectively constrained—if not eliminated—through thermodynamic bracketing criteria. As such, when individual data points are well-constrained, the overall uncertainty of the $f\text{O}_2$ dataset is comparable to, and no greater than, that reported in prior studies, as we have also demonstrated in the manuscript.

Second, in terms of the simulations: once a specific P–T path is defined, the resulting geochemical composition is deterministically calculated (on the basis of experimentally calibrated thermodynamic parameters), and thus the concept of uncertainty in individual data points does not apply. The relevant uncertainties instead stem from differences in thermodynamic or geodynamic parameterizations across model scenarios. These are well-understood sources of variation that we have already summarized in tabular form in the manuscript, and their impact on the results is limited.

Notably, Reviewer 1 has already pointed out that our calculated $\text{Fe}^{3+}/\Sigma\text{Fe}$ values for Archean basaltic rocks are consistent with their previous studies (Aulbach et al., 2019; please see also the comments for Line 139 of Reviewer 1). This also suggests that our calculations are robust and do not exhibit the large uncertainties you mentioned. For modern samples, we also compared our results with the observed $\text{Fe}^{3+}/\Sigma\text{Fe}$ values of present-day fresh MORB basalts (Lines 207-209), and the agreement further confirms that our thermodynamic forward and inverse calculations are reliable. Moreover, the mantle $\text{Fe}^{3+}/\Sigma\text{Fe}$ values we inversely calculated from modern mantle samples fall perfectly within the range of previously reported observational data (Canil et al., 1994; Davis and Cottrell, 2021; Sossi et al., 2020). Our thermodynamic simulations also show good agreement with experimental petrology results (new Fig. S9; Davis and Cottrell, 2018). Finally, the thermodynamic methods we used have been validated by numerous previous studies, many of which have been published in high-impact journals such as *Nature* (e.g., Walsh et al., 2023; Ge et al., 2023) and *Nature Geoscience* (e.g., Hernández-Uribe, 2024). Therefore, the mantle $\text{Fe}^{3+}/\Sigma\text{Fe}$ values derived using the same consistent and robust methodology should not suffer from the issue you raised either.

Given these considerations, we are unsure what specific aspect prompted this concern, as neither the observational dataset nor the simulation results exhibit unusually large or unaddressed uncertainties. Based on this, we are confident that our study does not present any risk of drawing misleading conclusions.

Finally, while not related to computational uncertainty per se, we would like to acknowledge that our dataset—as is also the case with most others in this field—contains fewer data points for certain geological periods. This reflects reduced tectonic activity and sampling biases during those intervals. To address this, we first conducted a sensitivity test on the redox trend with respect to sample age, which demonstrates that our observed trend and conclusions are robust.

We conducted two sets of sensitivity tests. The first set used tectonic regimes—episodic mobile lid and stagnant lid—as temporal boundaries, dividing the data into four intervals: Early to Middle Archean (mantle $\text{Fe}^{3+}/\Sigma\text{Fe}=0.02 \pm 0.002$; $n = 19$), Late Archean to Paleoproterozoic (0.025 ± 0.002 ; $n = 39$; specifically, 0.022 for the Neoproterozoic [$n = 24$] and 0.028 for the Paleoproterozoic [$n = 15$]), and Mesoproterozoic (0.02 ± 0.006 ; $n = 10$). This binning strategy increases the number of samples per group. The results show that the overall trend remains consistent with that derived from the original 250 Ma interval-based classification.

The second set of tests divided the data more broadly into: Archean (mantle $\text{Fe}^{3+}/\Sigma\text{Fe} = 0.021 \pm 0.002$; $n = 43$), Proterozoic (0.025 ± 0.003 ; $n = 28$), and compare them with the modern era. Although this binning masks some finer details, it still clearly captures the overall doubling trend in mantle $\text{Fe}^{3+}/\Sigma\text{Fe}$ from the Archean.

We would also emphasize that data sparsity in specific intervals is a recognized and widespread challenge in studies of mantle redox evolution, as well as in broader research on Earth's tectonic and thermal history (e.g., Herzberg et al. 2010; Zhang et al., 2024; Aulbach and Stagno, 2016; Aulbach and Arnd 2019). And also, as noted in both the original and revised versions of the manuscript, this limitation warrants a degree of caution when interpreting trends over such timescales. Within the context of currently available data, we do not believe this constitutes a technical or conceptual flaw in our work, nor should it be considered grounds for limiting further investigation.

Importantly, this limitation does not affect the main conclusion of our study: that the redox state of the modern mantle—as expressed by $\text{Fe}^{3+}/\Sigma\text{Fe}$ —has approximately doubled relative to that of the Archean mantle. This conclusion is underpinned by a substantial and robust whole-Archean dataset and remains well supported.

Minor points:

The title is pompous, the word “oxidation degree” is unclear. If the authors mean “ $\text{Fe}^{3+}/\Sigma\text{Fe}$ ratio, they should

use $\text{Fe}^{3+}/\Sigma\text{Fe}$.

Answer: We have revised the title in accordance with your suggestion.

Lines 11-13 The expression is vague, how mantle's redox properties enables cycling of redox sensitive materials? Or the redox state of the mantle regulated by material cycling? What's the definition of "redox-sensitive materials"

Answer: We have revised the wording. What we intended to express is that the redox state of the mantle is regulated by the cycling of these redox-sensitive materials.

Lines 33-34: Yes, the mass of the mantle is large, however, the MORBs are derived from the very shallow part of the mantle, how could the authors use MORB represent the whole mantle?

Answer: First, we acknowledge that characterizing the entire mantle based on direct observations is extremely difficult due to the lack of direct records, and currently no study is capable of fully achieving this. However, many relevant studies still use terms such as "mantle" or "upper mantle" in their titles, despite facing the same limitations. For example, the recent study by Zhang et al., (2024) is titled "The constant oxidation state of Earth's mantle since the Hadean", even though it does not characterize the entire mantle. We use the term "mantle" to enable a direct and meaningful comparison with such works.

Similarly, the study by Li and Lee (2004), which you recommended, also investigates basalts, yet their title refers to the "upper mantle". According to this comment, their study would also not fully represent the whole upper mantle, yet the term remains widely accepted in such contexts. Of course, we could choose to calculate the redox state of komatiite mantle sources to extend the depth of investigation to ~8 GPa. However, even in doing so, it would still not be possible to reliably infer the redox state of the entire mantle based solely on such observational data. Moreover, the pressure–temperature estimates for komatiites generally have much larger uncertainties compared to those of basaltic rocks, which would, in turn, result in significantly higher uncertainties in the calculated oxygen fugacity. For these reasons, we did not include such rocks in our calculations.

Second, our use of "mantle" is supported by prior experimental petrology, geophysical observations, and numerical simulations. Modern geophysical evidence supports whole-mantle convection, and over geological time scales, this long-term convection likely leads to approximate large-scale chemical homogenization—although we recognize that anomalous domains (e.g., OIB source regions) do exist. The relatively stable redox state observed in modern MORB sources further supports the view of a broadly homogeneous mantle.

In this context, from the perspective of the mantle's redox budget, MORB source regions can serve as a

reasonable proxy for the oxidation state of the mantle as a whole. For the Archean mantle, we acknowledge that the redox state of the lower mantle remains uncertain, and we have addressed this explicitly in the discussion section.

Lastly, we have also emphasized in the manuscript that **our study focuses specifically on the redox state of MORB-related mantle source regions**—because they are the most accessible through observation and are also the most debated. We believe it is important to resolve this issue as a foundational step.

Line 39-41: It's very abrupt to compare the redox state of Earth with other terrestrial bodies, what's the point here?

Answer: It has been revised.

Line 62: change “the mantle and derived-melt redox state” to “the redox state of the mantle and mantle-derived melts”

Answer: It has been revised.

Line 70: According to which study, the authors alleged “The $\text{Fe}^{3+}/\text{Fe}^{\text{tot}}$ in melts decreases with increasing temperature”?

Answer: This conclusion is based on the results of our own study (Figure 1A). To the best of our knowledge, this is the first redox thermodynamic modeling of a mantle system incorporating Fe^{3+} -bearing silicate melts. Therefore, this represents one of the novel aspects of our work.

Line 78-79: That's also the reason why Aulbach and Stagno (2016) normalized the V/Sc ratio to constant P of 1 GPa.

Answer: Reviewer 1 is Dr. Sonja Aulbach, who pointed out that applying a constant pressure of 1 GPa to V/Sc ratios is invalid—this is precisely the issue we aim to address in this study. Please refer to her comments on lines 58–61 and 144–146 for details. Given that more extrapolated results—though based on different methods—have recently been published (e.g., Zhang et al., 2024), we believe that this further highlights the necessity of our work. Our study can help correct these methodological misconceptions, which have led to some incorrect conclusions, in a timely manner.

Finally, we are truly grateful for your feedback on our manuscript.

References

Aulbach, S., & Arndt, N. T. (2019). Eclogites as palaeodynamic archives: evidence for warm (not hot) and depleted (but heterogeneous) Archean ambient mantle. *Earth and Planetary Science Letters*, 505, 162-172.

- Aulbach, S., & Stagno, V. (2016). Evidence for a reducing Archean ambient mantle and its effects on the carbon cycle. *Geology*, *44*(9), 751-754.
- Aulbach, S., Woodland, A. B., Stern, R. A., Vasilyev, P., Heaman, L. M., & Viljoen, K. S. (2019). Evidence for a dominantly reducing Archaean ambient mantle from two redox proxies, and low oxygen fugacity of deeply subducted oceanic crust. *Scientific Reports*, *9*(1), 20190.
- Boukaré, C. É., Badro, J., & Samuel, H. (2025). Solidification of Earth's mantle led inevitably to a basal magma ocean. *Nature*, 1-6.
- Canil, D., O'Neill, H. S. C., Pearson, D. G., Rudnick, R. L., McDonough, W. F., & Carswell, D. A. (1994). Ferric iron in peridotites and mantle oxidation states. *Earth and Planetary Science Letters*, *123*(1-3), 205-220.
- Catling, D. C., Zahnle, K. J., & McKay, C. P. (2001). Biogenic methane, hydrogen escape, and the irreversible oxidation of early Earth. *Science*, *293*(5531), 839-843.
- Davis, F. A., & Cottrell, E. (2018). Experimental investigation of basalt and peridotite oxybarometers: Implications for spinel thermodynamic models and Fe³⁺ compatibility during generation of upper mantle melts. *American Mineralogist*, *103*(7), 1056-1067.
- Davis, F. A., & Cottrell, E. (2021). Partitioning of Fe₂O₃ in peridotite partial melting experiments over a range of oxygen fugacities elucidates ferric iron systematics in mid-ocean ridge basalts and ferric iron content of the upper mantle. *Contributions to Mineralogy and Petrology*, *176*(9), 67.
- Duan, W., Connolly, J., van Keken, P., Gerya, T., Schertl, H. P., Li, S., (2024). Mantle oxidation controlled by redox dynamics of Mariana-type subduction settings. <https://www.researchsquare.com/article/rs-3936877/v1>
- Evans, K. A. (2006). Redox decoupling and redox budgets: Conceptual tools for the study of earth systems. *Geology*, *34*(6), 489-492.
- Evans, K. A. (2012). The redox budget of subduction zones. *Earth-Science Reviews*, *113*(1-2), 11-32.
- Fischer-Gödde, M., Elfers, B. M., Münker, C., Szilas, K., Maier, W. D., Messling, N., ... & Smithies, H. (2020). Ruthenium isotope vestige of Earth's pre-late-veener mantle preserved in Archaean rocks. *Nature*, *579*(7798), 240-244.
- Gao, L., Liu, S., Cawood, P. A., Hu, F., Wang, J., Sun, G., & Hu, Y. (2022). Oxidation of Archean upper mantle caused by crustal recycling. *Nature Communications*, *13*(1), 3283.
- Gard, M., Hasterok, D., & Halpin, J. A. (2019). Global whole-rock geochemical database compilation. *Earth System Science Data*, *11*(4), 1553-1566.
- Ge, R. F., Wilde, S. A., Zhu, W. B., & Wang, X. L. (2023). Earth's early continental crust formed from wet and oxidizing arc magmas. *Nature*, *623*(7986), 334-339.
- Hastie, A. R., & Kerr, A. C. (2010). Mantle plume or slab window?: Physical and geochemical constraints on the origin of the Caribbean oceanic plateau. *Earth-Science Reviews*, *98*(3-4), 283-293.
- Hernández-Urbe, D. (2024). Generation of Archaean oxidizing and wet magmas from mafic crustal overthickening. *Nature Geoscience*, *17*(8), 809-813.
- Herzberg, C., Condie, K., & Korenaga, J. (2010). Thermal history of the Earth and its petrological expression. *Earth and Planetary Science Letters*, *292*(1-2), 79-88.
- Keller, B., & Schoene, B. (2018). Plate tectonics and continental basaltic geochemistry throughout Earth history. *Earth and Planetary Science Letters*, *481*, 290-304.
- Kress, V. C., & Carmichael, I. S. (1991). The compressibility of silicate liquids containing Fe₂O₃ and the effect of composition, temperature, oxygen fugacity and pressure on their redox states. *Contributions to Mineralogy and Petrology*, *108*(1), 82-92.
- Lee, C. T. A., Luffi, P., Plank, T., Dalton, H., & Leeman, W. P. (2009). Constraints on the depths and temperatures of basaltic magma generation on Earth and other terrestrial planets using new thermobarometers for mafic magmas. *Earth and Planetary Science Letters*, *279*(1-2), 20-33.

- Li, Z. X. A., & Lee, C. T. A. (2004). The constancy of upper mantle fO₂ through time inferred from V/Sc ratios in basalts. *Earth and Planetary Science Letters*, 228(3-4), 483-493.
- Liu, C. T., Liu, X. M., & ZhangZhou, J. (2024a). Data-driven investigation reveals subaerial proportion of basalts since the early Archean. *Geophysical Research Letters*, 51(12), e2023GL107066.
- Liu, C. T., Ye, C. Y., & ZhangZhou, J. (2024b). Secular changes in the occurrence of subduction during the Archean. *Geophysical Research Letters*, 51(9), e2023GL107996.
- Maffei, A., Frezzotti, M. L., Connolly, J. A. D., Castelli, D., & Ferrando, S. (2024). Sulfur disproportionation in deep COHS slab fluids drives mantle wedge oxidation. *Science Advances*, 10(12), eadj2770.
- McDonough, W. F., & Sun, S. S. (1995). The composition of the Earth. *Chemical geology*, 120(3-4), 223-253.
- Messling, N., Willbold, M., Kallas, L., Elliott, T., Fitton, J. G., Müller, T., & Geist, D. (2025). Ru and W isotope systematics in ocean island basalts reveals core leakage. *Nature*, 1-5.
- Mundl, A., Touboul, M., Jackson, M. G., Day, J. M., Kurz, M. D., Lekic, V., ... & Walker, R. J. (2017). Tungsten-182 heterogeneity in modern ocean island basalts. *Science*, 356(6333), 66-69.
- Nicklas, R. W., Day, J. M., Trumbull, R. B., Rangwalla, H., & Kelly, S. (2024). Continental flood basalts sample oxidized mantle sources. *Lithos*, 482, 107697.
- Nicklas, R. W., Puchtel, I. S., Ash, R. D., Piccoli, P. M., Hanski, E., Nisbet, E. G., ... & Anbar, A. D. (2019). Secular mantle oxidation across the Archean-Proterozoic boundary: Evidence from V partitioning in komatiites and picrites. *Geochimica et Cosmochimica Acta*, 250, 49-75.
- Pan, F., Wu, X., Wang, C., Zhang, Y., Yang, Y., He, X., ... & Zhang, J. (2025). Iron disproportionation in peridotite fragments from the mantle transition zone. *Nature Communications*, 16(1), 5440.
- Perchuk, A. L., Gerya, T. V., Zakharov, V. S., & Griffin, W. L. (2020). Building cratonic keels in Precambrian plate tectonics. *Nature*, 586(7829), 395-401.
- Sleep, N. H., Bird, D. K., & Pope, E. C. (2011). Serpentinite and the dawn of life. *Philosophical Transactions of the Royal Society B: Biological Sciences*, 366(1580), 2857-2869.
- Sossi, P. A., Burnham, A. D., Badro, J., Lanzirotti, A., Newville, M., & O'Neill, H. S. C. (2020). Redox state of Earth's magma ocean and its Venus-like early atmosphere. *Science advances*, 6(48), eabd1387.
- Stixrude, L., & Lithgow-Bertelloni, C. (2024). Thermodynamics of mantle minerals—III: the role of iron. *Geophysical Journal International*, 237(3), 1699-1733.
- Stolper, D. A., & Keller, C. B. (2018). A record of deep-ocean dissolved O₂ from the oxidation state of iron in submarine basalts. *Nature*, 553(7688), 323-327.
- Stolper, E. M., Shorttle, O., Antoshechkina, P. M., & Asimow, P. D. (2020). The effects of solid-solid phase equilibria on the oxygen fugacity of the upper mantle. *American Mineralogist: Journal of Earth and Planetary Materials*, 105(10), 1445-1471.
- Tamblyn, R., & Hermann, J. (2023). Geological evidence for high H₂ production from komatiites in the Archean. *Nature Geoscience*, 16(12), 1194-1199.
- Tamblyn, R., Hermann, J., Hasterok, D., Sossi, P., Pettke, T., & Chatterjee, S. (2023). Hydrated komatiites as a source of water for TTG formation in the Archean. *Earth and Planetary Science Letters*, 603, 117982.
- Tomkins, A. G., & Evans, K. A. (2015). Separate zones of sulfate and sulfide release from subducted mafic oceanic crust. *Earth and Planetary Science Letters*, 428, 73-83.
- van Hunen, J., & van den Berg, A. P. (2008). Plate tectonics on the early Earth: limitations imposed by strength and buoyancy of subducted lithosphere. *Lithos*, 103(1-2), 217-235.
- Walsh, C., Kamber, B. S., & Tomlinson, E. L. (2023). Deep, ultra-hot-melting residues as cradles of mantle diamond. *Nature*, 615(7952), 450-454.
- Walters, J. B., Cruz-Urbe, A. M., & Marschall, H. R. (2020). Sulfur loss from subducted altered oceanic crust and

implications for mantle oxidation. *Geochemical Perspectives Letters*.

Wang, J., Xiong, X., Takahashi, E., Zhang, L., Li, L., & Liu, X. (2019). Oxidation state of arc mantle revealed by partitioning of V, Sc, and Ti between mantle minerals and basaltic melts. *Journal of Geophysical Research: Solid Earth*, *124*(5), 4617-4638.

Workman, R. K., & Hart, S. R. (2005). Major and trace element composition of the depleted MORB mantle (DMM). *Earth and Planetary Science Letters*, *231*(1-2), 53-72.

Zhang, F., Lai, S., Stagno, V., Chen, L., Zhang, C., Zhu, R., ... & Wang, J. (2024). The Redox state of the asthenospheric mantle and the onset of melting beneath mid-ocean ridges. *Journal of Geophysical Research: Solid Earth*, *129*(5), e2023JB027033.

Responses to Reviewers' Comments Point by Point

REVIEWER COMMENTS

Reviewer #1 (Remarks to the Author):

I have read the authors' rebuttal letter and the manuscript version with tracked changes. I am impressed with the detailed, careful and mostly well-argued responses they have given. Although I don't agree with all of the authors' reasonings, this concerns fine details that have no significant influence on the main outcomes of the study, which is novel and robust as far as I can tell.

Here, I first discuss remaining instances where I am not in complete agreement with the authors with respect to their rebuttal, the consideration of some of which is at the authors' discretion. Then, I make detailed comments on the revised text, many of which do require the authors' attention.

Answer: We sincerely thank you for the careful reading of our revised manuscript and the rebuttal letter, and for the constructive comments and thoughtful suggestions. We have carefully considered all remaining points raised and have revised the manuscript accordingly, as detailed below.

1. For example, to my main point #4, they say that "although at greater depths, carbon may exist as C^0 . Upon ascent, the transformation of C^0 to C^{4+} would consume Fe^{3+} (i.e., cause reduction), but the overall redox budget remains conserved in MORB mantle source". I think this is only true so long as mantle remains subsolidus, but if you generate CO_2 , then you have redox melting, in which case the redox budget in the residue should decrease, CO_2 being basically perfectly incompatible, Fe^{3+} mildly incompatible.

Answer: We agree that when the temperature is above the solidus, the conversion of C^0 to C^{4+} accompanied by degassing removes part of the rock's redox budget. However, please note that our discussion concerns the initial redox state and budget of mantle rocks, rather than that of the residues. The initial state implies that post-melting processes such as degassing have not yet occurred. Moreover, the conversion of C^0 to C^{4+} occurs in both the modern and Archean mantle, which means that this process does not affect the comparison between MORB mantle sources from these two periods. To avoid introducing excessive uncertainty in the discussion, we have removed the related statements.

2. They further say a few lines later that “oxygen fugacity from QFM-2 to 0 likely reflect changes in the oxidation states of both Fe and S”, whereas I would say that there is no appreciable S⁴ at those conditions.

Answer: We agree with you. As we mentioned in our initial response and in the manuscript (Lines 118–132), in the typical MORB-source mantle, the valence states of several multivalent elements are generally considered to be constant, including sulfur, which occurs predominantly as S²⁻ (Evans, 2006; 2012). In addition, we also noted that variations in the mantle redox state are mainly controlled by changes in Fe³⁺, whereas the concentrations of other redox-sensitive elements such as S and C are very low.

3. In their reply to my comment #8, they write that “sulphide minerals (e.g., FeS_x) within subducted lithologies can be oxidized to sulphate in situ via internal redox reactions, driven by the reduction of Fe³⁺ within the rock (S²⁻ + 8Fe³⁺ = S⁶⁺ + 8Fe²⁺).” I only know of iron-reducing sulphate-producing bacteria. Where do you have fO₂ favouring appreciable S⁶⁺ over Fe³⁺ in subduction zones? I can envision that highly oxidised surficial material (e.g. sediments) releases oxidising fluids and S⁶⁺ upon subduction that are then reduced by the mantle (via oxidation primarily of Fe), but not that the reaction is shifted to the right. E.g. Tomkins and Evans 2015 EPSL model early anhydrite breakdown and later pyrite breakdown releasing H₂S, while Maffei et al. 2024 SciAdv suggest that pyrite might disproportionate to generate also sulphate, using modelling relevant to metacarbonate sediments.

Answer: We appreciate your insightful comment. As noted, our calculations are based on a conservative redox budget, excluding highly oxidized lithologies such as surficial sediments (e.g., Ague et al., 2022, NG). Our aim was to evaluate whether even moderately oxidized components could, in principle, oxidize the mantle to its present-day state.

During subduction, dehydration of serpentinites can release oxidizing fluids that create localized oxidizing conditions in the slab or mantle wedge. Under such conditions (around FMQ to FMQ+2), sulfur can exist in mixed valence states (S²⁻ and S⁶⁺ together), as shown by Jugo et al. (2005, GCA) and Nash et al. (2019, EPSL). The reaction S²⁻ + 8Fe³⁺ = S⁶⁺ + 8Fe²⁺ shows that sulphide oxidation to sulphate can also occur through internal redox exchange, not necessarily requiring external or biological oxidants.

We agree that pyrite and anhydrite in sediments follow different breakdown paths (e.g., Tomkins & Evans, 2015; Maffei et al., 2024), but these phases occur mainly in surface-derived sediments, whose total redox contribution is much smaller than that of Fe³⁺-bearing serpentinites.

Importantly, even if this internal S²⁻ to S⁶⁺ conversion was limited, the unreacted Fe³⁺ would still be

transported into the mantle, further supporting our conclusion that subduction can progressively oxidize the mantle over time. Therefore, our overall redox budget and conclusions remain unaffected.

4. Reply to my comment to L271-274: The authors “We are not sure whether the $\text{Fe}^{3+}/\Sigma\text{Fe}$ values in the cratonic eclogites mentioned here refer to the original magma composition or the post-alteration state ... it is worth noting that eclogites likely underwent dehydration. Assuming the altered rocks originally had $\text{Fe}^{3+}/\Sigma\text{Fe}$ ratios of 0.2–0.3 ... This implies that the RB remaining in exhumed eclogites may not reflect the total redox budget acquired during seafloor alteration prior to subduction.” Sure, but dehydration and subduction-related reduction of the redox budget would also apply to the ancient oceanic crust that the authors infer subducted and increased the mantle redox.

Answer: We agree that dehydration and the associated reduction of the redox budget occur progressively during subduction. However, regardless of these processes, the redox budget acquired at the Earth’s surface is indeed transferred to the mantle through subduction and subsequent dehydration (and melting) processes. Whether this transfer occurs in the shallow or deep mantle, the overall trend is that the mantle’s redox budget is continually increasing due to ongoing subduction and surface oxidation.

5. Reply to Reviewer 3 comment on L78-79: I did not point out that applying a constant pressure of 1 GPa to V/Sc ratios is invalid, but that the V/Sc-derived $f\text{O}_2$ must be corrected to a common pressure (we chose 1 GPa) in order to make valid comparisons between melts generated at different average pressures, because differences in $f\text{O}_2$ may arise from the “intrinsic” (at constant bulk $\text{Fe}^{3+}/\Sigma\text{Fe}$) change in mantle $f\text{O}_2$ as a function of pressure alone. Our correction (assuming an increase of $f\text{O}_2$ by 0.4 orders of magnitude per GPa) was wrong because the complex redox profile of the mantle below 3 GPa was not known in 2016, but that’s a different issue.

Answer: We thank you for the clarification and have accordingly revised our response. Your method of correcting oxygen fugacity (including that of Zhang et al., 2024, Nature Communications) is indeed reasonable; however, as you pointed out, it requires accurate thermodynamic calculations, and applying a constant correction coefficient may lead to inaccurate results. Specifically, in theory, it is possible to use corrected oxygen fugacity to characterize the redox state, provided that the pressure–temperature correction coefficients are sufficiently accurate. However, this approach requires a series of thermodynamic back-calculations for each sample or modeling point. While this may be feasible for natural samples, it becomes significantly more complex when dealing with tens of thousands of modeling points.

6. Regarding a top-down oxidation of the mantle:

I am still not convinced that subduction of positive redox budget could have brought about the observed increase in mantle fO_2 , for the same reason I explained in the original review (also given by Reviewer 3). Here, the authors argue that “seafloor oxidized materials formed through oceanic hydration reactions subsequently accumulate in the oceanic lithosphere and remain isolated from the atmosphere by seawater”. However, the ocean-atmosphere-uppermost crust form part of the exosphere and in reservoir modelling, also those concerned with volatiles including oxygen, and are considered as an entity (e.g. Hirschmann 23 EPSL) that is interacting with the mantle on geological timescales. Taking redox budget out of the deep ocean water therefore means removing redox budget from the exosphere, thereby acting against oxygenation of the atmosphere. To then argue that “after global subduction (During Stage 4: ~3.2–1.8 billion years ago), through processes like arcs volcanic degassing, these subducted oxidized oceanic rocks are capable to deliver redox budget to the mantle and then to the atmosphere by volcanic degassing” illustrates that subduction of positive redox budget generates a short-circuit that diverts at least a fraction of it back to the atmosphere.

Answer: We agree that, in principle, the subduction of oxidized materials would indeed decrease the overall redox budget of the so-call exosphere. However, we argue that changes in the atmospheric redox budget cannot be simply equated with changes in the overall redox budget of the exosphere system (atmosphere–ocean–oceanic crust). The redox evolution of the exosphere and that of the atmosphere are not linearly correlated. A decrease in the exospheric redox budget does not necessarily imply an immediate or direct decline in atmospheric O_2 levels. If the logic were applied straightforwardly, one would expect that during tectonically quiescent periods (e.g., the early Archean and mid-Proterozoic), as oxidized materials continued to accumulate in the altered oceanic crust, the atmosphere should have become progressively more oxidized. However, geological evidence does not support such a trend. Conversely, during periods of low atmospheric oxygen, the oceanic crust should theoretically have been highly reduced, yet abundant BIFs, serpentinites, and altered oceanic crusts with moderate to high oxygen fugacities were still formed.

You suggested that the atmosphere–ocean–oceanic crust should be treated as an integrated and co-evolving system. From a geological perspective, a tectonic mechanism is required to transfer Fe^{3+} from oxidized seafloor rocks to the atmosphere. During tectonically quiescent periods, such a mechanism is difficult to envisage, because seawater acts as an effective insulating layer, isolating most of the altered oceanic crust from the atmosphere. The oxidized materials (Fe^{3+}) formed through seafloor hydration reactions (e.g., serpentinization)

remain largely isolated from the atmosphere while residing on the ocean floor. Only after the onset of global subduction could these oxidized materials be transported into the mantle wedge, where redox exchange and arc magmatism facilitate the release of the redox budget back to the atmosphere.

You further pointed out that subduction may create a “short circuit,” allowing part of the positive redox budget to return to the atmosphere through arc volcanism. We do not consider this to be a complete short circuit but rather a kind of “leakage” process — in which subduction-related arc volcanic degassing serves as a diversion pathway for the redox budget: (1) a portion is released to the atmosphere, and (2) another portion is transferred into the mantle, thereby modifying its redox state. In this sense, subduction-related arc volcanism represents a plausible tectonic mechanism linking altered seafloor rocks to the atmosphere. Therefore, although subduction may transiently redistribute the redox budget within the exosphere–mantle system, arc volcanic degassing provides an effective pathway for oxidized materials to return to the atmosphere. From this perspective, subduction-driven degassing could represent a key mechanism connecting the oxidation of the mantle and the atmosphere, consistent with the geological record of rising atmospheric O₂ between 3.2 and 1.8 billion years ago.

If the subduction mechanism worked, the question also arises as to why the mantle fO_2 has started to increase before 3.2 Ga (in the single-lid phase according to Fig. 5) and why it has not continually increased after the onset of global subduction and before the Proterozoic tectonic lull. Instead, the $Fe^{3+}/\Sigma Fe$ evolution ends several 100 Ma before the tectonic lull. I suggest this shows that after the transition to plate tectonics the exosphere and mantle are in some type of equilibrium on geologic scales, with the subducted redox budget being more or less balanced by release of redox budget via magmatism.

Answer: Please note that following the suggestions from you and Reviewer 2, we conducted geochemical screening optimization, and the relevant datasets and figures were updated in the first round of revisions. The mantle was not oxidized before 3.2 Ga, and the available data prior to 3.2 Ga are extremely limited. Therefore, particular caution should be exercised when evaluating whether the mantle redox state increased during that time, and we have explicitly noted this in the manuscript. In contrast, the dataset between 3.2 and 1.8 Ga is sufficiently large to reveal a clear trend: the $Fe^{3+}/\Sigma Fe$ ratio begins to rise between 3.2 and 3.0 Ga, which coincides closely with the onset of global subduction. This upward trend continues through the period of active plate tectonics until approximately 1.8 Ga, after which the record becomes sparse—likely due to reduced sampling or the decline in tectonic activity during the Proterozoic tectonic lull. Hence, the timing of global

subduction aligns remarkably well with the observed $\text{Fe}^{3+}/\Sigma\text{Fe}$ evolution, suggesting that subduction played a critical and indispensable role in contributing to the mantle redox budget.

Moreover, we argue that not all subducted redox budgets are immediately released back to the surface through arc magmatism. Fluid release from subducting slabs is not entirely completed beneath arcs; only the early-released oxidized fluids are recycled to the surface, while the remaining oxidized components may be retained within the mantle wedge or continue to be transported into the deeper mantle together with the subducting slab and associated crustal or sedimentary materials. Therefore, subduction should not be regarded as a process that “short-circuits” all positive redox budgets back to the atmosphere. The occurrence of highly oxidized arc mantle domains and metasomatized peridotites indicates that a significant portion of subducted oxidants can be stored in the mantle in solid form rather than being fully balanced by volcanic release. This scenario is inconsistent with a perfectly steady-state or completely short-circuited system.

It should be noted that the increase in the mantle’s $\text{Fe}^{3+}/\Sigma\text{Fe}$ also carries petrologic consequences, as investigated by Asimow (2022 in Book: Magma Redox Geochemistry) with respect to the argument on the modern MORB $\text{Fe}^{3+}/\Sigma\text{Fe}$: “Oxidizing conditions predict cold, low-MgO primary aggregate magmas that have difficulty crystallizing the most magnesian olivine phenocryst compositions found in MORB, that imply potential temperatures too cold to generate the traditionally assumed typical thickness of oceanic crust, and that are so close to the erupted basalt composition that it becomes difficult to explain the origin of a thick, cumulate lower crust as the complementary product of fractional crystallization”.

Answer: We appreciate your thoughtful comment. The study by Asimow (2021, in Magma Redox Geochemistry) focused specifically on high-MgO MORB samples from the Siqueiros Transform region in the western Pacific, which represent a particular subset of MORB compositions that are closest to primary melts, rather than the global MORB average. Therefore, the petrologic implications discussed in that study cannot be directly extended to the global mantle or to other geological periods.

Asimow (2021) summarized the wide range of previously reported $\text{Fe}^{2+}/\Sigma\text{Fe}$ (or $\text{Fe}^{3+}/\Sigma\text{Fe}$) ratios in MORB—from ~0.86 to 0.93—and emphasized the substantial analytical and interpretational uncertainties associated with determining these ratios. These variations strongly influence the inferred mantle potential temperature, degrees of partial melting, and the formation and thickness of oceanic crust. This recognition is precisely why we emphasize the importance of the $\text{Fe}^{3+}/\Sigma\text{Fe}$ ratio in our study.

In our forward modeling, we adopted an Archean mantle $\text{Fe}^{3+}/\Sigma\text{Fe}$ value of 0.02 and tested multiple

conditions before using a modern mantle value of 0.04. The resulting simulated MORB $\text{Fe}^{3+}/\Sigma\text{Fe}$ ratio of ~ 0.10 – 0.11 is fully consistent with Asimow's conclusion that $\text{Fe}^{2+}/\Sigma\text{Fe}$ values lower than ~ 0.88 correspond to 6–7 km-thick oceanic crust. It is important to note that our approach infers $\text{Fe}^{3+}/\Sigma\text{Fe}$ ratios indirectly from MORB oxygen fugacity rather than from direct measurements, and thus modern analytical uncertainties are not central to our interpretation. We consider Asimow's discussion to underscore the experimental challenges and variability in MORB redox measurements, whereas our study provides a thermodynamically constrained, long-term, and global framework for mantle oxidation evolution.

7. Regarding the inferred role of komatiite:

I am not sure that komatiitic oceanic crust could have subducted in the modern sense. Once metamorphosed, this lithology should convert to dense pyroxenite that is prone to delamination, as discussed by Foley et al 2003 Nature who also emphasised that continental crust production would have required partial melting of mafic, not ultramafic oceanic crust.

Regarding measured $\text{Fe}^{3+}/\Sigma\text{Fe}$ in komatiites, how do you distinguish oceanic, quasi syn-volcanic serpentinisation from post-emplacement (near) surface weathering-related serpentinisation?

Answer: We suggest that if Archean komatiites had not been subducted, their preservation proportion would likely be much higher than what is observed. In addition, altered komatiites, being rich in water, would have reduced densities, which could have facilitated their subduction. Recent studies have also shown that hydrated komatiites can generate TTGs (e.g., Tamblyn et al., 2023).

A detailed discussion in Tamblyn and Hermann (2023) was provided on the relationship between the rock's internal textures, geochemistry, and whole-rock ferric iron proportions, suggesting that the high $\text{Fe}^{3+}/\Sigma\text{Fe}$ ratios were produced by oceanic alteration reactions within the komatiites. They suggested that komatiites were most likely hydrothermally altered during or shortly after their eruption underwater (Tamblyn & Hermann, 2023 and refs in their study). In addition, these rocks contain abundant magnetite, which forms at temperatures of approximately 200 to 400 °C. These are typical seafloor hydrothermal conditions.

Comments on the manuscript with tracked changes:

There are some formatting issues related to $\text{Fe}^{3+}/\Sigma\text{Fe}$ (superscript missing in L25, bold font in L33) – other instances later in text

Answer: We thank you for noticing this formatting issue. The superscript and font inconsistencies for $\text{Fe}^{3+}/\Sigma\text{Fe}$ have been corrected in the clean version. The discrepancy arose because the tracked-changes and clean versions were not fully synchronized in the previous submission. We have now ensured consistency throughout the revised manuscript.

L28 “especially when” – ambiguous, “given that” or “assuming that”

Answer: It has been changed to “given that”.

L37 “tectonic activity”, “tectonic reorganization events” – vague, perhaps “Earth’s evolving tectonic regime” and “a transition from single-lid to mobile-lid tectonics” or something more concrete as per your Fig. 5. I’m not sure about the abstract word count, but you could specifically mention the Proterozoic tectonic lull or refer to the relatively well-known “boring billion”

Answer: The expression has been revised to be more specific, now reading “a transition from a single-lid to a mobile-lid tectonic regime.”

L58 “log units than QFM” should be “log units relative to the QFM”

Answer: It has been revised.

L71 sentence needs rewriting. The garnet-bearing mantle extends down further than 8-10 GPa, but metal saturation buffers the mantle $f\text{O}_2$ at higher pressure. The word “where” could be misunderstood to imply that it is above 8-10 GPa that $f\text{O}_2$ decreases

Answer: It has been revised.

L86-87 Birner et al. (ref 15) should be quoted here

Answer: It has been cited correctly here.

I note that the f in $f\text{O}_2$ is not italicized consistently, other state variables (P, T) also not italicized

Answer: The italicization of f in $f\text{O}_2$ and other state variables (P, T) has been corrected for consistency in the revised clean version.

L94 fO_2 is not measured directly in any of the studies. It is related to $Fe^{3+}/\Sigma Fe$ and some thermodynamic equilibrium; not sure what you mean by “corrected”

Answer: Has been modified.

L98 reference 18 is Canil et al. 1994 and is not recent. Ref 17 would be appropriate – please check all your references! [I later see that the references in the ms with tracked changes don't agree with the clean ms so I am no longer checking]

Answer: In the clean version, the reference at line 98 has been corrected to Wang et al. (2019). We have also carefully checked all references to ensure consistency between the text and the reference list.

L98-99 not sure what you mean by “using direct ratios... even greater discrepancies” – Ref 17 showed specifically the great impact of temperature (while also considering variations in source compositions, with minor effects on V-derived fO_2 estimates). The preceding lines already establish that other effects, such as pressure, were considered, so direct ratios were not used.

Answer: The reference has been corrected to Wang et al. (2019, now Ref. 18), which highlights the strong temperature effect on V-derived fO_2 estimates.

114 “under high-pressure conditions” – give the pressure, as high pressure means different things to different readers

Answer: Have done.

L129 which source pressure is given here? There is no single source pressure...The average pressure of melt extraction?

Answer: The pressure mentioned here refers to the average melt extraction pressure representing modern MORB generation.

L129 it is not the Archean melt that is extrapolated, but some parameter – formatting off in this and following sentence

Answer: It is the fO_2 and $Fe^{3+}/\Sigma Fe$ ratios that were extrapolated, not the melt itself. The description and formatting have been revised accordingly.

L159 “under uppermost mantle conditions” – the mantle is vast and most carbon is not in the 4+ state...

Answer: Our statement refers to the mantle redox reference state (uppermost mantle conditions) following Refs. 23–24, where carbon and sulfur are represented by their reference oxidation states (C^{4+} and S^{2-}). We have revised the sentence to clarify this point.

L188 it would be better to list some attributes that you used to identify MORB-like basalts (e.g. REE patterns, Nb/La)

Answer: Detailed criteria for identifying MORB-like mantle sources are provided in the Methods section, and we have now added a short summary of these features in the main text.

L196 “mutually” is redundant with “each other”

Answer: We have removed the redundant word “mutually”.

L213 and because its source and the melt extraction processes leading to MORB generation are relatively well-constrained

Answer: We have revised the sentence to include the additional rationale.

L276 what is “melt-extracted temperature”?

Answer: It has been revised to “temperature of melt extraction” for clarity.

L301 unclear what you mean by “within the fewer samples”?

Answer: It refers to the smaller number of samples in the V–Sc dataset compared with the V–Ti dataset, and we have revised the sentence for clarity.

L303 what is “primitive fO_2 ”? That of un/little differentiated melts?

Answer: That is oxygen fugacity not affected by post degassing and other effects.

L333-335 this begs the question how the alternative assumption affects the result. Here, I am particularly interested in the effect of redox melting, occurring as shallow as 120 km for a more reduced Archaean uppermost mantle source (Aulbach and Stagno 16, cited)? My feeling is that even at higher TP, the onset of partial melting in Archaean ridges was (dominantly) shallower than 120 km, so that no complication from redox melting is

expected.

Answer: The pressure–temperature conditions we modeled already cover depths shallower than ~120 km (Fig. A), corresponding to the range mentioned by you. Therefore, the potential effects of redox melting at such depths are inherently included in our results. Within this range, our calculations show that variations in $\text{Fe}^{3+}/\Sigma\text{Fe}$ are mainly governed by mantle potential temperature (T_p), rather than pressure.

L340 the reference to subarc depths comes a bit out of the blue, as up until here you seem mostly concerned with ridge settings and decompression melting

Answer: The discussion of subarc depths has been removed in the clean version of the manuscript.

L342-344, L350 I strongly disagree. For the solid-melt system combined, redox melting has of course no consequence, but if you oxidise C to C^{4+} , you induce redox melting, whereby CO_2 behaves like a highly incompatible component that is extracted. At the same time, you have a residue the $\text{Fe}^{3+}/\square\text{Fe}$ of which was reduced to make C^{4+} . Moreover, the remaining Fe^{3+} behaves like a moderately incompatible component. Combined, this implies a reduction of redox budget in the residue. However, as I note in the comment to L333, it may not be an issue for melting even of Archaean ambient mantle.

Answer: We agree that oxidation of C^0 to C^{4+} would, in principle, couple with Fe^{3+} reduction and might affect redox state after degassing. Please note that our discussion concerns the initial redox state and budget of mantle rocks before degassing, rather than that of the residues. Some other factors suggest that this process has negligible influence on the overall redox budget of the MORB-source mantle.

(1) The abundance of carbon in the MORB mantle source is extremely low compared with Fe, so even full oxidation of C^0 to C^{4+} would contribute little to the bulk redox balance.

(2) Under the P–T– $f\text{O}_2$ conditions relevant to MORB melting, C^{4+} is the thermodynamically stable carbon species—remaining stable down to $f\text{O}_2 \approx \text{QFM}-1.5$ to $\text{QFM}-2$ at 2–3 GPa (Maffei et al., 2024; Walters et al., 2020).

Therefore, within the typical melting regime of the MORB source, carbon redox transformations are limited, and variations in mantle redox state mainly reflect Fe valence changes rather than carbon oxidation.

To avoid introducing excessive uncertainty in the discussion, we have removed the related statements.

L353-355 this is an odd way to end this section. Either delete, or give your estimate for the Archaean mantle for

comparison.

Answer: Thanks for the suggestion. We have removed this sentence accordingly.

L388 “redox state”

Answer: It has been changed to $\text{Fe}^{3+}/\Sigma\text{Fe}$ Ratio

L388-440 this section still fails to convince, there are too many unknowns and no rock record. Given the uncertainties regarding disproportionation, the comparison to Venus is also not warranted. This is really a topic for a different paper in my opinion and distracts from the main points in the present manuscript.

Answer: We believe that the possible mechanisms must be discussed, as such discussion will help promote more refined investigations in the future.

L457-460 what do you mean by “making it difficult for bridgmanite to crystallize” – obviously, the magma ocean does not still exist?

Answer: We meant the magma ocean stage of the early Earth, not the present mantle. We have revised the sentence to clarify this point.

L474 it is not highly plausible for reasons I explain in point 6 above. At the very least, please moderate the language.

Answer: We thank you for the comment. The word “highly” has been removed to moderate the statement accordingly.

L490-491 clearly, alteration/weak metamorphism has had a major effect on the $\text{Fe}^{3+}/\square\text{Fe}$ of komatiites, the compositions of which we can measure today, such that e.g. fluid-mobile elements are typically not used to constrain komatiite petrogenesis.

Answer: Multiple lines of geological evidence demonstrate that serpentinization and associated oxidation processes were already widespread on the early Earth. Well-preserved Archean komatiites from different cratons record extensive hydration and oxidation of ultramafic lithologies, indicating that such processes were a global and fundamental feature of early lithospheric evolution (Pons et al., 2011; Tamblyn & Hermann, 2023).

L529 typo

Answer: It has been revised to redox budgets.

L1101-1102 syntax off – needs fixing (“indicating that”)

Answer: The sentence has been revised for correct syntax as suggested.

L1154 I would delete “historical” as this adjective is typically used in the context of human history. The word “trend” already implies the evolution

Answer: Thanks, “Historical” has been removed as suggested.

10 August 2025 Sonja Aulbach

Reviewer #2 (Remarks to the Author):

This is a much improved version of the original submission of this manuscript, and all of my previous comments have been addressed satisfactorily. I am happy to recommend this manuscript for publication in its current form.

Answer: We sincerely thank you for the positive evaluation and recommendation for publication. We greatly appreciate your previous insightful comments on our data, which have helped us improve the accuracy and strengthen the reliability of our results.

Reviewer #3 (Remarks to the Author):

I have read the rebuttal letter and the revised manuscript carefully. My major concerns are: 1) A close check of Fig. S3b and Fig. S4c shows that the $\text{Fe}^{3+}/\Sigma\text{Fe}$ for both the peridotitic mantle and the basalt whole rocks are highly variable. I suspect the alleged "doubled $\text{Fe}^{3+}/\Sigma\text{Fe}$ " of the mantle by the authors. It shows that no clear secular evolution of $\text{Fe}^{3+}/\Sigma\text{Fe}$ of the mantle. 2) If exists, the explanations for the tortuous evolution of the mantle are mainly qualitative. No figures were used to explain the possible causal relation between the elevated mantle $\text{Fe}^{3+}/\Sigma\text{Fe}$ and tectonic parameters.

Answer: We sincerely thank you for the careful evaluation of our rebuttal letter and the revised manuscript.

1) Regarding Figures 3, S3b, and S4c, we would like to emphasize that the “doubling” we refer to represents the average state of the MORB-source mantle and the majority of samples, rather than individual extreme values. For example, approximately 90% of basalts derived from the Archean mantle have $\text{Fe}^{3+}/\Sigma\text{Fe}$

ratios below 0.06 (Table S3), whereas in the modern, MORB $\text{Fe}^{3+}/\Sigma\text{Fe}$ ratios are around 0.10–0.14. The few Archean basalts with higher $\text{Fe}^{3+}/\Sigma\text{Fe}$ ratios (0.06–0.10) likely reflect localized, more oxidized mantle domains, which may have acquired additional redox budget through specific processes such as subduction-related inputs, as we discussed. Therefore, our conclusion regarding a doubling of the mantle's $\text{Fe}^{3+}/\Sigma\text{Fe}$ ratio remains robust. It is also important to note that long-term secular trends are not always visually apparent in scatter plots, because several extreme values in the Precambrian dataset can obscure the overall pattern. However, a direct comparison between Archean and modern samples clearly (Table S3) reveals the increase in mantle and basalt oxidation state through time.

2) As we emphasized earlier, the primary objective and conclusion of this study are to determine whether a long-term evolution occurred in the mantle $\text{Fe}^{3+}/\Sigma\text{Fe}$. Our conclusion is that the mantle experienced a doubling process in $\text{Fe}^{3+}/\Sigma\text{Fe}$. However, we believe that it remains necessary to further discuss the possible mechanisms responsible for this result, rather than leaving the topic unresolved. In the previous revised version, both mantle plume and subduction-related models were discussed as potential explanations, but these were presented as interpretative discussions, not definitive conclusions.

Minor points:

1) The authors used many vague nouns, such as "oxidized characteristics, L.44" and "redox characteristics, L752."

Answer: The term “oxidized characteristics” was used intentionally in the introduction as a broad and general description to introduce the long-standing debate on the oxidation state of the mantle. In the main text, however, we have replaced such general expressions with more precise terms, including “redox budget” and “ $\text{Fe}^{3+}/\Sigma\text{Fe}$ ratio”, to ensure quantitative clarity throughout the manuscript.

2) L.45 "the evolution trends of mantle that forms mid ocean ridge basalt" could be "the secular redox evolution of the MORB mantle"

Answer: We have revised the sentence as suggested.

3) L.47 what's the meaning of "exceeding 0.5 log units than QFM", do you mean "0.5 log unit increase in $f\text{O}_2$ "? the same for L.435

Answer: Following Reviewer #1's comment, we have revised the expression to “log units relative to the QFM.”

4) The last paragraph of the main text is really confusing, is the redox state an indicator or a driving force? In the first sentence, the authors claimed that "shifts in mantle redox state track major tectono-magmatic events", but in the last sentence, they said "this underscores the fundamental role of mantle redox evolution in driving the coupled ..."

Answer: The redox evolution of the mantle plays a dual role in Earth's system. It primarily responds to major tectono-magmatic reorganizations (thus serving as an indicator of deep geodynamic changes), but it also feeds back on surface environments by influencing oxygen fugacity and atmospheric evolution. To clarify this logic, we have slightly revised the last paragraph to emphasize this dual role more clearly.

Hope the above suggestions will be helpful to the authors.

We sincerely thank all three reviewers for their constructive comments and thoughtful suggestions. Their feedback has greatly helped us improve the clarity, robustness, and overall quality of our data interpretation and conclusions.

Reference:

- Ague, J. J., Tassara, S., Holycross, M. E., *et al.* (2022). Slab-derived devolatilization fluids oxidized by subducted metasedimentary rocks. *Nature Geoscience*, *15*(12), 1011–1017.
- Asimow, P. D. (2021). The petrological consequences of the estimated oxidation state of primitive MORB glass. In R. Moretti & D. R. Neuville (Eds.), *Magma redox geochemistry* (Vol. 256, pp. 141–166). *Geophysical Monograph Series*.
- Evans, K. A. (2006). Redox decoupling and redox budgets: Conceptual tools for the study of earth systems. *Geology*, *34*(6), 489–492.
- Evans, K. A. (2012). The redox budget of subduction zones. *Earth-Science Reviews*, *113*(1-2), 11–32.
- Jugo, P. J., Luth, R. W., & Richards, J. P. (2005). Experimental data on the speciation of sulfur as a function of oxygen fugacity in basaltic melts. *Geochimica et Cosmochimica Acta*, *69*(2), 497–503.
- Maffeis, A., Frezzotti, M. L., Connolly, J. A. D., Castelli, D., & Ferrando, S. (2024). Sulfur disproportionation in deep COHS slab fluids drives mantle wedge oxidation. *Science Advances*, *10*(12), eadj2770.
- Nash, W. M., Smythe, D. J., & Wood, B. J. (2019). Compositional and temperature effects on sulfur speciation and solubility in silicate melts. *Earth and Planetary Science Letters*, *507*, 187–198.
- Pons, M.-L., Quitté, G., & Fujii, T., *et al.* (2011). Early Archean serpentine mud volcanoes at Isua, Greenland, as a niche for early life. *Proceedings of the National Academy of Sciences*, *108*(43), 17639–17643.
- Tamblyn, R., & Hermann, J. (2023). Geological evidence for high H₂ production from komatiites in the Archaean. *Nature Geoscience*, *16*(12), 1101–1106.
- Tamblyn, R., Hermann, J., Hasterok, D., Sossi, P., Pettke, T., & Chatterjee, S. (2023). Hydrated komatiites as a source of water for TTG formation in the Archaean. *Earth and Planetary Science Letters*, *603*, e2023EPSL117982.
- Tomkins, A. G., & Evans, K. A. (2015). Separate zones of sulfate and sulfide release from subducted mafic oceanic

crust. *Earth and Planetary Science Letters*, 428, 73-83.

Wang, J., Xiong, X., Takahashi, E., Zhang, L., Li, L., & Liu, X. (2019). Oxidation state of arc mantle revealed by partitioning of V, Sc, and Ti between mantle minerals and basaltic melts. *Journal of Geophysical Research: Solid Earth*, 124(5), 4617-4638.

Zhang, F., Lai, S., Stagno, V., Chen, L., Zhang, C., Zhu, R., ... & Wang, J. (2024). The Redox state of the asthenospheric mantle and the onset of melting beneath mid-ocean ridges. *Journal of Geophysical Research: Solid Earth*, 129(5), e2023JB027033.

**Figure**

**Fig. 1** The influence of different thermodynamic conditions on oxygen fugacity and V/Sc: (A) P-T phase

fO₂ conditions better than “QFM levels”

pressures, and melting conditions under varying QFM levels^{13, 17}

Sc

This caption is not nearly sufficiently descriptive. What are the stars and arrows, what are the black solid lines and white stippled lines, what is the thick red stippled line (I can gather that, but you are angling for a wide, non-specialised readership). The readers are left to their own device to figure everything out. It would help to let them know that this is a phase diagram with a subsolidus and a supersolidus region, the latter with colour shades and isopleths

what is “a depleted mantle” - the Depleted Mantle reservoir for which there is an estimate? Otherwise, depleted mantle could be anything from ultrarefractory to mildly depleted. Where is the 0.04 coming from? Reference? If the composition of this depleted mantle, which is the basis of your phase eq. modelling, is found in an appendix (which it should), then please refer to it here - Fe³⁺/Fe(T) of 0.04 insufficiently describes the system. It would really help the reader to understand panel A if you chose a better description in the caption. It looks that you are looking at mantle residues that are increasingly depleted as a function of increasing distance from the solidus.

The topology of the fO₂ isopleths would merit description and discussion. Eg. there is a marked bend at opx-out, on the opx-bearing side of which fO₂ is very sensitive to P but not T and on the opx-free side of which fO₂ is very sensitive to T but not P. There are not many settings in which you make a dunite residue, but it is not unheard of.

Why are there no isopleths in the subsolidus region? It would be very interesting what you find compared to Stolper+20

a lot of the font in these panels is much too small. Since minor rock types are not discernible, I wonder whether it is useful to display them as part of a figure in the main document? Perhaps remove here and refer to supplements for those wanting to know about minor rock types not essential to the main thrust of your paper?

make it clear that these are not for ambient mantle only. Perhaps refer to "possible Archean and modern Tp conditions"

**Fig. 2** Reference numerical experiments of mid-ocean ridge spreading under Archean and modern T_p
conditions: (A-E) Rock types and temperature fields in the experiments after ~ 6.5 Ma; (F-J) Spatial
information of newly formed basalt and its in-situ temperature, pressure, and whole-rock Fe^{3+}/Fe_{tot} during melt
extraction marked, with initial mantle whole-rock $Fe^{3+}/Fe_{tot} = 0.04$ and 0.02 , respectively; (K-O) Bar
frequency diagrams of the whole-rock Fe^{3+}/Fe_{tot} evolution of basaltic melts under varying T_p and initial mantle
whole-rock Fe^{3+}/Fe_{tot} .

F-J unclear what this shows: melt present during polybaric melting that would ultimately pool to form an aggregated melt? All the melts between initial and final pressure of melting? Are they there simultaneously over the pressure interval. One would like to understand....

What motivates the choice of 0.04 and 0.02? What are the individual dots? Pixels from the numerical model?

We should understand this from the caption alone

K-O What is in the number making up the bars? All the individual dots in F-J?

should be consistent with use of AE vs. BE. I think you dominantly use AE, so should be Archaean

it seems strange to infer WR Fe³⁺/Fe(T) based on fO₂, and then calculate fO₂ from WR Fe³⁺/Fe(T). Even if one is based on a V-based redox proxy and the other is based on Fe³⁺, this must imply some circularity. Note that it is incorrect to back-calculate from the fO₂ calculated based on melt V/Ti or V/Sc redox proxy because this is already recording source fO₂ and assumed to remain relatively unchanged so long as the basalt has not proceeded to cpx fractionation (this is different from ol-melt V oxybarometry which reflects fO₂ of the melt at eruption).

Refer to a text and supp table where this can be followed up

based on what oxybarometer?

redox evolution of collected Precambrian basalts; (A) Age of basalt samples; (B) Oxygen fugacity of basalts from thermodynamic calculations; (D) Mantle oxygen fugacity from thermodynamic calculations; (E) Whole-rock Fe³⁺/Fe_{tot} of basalts from thermodynamic calculations

what is the star, what is the red rectangle, what are the solid and dashed black lines?

**Fig. 4** Thermodynamic experiments on melt fractionation: (A) Whole-rock Fe^{3+}/Fe_{tot} of the residues after
 different melt fractionation from depleted mantles (initial $Fe^{3+}/Fe_{tot} = 0.04$ and 0.01 , respectively) at 3-5 GPa;
 (B) Comparison of the oxygen fugacity and spinel Cr# in the residues (F=26% and 38%) with the natural
 samples of refractory mantle residues. The data for natural samples from Hess Deep, Gakkel Ridge and
 Southwest Indian Ridge (SWIR) was collected by ref. ¹⁵.

Unclear what you mean by melt fractionation? melts fractionally crystallise melts and undergo differentiation, elements are isotopes are fractionated from one another. Are you showing the evolution of $Fe^{3+}/Fe(T)$ in the mantle as a function of %melt extracted? And is this isobaric batch melting? Would it not make more sense to tie the %melt extracted to decreasing pressure as applies to decompression melting?

It is not clear what the purpose of panel B is. 26 and 38% for which P?

please be aware that any surface oxidised flux that you want to stick into the huge mantle reservoir to explain its redox evolution will (1) correspondingly retard the oxygenation of the ocean-atmosphere system and (2) resurface at some stage due to mantle convection, so result in no net redox budget change on timescales of ?0.5 Ga or however long it takes to complete the cycle

a cartoon is schematic by nature

**Fig. 5** The tortuous evolution of the Earth's redox state during Earth history and its correlation with major

geological events: (A) A cartoon schematic diagram showing the coupled evolution of mantle redox state and

major geological events; (B) The trends in atmospheric and biological evolution; (C) The record of mantle

whole-rock Fe^{3+}/Fe_{tot} since the Hadean; (D) Age distribution diagram of arc (basaltic) magma samples since

the Archean; (E) Trends in W and Nd isotopes since the Archean; (F) Metamorphic T/P ratio, cooling rate,

exhumation rate, and the first occurrence of characteristic metamorphic rocks (marked with stars); (G)

Historical trends in zircon distribution, seawater Sr isotopes, and the number of passive margins.

Historical is not a good word here

clearly this synopsis is based on a lot of others' works which should be duly referenced here

In general, the captions aren't very informative. Don't ask the reader to have read the entire main document to make sense of these figures! They should to some extent be self-explanatory

for completeness, refer to both panels (say what the difference is between the two). Be explicit about what the rock records are (modern MORB and purported ancient equivalents?)

Fig. S1 P-T output of reference model (circle) and rock records (triangle).

please be consistent with the lettering (capitals in panels, but not in caption)

Fig. S2 Changes of oxygen fugacity and $\text{Fe}^{3+}/\text{Fe}_{\text{tot}}$ ratio in MORB-like basalts and mantle source over time: (a) the oxygen fugacity of basalt varying over time; (b) whole rock $\text{Fe}^{3+}/\text{Fe}_{\text{tot}}$ of basalt corresponding to different oxygen fugacity; (c) the whole rock $\text{Fe}^{3+}/\text{Fe}_{\text{tot}}$ of basalt varying over time; (d) the oxygen fugacity of mantle varying over time; (e) whole rock $\text{Fe}^{3+}/\text{Fe}_{\text{tot}}$ of mantle corresponding to different oxygen fugacity; (f) the whole rock $\text{Fe}^{3+}/\text{Fe}_{\text{tot}}$ of mantle varying over time.

fix the caption. The font in the panel is VERY small. You could indicate what the oxygen fugacity is based on (which redox proxy/oxybarometer you used). What is the solid vs. stippled line? What are the blue shades?

are these now measured bulk Fe3+/Fe total? Be explicit! What are the "upper mantle" samples? Orogenic, ophiolites, xenoliths?

Fig. S3 The whole rock Fe^{3+}/Fe_{tot} in Precambrian MORB-like basalt and mantle recorded in three databases. Different colored lines represent the Archean average value from three databases.

3 databases in the references listed on top of panel A?

Fig. S4 Record results of V-Sc oxybarometer and corresponding whole rock Fe^{3+}/Fe_{tot} of MORB-like basalt.

whose oxybarometer? Wang+19 considering also T and composition? Please provide more info, see comment to Fig. S2

Fig. S5 Phase diagram of mantle remnants after high degree of partial melting.

There is not nearly enough info here.
 Explain the isopleths, for which bulk composition and with which thermodynamic model? Give mineral abbreviations. Fix Gpa in panel (P should be capital).
 A percentage is not a melt fraction. Should be 0.26 etc.

Fig. S6 Numerical simulation results of early Archean geodynamic evolution.

Provide some info in the caption of where more details on the setup etc. of the model are described

Fig. S7 Initial model of mid ocean ridge expansion experiment.

Fig. S8 The comparison between thermodynamic simulation and experimental petrology data (Davis and Cottrell, 2018; ref. 65) indicates that the simulation method in this study will not deviate due to the inappropriate Fe^{3+} behavior of spinel.

Fig. S9 The thermodynamic method and two empirical methods calculated the whole-rock $\text{Fe}^{3+}/\text{Fe}_{\text{tot}}$ of basalt under the same P-T-fO₂ conditions. The empirical results used method of Kress and Carmichael (1991) come from Zhang et al. (2024). The empirical results used method of O'Neill et al. (2018) and the thermodynamic results are calculated by this study. These calculation results indicate that the method of Kress and Carmichael (1991) significantly overestimates the whole rock $\text{Fe}^{3+}/\text{Fe}_{\text{tot}}$ of basalt.

using the

maybe ask your colleague Paolo Sossi about this substantial discrepancy - I find it too much to be straightforward believable but don't know enough about thermodynamics

References

Davis, F. A. & Cottrell, E. Experimental constraints on the use of spinel as an oxygen barometer: Implications for spinel thermodynamic models and Fe³⁺/Fe²⁺ compatibility during generation of upper mantle melts. *Am. Mineral.* **103**, 1056–1067 (2018).

Gao, L., Liu, S., Cawood, P. A., Hu, F., Wang, J., Sun, G. & Hu, Y. Oxidation of Archean upper mantle caused by crustal recycling. *Nat. Commun.* **13**, 3283 (2022).

Kress, V. C. & Carmichael, I. S. E. The compressibility of silicate liquids containing Fe₂O₃ and the effect of composition, temperature, oxygen fugacity and pressure on their redox states. *Contrib. Mineral. Petrol.* **108**, 82–92 (1991).

O'Neill, H. St. C., Berry, A. J. & Mallmann, G. The oxidation state of iron in mid-ocean ridge basaltic (MORB) glasses: implications for their petrogenesis

and oxygen fugacities. *Earth Planet. Sci. Lett.* **504**, 152–162 (2018).

Wang, J., Xiong, X., Takahashi, E., Zhang, L., Li, L. & Liu, X. Oxidation state of arc mantle revealed by partitioning of V, Sc, and Ti between mantle minerals and basaltic melts. *J. Geophys. Res. Solid Earth* **124**, 4617–4638 (2019).

Zhang, F., Stagno, V., Zhang, L., Chen, C., Liu, H., Li, C. & Sun, W. The constant oxidation state of Earth's mantle since the Hadean. *Nat. Commun.* **15**, 6521 (2024).

COMMENTS ON S TABLES

Table Materials properties: Although the units could be inferred from the columns, it would probably be better to explicitly state what the units are when formulae are displayed using P T etc.

Table with databases: Since there are no explanatory headers or footers, it would be useful to add a sheet where you explain what is shown in the various sheets. You could make it a little easier for the readers

Databases: did you ensure there was no overlap in sample identities?

Table Hadean zircon: Header: what is thermodynamic? Also elsewhere in the xls file Is the Ce/Ce* really relative to CHUR (Ce isotope ratios are but that is something different)? Or chondrite?

xls file showing inputs and results:

No headers or footers, readers left to their own devices figuring out what they're looking at.

Sheet 1: You could adjust the number of significant digits in the results comma used as decimal in column F, Chinese characters in column E

half spreading rate needs a unit, nothing in column J, K, L - maybe you forgot to delete this sheet altogether?

I think the database could have been more rigorously screened for samples that are clearly not mantle-derived melts that could have formed by fractionation of olivine +/- plagioclase only. It is irrelevant whether these databases were published and proposed before by others as such. There are samples with very high TiO₂ or K₂O at high MgO that are either highly altered, or have accumulated oxides. I also think two samples with high MgO could be cumulates. From my work on eclogite xenoliths (metamorphosed oceanic crust predominantly formed around 3.0-2.7 Ga ago), I can see that those not affected by mantle metasomatism have a bulk rock maximum of 15 wt.% MgO. While I understand the desire to retain as many samples as possible, there are some 7 samples that I would exclude on the above basis